# Combining discovery and targeted proteomics reveals a prognostic signature in oral cancer

Carolina Moretto Carnielli [1], Carolina Carneiro Soares Macedo[1,2], Tatiane De Rossi[1], Daniela Campos Granato[1], César Rivera [1,2], Romênia Ramos Domingues[1], Bianca Alves Pauletti[1], Sami Yokoo[1], Henry Heberle [3], Ariane Fidelis Busso-Lopes[1], Nilva Karla Cervigne[2,4], Iris Sawazaki-Calone[5], Gabriela Vaz Meirelles[1], Fábio Albuquerque Marchi[6], Guilherme Pimentel Telles[7], Rosane Minghim[3], Ana Carolina Prado Ribeiro[8,9], Thaís Bianca Brandão[8], Gilberto de Castro Jr[10], Wilfredo Alejandro González-Arriagada[11], Alexandre Gomes[12], Fabio Penteado[12], Alan Roger Santos-Silva[2], Márcio Ajudarte Lopes[2], Priscila Campioni Rodrigues[13,14], Elias Sundquist[13,14], Tuula Salo[13,14,15], Sabrina Daniela da Silva[16,17], Moulay A. Alaoui-Jamali[17], Edgard Graner[2], Jay W. Fox[18], Ricardo Della Coletta[2] & Adriana Franco Paes Leme [1]

Different regions of oral squamous cell carcinoma (OSCC) have particular histopathological and molecular characteristics limiting the standard tumor—node—metastasis prognosis classification. Therefore, defining biological signatures that allow assessing the prognostic outcomes for OSCC patients would be of great clinical significance. Using histopathology-guided discovery proteomics, we analyze neoplastic islands and stroma from the invasive tumor front (ITF) and inner tumor to identify differentially expressed proteins. Potential signature proteins are prioritized and further investigated by immunohistochemistry (IHC) and targeted proteomics. IHC indicates low expression of cystatin-B in neoplastic islands from the ITF as an independent marker for local recurrence. Targeted proteomics analysis of the prioritized proteins in saliva, combined with machine-learning methods, highlights a peptide-based signature as the most powerful predictor to distinguish patients with and without lymph node metastasis. In summary, we identify a robust signature, which may enhance prognostic decisions in OSCC and better guide treatment to reduce tumor recurrence or lymph node metastasis.

[1] Laboratório de Espectrometria de Massas, Laboratório Nacional de Biociências (LNBio), Centro Nacional de Pesquisa em Energia e Materiais (CNPEM), Campinas, 13083-970 SP, Brazil. [2] Departamento de Diagnóstico Oral, Faculdade de Odontologia de Piracicaba Universidade Estadual de Campinas (UNICAMP), Piracicaba, 13414-903 SP, Brazil. [3] Instituto de Ciências Matemáticas e de Computação, Universidade de São Paulo (USP), São Carlos, 13560-970 SP, Brazil. [4] Departamento de Morfologia e Patologia Básica, Faculdade de Medicina de Jundiaí (FMJ), Jundiaí, 13202-550 SP, Brazil. [5] Centro de Ciências Biológicas e da Saúde, Faculdade de Odontologia, Universidade Estadual do Oeste do Paraná, Cascavel, 85819- 110 PR, Brazil. [6] Centro Internacional de Pesquisa-CIPE - A.C.Camargo Cancer Center and Instituto Nacional de Ciência e Tecnologia em Oncogenômica (INCiTO), São Paulo, 01508-010 SP, Brazil. [7] Instituto de Computação, Universidade Estadual de Campinas (UNICAMP), Campinas, 13083-852 SP, Brazil. [8] Serviço de Odontologia Oncológica, Instituto do Câncer do Estado de São Paulo, ICESP-FMUSP, São Paulo, 01246-000 SP, Brazil. [9] Universidade Brasil, Fernandópolis, 15600-000 SP, Brazil. [10] Serviço de Oncologia Clínica, Instituto do Câncer do Estado de São Paulo, Octavio Frias de Oliveira, ICESP, São Paulo, 01246000, Brazil. [11] Facultad de Odontología, Universidad de Valparaíso, Valparaíso, 2340-000, Chile. [12] Waters Corporation, Campinas, 13070-091 SP, Brazil. [13] Cancer and Translational Medicine Research Unit, University of Oulu, Oulu, FI-90014, Finland. [14] Medical Research Center, Oulu University Hospital, Oulu, FI-90014, Finland. [15] Department of Oral and Maxillofacial Diseases, Clinicum, University of Helsinki, Helsinki, FI-00014, Finland. [16] Department of Otolaryngology-Head and Neck Surgery, Jewish General Hospital, McGill University, Montréal, H4A 3J1, Canada. [17] Lady Davis Institute for Medical Research and Segal Cancer Center, SMBD Jewish General Hospital, Departments of Medicine and Oncology, Faculty of Medicine, McGill University, Montreal, H3T 1E2, Canada. [18] Department of Microbiology, Immunology and Cancer Biology, University of Virginia, Charlottesville, 22903 VA, USA. These authors contributed equally: Carolina Moretto Carnielli, Carolina Carneiro Soares Macedo. Correspondence and requests for materials should be addressed to A.F.P.L. (email: adriana.paesleme@lnbio.cnpem.br)

O ral squamous cell carcinoma (OSCC) is the most common type of head and neck malignant tumor and is ranked the eighth leading cause of cancer worldwide. OSCC exhibits high prevalence and morbidity, with 300,000 new cases and 145,000 deaths per year worldwide[1]. Standard multimodal management of OSCC is based on the tumor−node−metastasis (TNM) classification[2], in which the tumor size and location and the presence of metastasis are used to define OSCC prognosis and treatment in the clinical setting[3]. However, this system has several flaws, such as patients with the same TNM stage exhibit different clinical behaviors, different treatment responses, and substantial variability in clinical outcomes[4,5]. Despite efforts to improve imaging and therapeutic modalities, OSCC prognosis, including survival rates, remains poor and may widely vary, even in the early stages of the disease, e.g., 20−40% of occult metastases are detected at the initial diagnosis[5–8]. Furthermore, OSCC recurrence rates range from 18 to 76% in patients undergoing standard treatment, and local relapse represents a clinical challenge for therapeutic management[5]. Thus, the identification of complementary biological signatures that assist in the prognostic prediction of patients with OSCC is needed.

Histopathological parameters have previously been employed in numerous studies that aimed to improve the OSCC prognostic prediction and overcome the shortcomings of the TNM staging system[4,9–11]. Histopathological-based models may be used to stratify patients into low- and high-risk classes[9] and further identify patients at risk of aggressive early-stage OSCC, thus contributing to disease predictability in terms of clinical progression and treatment outcomes. Moreover, histopathology contributes to the molecular characterization of specific tumor regions, such as the invasive tumor front (ITF) and the inner tumor, which exhibit different morphologic and molecular features[12,13]. For example, the epithelial−mesenchymal transition phenotype presents more cells with a lower degree of differentiation and greater cell dissociation in the ITF than in other tumor areas[14,15].

Grading systems based on the ITF[4,16] have demonstrated reliable predictive value for OSCC prognosis, thus highlighting the importance of this area for molecular profiling and for the identification of potential biomarkers, as it is considered a key region in the dynamic progression of malignant tumors[12,13,17]. The presence of neoplastic islands, classified as large or small according to the number of cells in the ITF, has been described as the most aggressive pattern compared to tumors with a more uniform growth pattern, as tumor invasion occurs in a more widespread manner as cellular islands or single cells[4]. Furthermore, there is evidence that components of the tumor stroma critically influence carcinogenesis and the malignant phenotype in multiple stages of tumor development and progression[18,19]. The complex interactions among tumor cells and the various types of cells and matrix components within the microenvironment play important roles in cancer onset, progression, invasion, and metastasis[20,21].

In addition to the routine use of tissue histopathology, saliva testing may represent a promising noninvasive tool to validate prognostic biomarkers, such as proteins, lipids, mRNA, miRNA, and exosomes, and better classify patients into low- and high-risk groups[22–25].

In this study, we combine discovery and targeted proteomics approaches to identify prognostic signatures for OSCC patients. In the initial discovery phase, we integrate knowledge of the histopathology, discovery proteomics analysis of formalin-fixed paraffin-embedded (FFPE) OSCC tissues, and clinical features of patients. Assessing the protein profiles of large and small neoplastic islands and their surrounding stroma by combining laser microdissection (LMD) and proteomics reveals several proteins—including CSTB, NDRG1, LTA4H, PGK1, COL6A1, ITGAV, and MB—with distinct expression patterns between ITF and inner tumor, suggesting a potential prognostic value by clinicopathological association analysis. In the subsequent targeted phase, we use two follow-up approaches to verify these signatures in two independent patient cohorts. First, analysis of clinical significance and immunohistochemical staining are performed in a 125-OSCC patient cohort, indicating CSTB, at low expression levels in the ITF, as an independent marker for local recurrence. Second, selected reaction monitoring mass spectrometry (SRM-MS) is applied to study the abundance of the above-mentioned seven proteins in saliva samples from an independent 40-OSCC patient cohort. Analyzing the SRM-MS results with machine-learning approaches demonstrates that a combination of LTA4H-, COL6A1-, and CSTB-specific peptides in saliva are able to distinguish patients with and without lymph node metastasis with good estimated prediction performance, outperforming predictors based on individual or grouped proteins.

Taken together, our results identify a prognostic signature that may assist in the clinical decision-making process leading to appropriate treatment, thus improving the prognosis and survival of patients with OSCC.

## Results

**Spatial characterization of OSCC by discovery proteomics.** We aimed to identify proteins that were spatially organized in distinct histological areas of tongue squamous cell carcinoma. For this purpose, we mapped the proteome of neoplastic islands and their surrounding tumor stroma from the ITF and inner tumor FFPE tissue samples from 20 patients (Fig. 1a, b; Supplementary Fig. 1; Supplementary Data 1). Using histology-guided LMD, we isolated six different areas of the tumor: (1) small neoplastic islands from the ITF; (2) large neoplastic islands from the ITF; (3) small neoplastic islands from the inner tumor; (4) large neoplastic islands from the inner tumor; (5) stroma from the ITF; and (6) stroma from the inner tumor.

The six proteomes were analyzed using quantitative mass spectrometry (Supplementary Data 2) and label-free protein quantitation (LFQ intensity) to compare the relative abundance of the proteins (Fig. 1c–e). The reproducibility and correlation coefficient among the LFQ intensities of discovery proteomics data are illustrated in Fig. 2e, f and in Supplementary Data 3, 4.

Combining the data for small and large neoplastic cells resulted in the quantitation of 2049 proteins from the ITF and inner tumor. After excluding reverse sequences and those identified "only by site" entries, and considering proteins with at least ten valid LFQ intensity values in at least one group (20 samples), 799 proteins were confidently identified (ITF and inner tumor) (Supplementary Data 5−9). For the tumor stroma dataset (ITF and inner tumor), 1733 proteins were quantified. After excluding reverse sequences and those identified "only by site" entries, and considering proteins with at least eight valid values in at least one group (17 samples), 704 proteins were quantified (Supplementary Data 10−14).

The proteomic data analysis identified common and exclusive proteins from neoplastic islands (Fig. 2a) and tumor stroma (Fig. 2b, Supplementary Fig. 2). The filtered dataset of the neoplastic islands from the ITF and inner tumor was subjected to statistical analysis using Student's $t$ test ($P$ value < 0.05), which resulted in 32 proteins with differential abundances (Supplementary Data 8). Similarly, paired Student's $t$ test ($P$ value < 0.05) indicated 101 proteins that were differentially expressed between the tumor stroma from the ITF and the inner tumor (Supplementary Data 13).

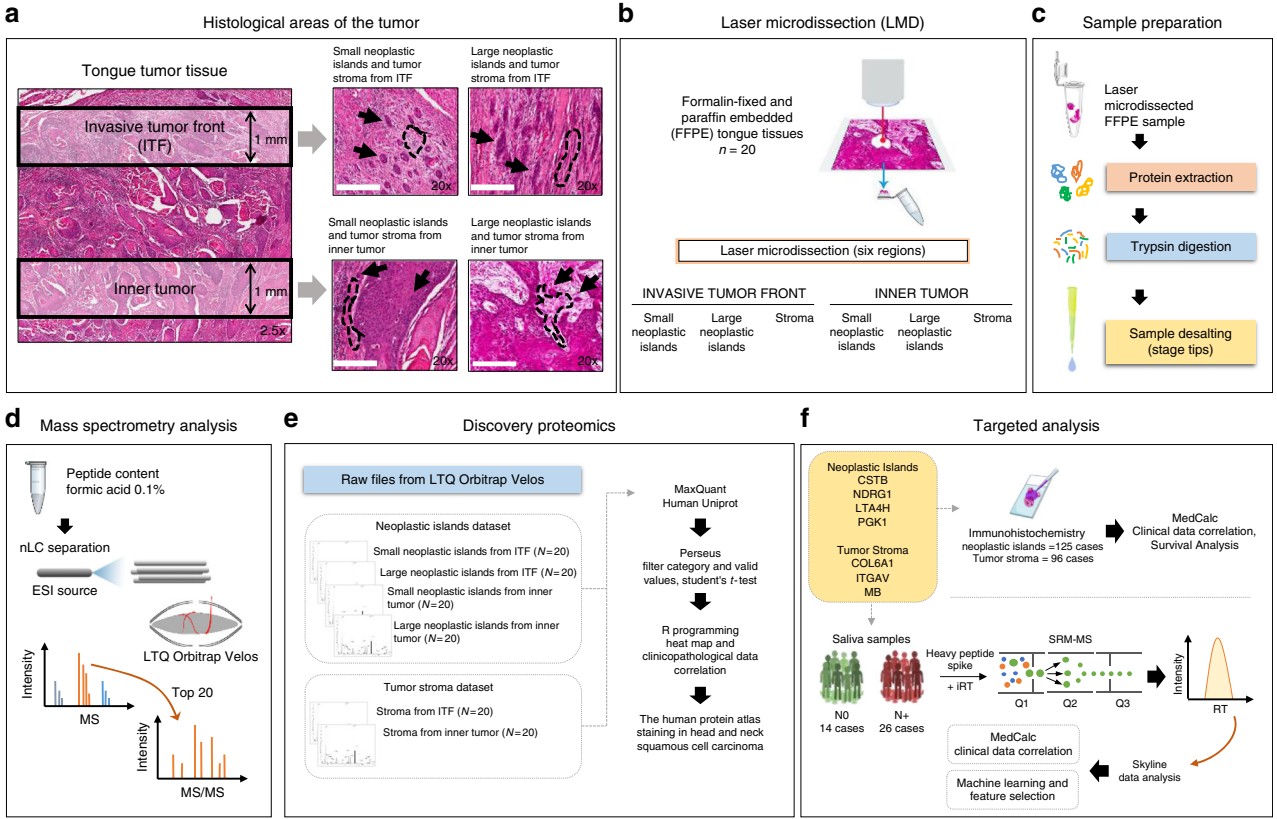

**Fig. 1** Experimental design. **a** The ITF was delimited as a 1 mm depth from the edge of the tumor slice, and the inner tumor was defined as up to 1 mm from the epithelial tumor tissue origin. In more detail, the ITF and inner tumor with small and large neoplastic islands (arrows) are surrounded by the tumor stroma (dashed lines) (scale bars, 200 μm). **b** Laser microdissection of the six regions of interest. **c, d** Protein extraction from microdissected tissues and trypsin digestion. The peptide mixture was desalted in stage tips and analyzed by liquid chromatography coupled with tandem mass spectrometry (LC-MS/MS). **e** To select protein targets, MaxQuant and the Perseus package were used to identify and quantify proteins, and R software was used for statistical analysis of the clinicopathological parameters and for analysis of the proteins with positive staining for OSCC tissues in The Human Protein Atlas. **f** Targeted proteins were evaluated following two different strategies by immunohistochemistry in OSCC tissues and by SRM of saliva samples from OSCC patients with (N+) or without (N0) lymph node metastasis

Unsupervised hierarchical clustering analysis of the identified proteins showed not only the proteomic diversity between samples from the ITF and the inner tumor, but also the variation within the ITF samples and the inner tumor samples for both neoplastic island (Fig. 2c; Supplementary Fig. 3a) and tumor stroma proteomes (Fig. 2d; Supplementary Fig. 3b). The lower clustering identified in neoplastic islands compared with tumor stroma may be associated with different abilities to extract proteins from FFPE tissues and may also reflect the intrinsic tumor heterogeneity[7,12], as OSCCs are known for their biological variability, which leads to specific clinical behaviors, i.e., it has been observed that tumors at the same stage may present different clinical outcomes[4,5,26].

To investigate whether the biological processes could spatially separate neoplastic islands from tumor stroma, we searched for biological processes in the Gene Ontology (GO) database that were enriched for proteins uniquely identified in each proteome. Cellular metabolic processes primarily represented the neoplastic island proteins, whereas cellular adhesion processes and protein cleavage processes overrepresented the tumor stroma proteins (Fig. 2g; Supplementary Data 15, 16), which indicate that proteome annotation discriminates neoplastic islands from adjacent stroma. Moreover, the analysis of the 601 significant proteins, which were significantly different between the neoplastic islands and the tumor stroma (Student's t test, P value < 0.05), also indicates metabolic processes overrepresented, among other

annotations, for upregulated proteins of neoplastic islands (Fig. 2g; Supplementary Data 17).

We used linear regression to analyze the proteome LFQ dataset and clinicopathological data to identify the proteins associated with patient features (Table 1; Supplementary Fig. 4). The majority of proteins (ACTR2, CSTB, LTA4H, PGK1, NDRG1, FSCN1, ITGAV, THBS2) significantly associated with clinical parameters showed lower expression in the ITF of the tumor stroma or neoplastic islands, with the exception of COL6A1, COL1A2, S100A8, S110A9, and MB.

**Prioritization of proteins for IHC and SRM-MS analysis.** The targeted proteins evaluated in the subsequent steps of verification using immunohistochemistry (IHC) in a 125-patient cohort and SRM in an independent 40-patient cohort were selected if they filled the following criteria: (1) only proteins with different protein abundances between the ITF and the inner tumor in the discovery phase (Student's t test, P value < 0.05); (2) only proteins that present a significant association with clinical characteristics of patients (Linear regression, P value < 0.05, R < −0.7 or 0.7 < R and $R^2$ > 0.4) (Table 1); (3) only proteins with positive staining of squamous cell carcinoma in HNSCC in The Human Protein Atlas (https://www.proteinatlas.org/); and (4) only proteins not cited or cited only in limited studies related to oral cancer (Supplementary Data 18).

Cystatin-B (CSTB), leukotriene A-4 hydrolase (LTA4H), protein NDRG1 (NDRG1), and phosphoglycerate kinase 1 (PGK1) from the neoplastic island dataset and collagen alpha-1 (VI) chain (COL6A1), integrin alpha-V (ITGAV) and myoglobin (MB) from the tumor stromal dataset were prioritized (Fig. 3; Supplementary Fig. 5). All these proteins, according to the literature and to the domain predictions performed here, are nonclassically secreted (Supplementary Data 18).

**IHC analysis of prioritized proteins.** IHC analysis was performed using 125 FFPE OSCC cases for neoplastic island proteins and 96 FFPE OSCC cases for tumor stroma proteins (Supplementary Data 19). For the IHC analysis, the adopted score system described in Supplementary Data 20[27] was used to differentiate the staining among the two regions, the ITF and the inner tumor, in a blinded and independent manner by three pathologists (kappa = 0.706). Protein abundance varied

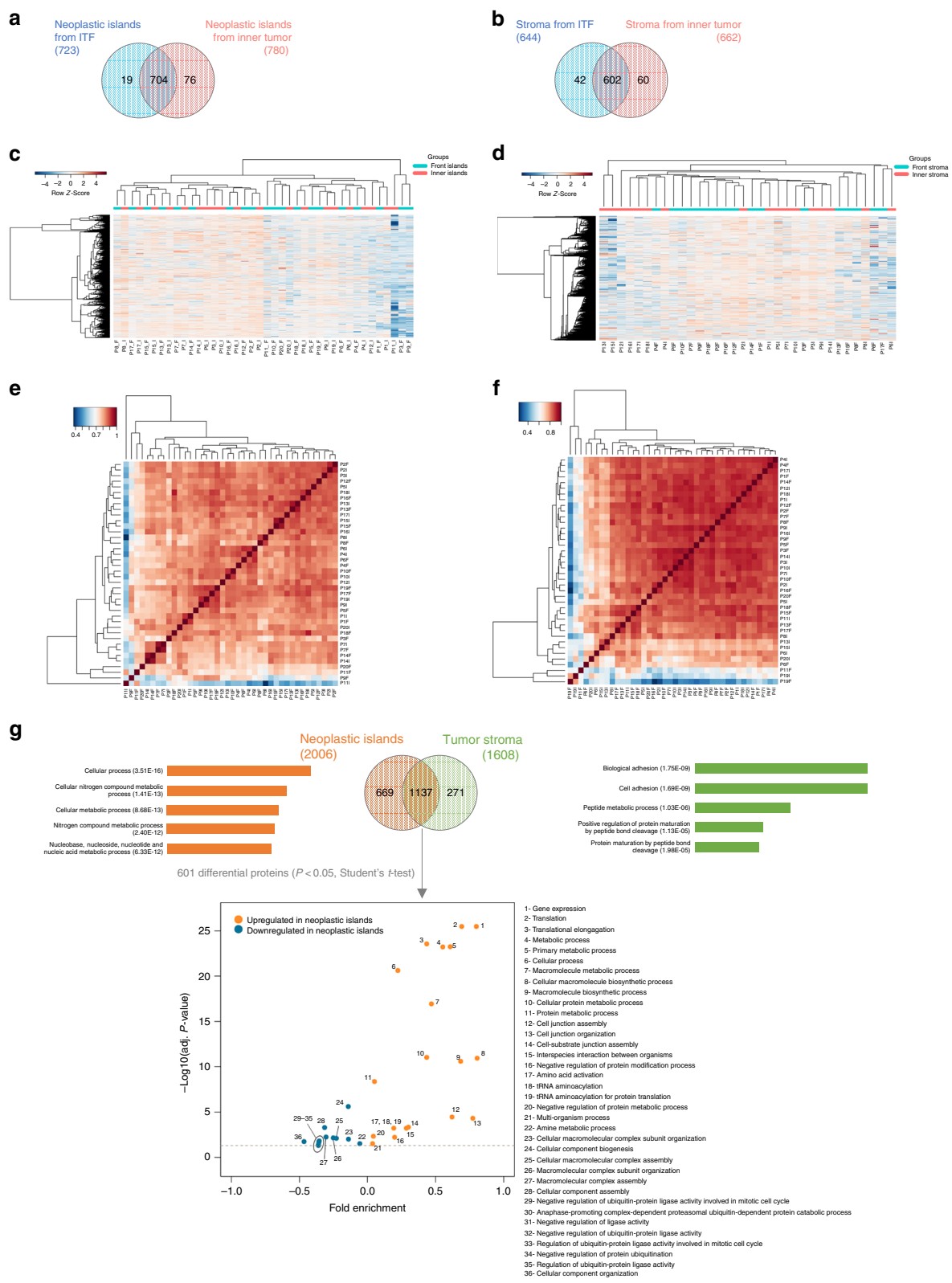

according to the score created on a scale of 0 to 6, which generated a staining scale for all cases (Fig. 4; Supplementary Fig. 6; Supplementary Data 20−22).

The IHC of the cases (approximately 14 cases for each selected protein) used in the discovery phase showed similar differential protein abundances and locations between the ITF and the inner tumor (Supplementary Data 21—Panel 1 and Table 1) for both the neoplastic islands and tumor stroma, despite the distinct dynamic range of the techniques. Further, the number of IHC cases was increased to 125 cases of neoplastic islands and 96 cases of tumor stroma, and most of the IHC results are similar to those for the original 14 IHC cases. However, with this significant increase in the size cohort, LTA4H, PGK1, and ITGAV staining were identified with a slight variation within the lower and higher scores in each region, either ITF or inner tumor (Supplementary

Data 21—Panels 1 and 2). In addition, in the IHC analysis of neoplastic island proteins (Fig. 4a, Supplementary Fig. 6a), LTA4H and PGK1 showed peripheral staining in neoplastic cells and were also detected in cells in the tumor stroma, such as inflammatory cells. Increased CSTB and NDRG1 expression was identified in the inner tumor, according to the MS discovery results, with staining only inside neoplastic cells of the OSCC, but in some cases, CSTB and NDRG1 were also detected in the adjacent normal epithelium. In turn, for tumor stroma proteins (Fig. 4b, Supplementary Fig. 6b), higher COL6A1 and MB expression was identified in the ITF according to the MS discovery analysis. Staining for COL6A1, ITGAV, and MB proteins indicated a preferential localization in the tumor stroma, with staining also identified in neoplastic cells. Positive staining in muscle was detected for MB. Taken together, nuclear and

**Table 1 Linear regression and correlation analyses of proteins with differential abundance in the invasive tumor front and inner tumor with clinicopathological variables of the cases used for LMD and LC-MS/MS**

| Tumor area | Protein | Gene name | Uniprot ID | Abundance (ITF/Inner tumor ratio) | Clinicopathological data | Linear regression (P value) | Correlation coefficient (R) | R-squared |
|---|---|---|---|---|---|---|---|---|
| Neoplastic islands | Actin-related protein 2 | ACTR2 | P61160 | −0.57 | Second primary tumor | 0.002 | −0.7 | 0.4 |
| Neoplastic islands | Cystatin-B | CSTB | P04080 | −1.55 | Treatment | 0.0009 | 0.7 | 0.5 |
| Neoplastic islands | Collagen alpha-2(I) chain | COL1A2 | P08123 | 1.07 | Treatment | 0.001 | 0.7 | 0.4 |
| Neoplastic islands | Leukotriene A-4 hydrolase | LTA4H | P09960 | −0.83 | Second primary tumor | 0.001 | −0.7 | 0.4 |
| Neoplastic islands | Phosphoglycerate kinase 1 | PGK1 | P00558 | −0.48 | Disease-free survival | 0.001 | — | 0.4 |
| Neoplastic islands | Phosphoglycerate kinase 1 | PGK1 | P00558 | −0.48 | Second primary tumor | 0.001 | −0.7 | 0.4 |
| Neoplastic islands | Protein NDRG1 | NDRG1 | Q92597 | −1.47 | Disease-free survival | 0.001 | — | 0.4 |
| Neoplastic islands | Protein S100-A8 | S100A8 | P05109 | 2.11 | Second primary tumor | 0.0003 | −0.7 | 0.5 |
| Neoplastic islands | Protein S100-A9 | S100A9 | P06702 | 2.35 | Second primary tumor | 0.0006 | −0.7 | 0.5 |
| Tumor stroma | Collagen alpha-1 (VI) chain | COL6A1 | P12109 | 0.534 | Lymph node status | 0.0006 | 0.7 | 0.5 |
| Tumor stroma | Fascin | FSCN1 | Q16658 | −0.673 | Poorly differentiated tumor (WHO) | 2.34e-05 | 0.8 | 0.7 |
| Tumor stroma | Integrin alpha-V | ITGAV | P06756 | −1.674 | Lymph node status | 0.0014 | 0.7 | 0.5 |
| Tumor stroma | Myoglobin | MB | P02144 | 2.901 | Clinical stage | 0.0015 | 0.7 | 0.5 |
| Tumor stroma | Thrombospondin-2 | THBS2 | P35442 | −0.712 | Lymph node recurrence | 0.0001 | 0.7 | 0.6 |

Abundance = Log2 of the LFQ intensity ratio (ITF/Inner tumor); Treatment: surgery, or surgery and radiotherapy, or combination of surgery, radiation and chemotherapy; WHO: Histopathological Grading System
*ITF* invasive tumor front

**Fig. 2** Quantitative proteome analysis indicates spatially distinct protein signatures. **a**, **b** Venn diagram of common and "exclusive" proteins identified for **a** neoplastic islands or **b** tumor stroma from the ITF and the inner tumor. **c**, **d** Clustering analysis of proteins identified in the ITF and the inner tumor of **c** neoplastic islands ($n = 20$ samples) and **d** tumor stroma ($n = 17$ samples). Values for each protein (rows) and for each microdissected sample (columns) are colored based on the protein abundance, in which high (red) and low (blue) values (Z-scored log2 LFQ intensity values) are indicated based in the color scale bar shown in the top left of the figure. The colored bars shown on the top of the figure indicate samples from the ITF (blue) or from inner tumor (pink). Hierarchical clustering was performed in the R environment using the Euclidean distance with complete ligation for neoplastic island data and the Euclidean distance with average ligation for stromal data. **e**, **f** Heat map of Pearson correlation coefficients derived from pairwise comparison of the 20-patient samples for **e** neoplastic island samples and for the **f** tumor stroma samples analyzed by discovery proteomics. Log2 LFQ intensity values of the protein dataset after filtering reverse and "only by site" entries were used to calculate the correlation coefficient using Perseus software, and the heat map was constructed using R language with the function heatmap.3. The dendrogram was built using Euclidean distance with complete ligation. Samples with low correlation values and low number of quantified proteins from tumor stroma dataset were removed, as shown in Supplementary Figure 2. **g** The five most enriched GO terms that distinguished neoplastic islands from tumor stroma are represented. GO terms for cellular metabolic processes are overrepresented for neoplastic island proteins, whereas cellular adhesion and protein cleavage processes are overrepresented for tumor stromal proteins. The statistically significant proteins between neoplastic islands compared with tumor stroma (two-sided Student's $t$ test, $P$ value $< 0.05$) also indicate, among other overrepresented processes, metabolic processes for proteins upregulated in neoplastic islands

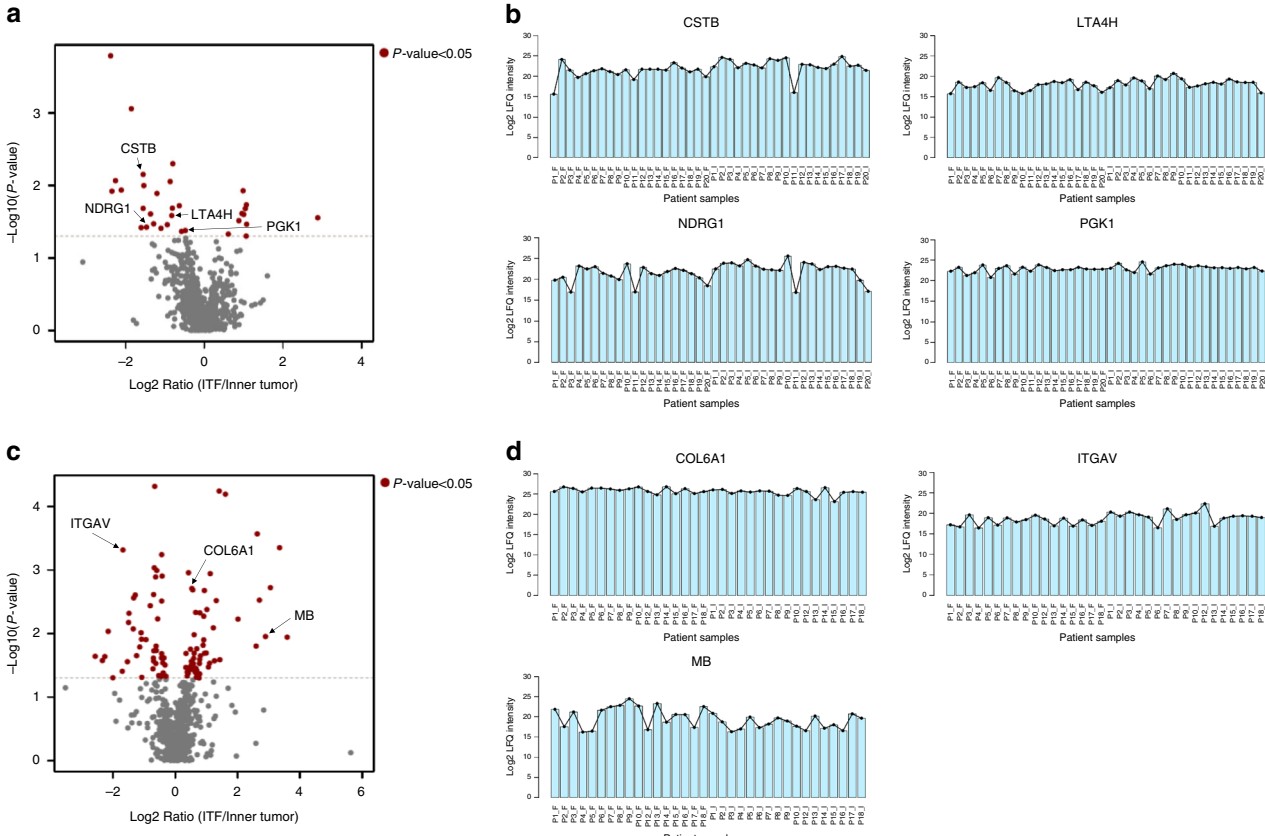

**Fig. 3** Abundance profile of proteins selected for verification steps. **a** Differentially regulated proteins between the ITF and inner tumor for neoplastic islands, as determined by plotting Student's *t* test (*P* value < 0.05, two-sided) *P* values versus the log2 ratio of the LFQ intensity (ITF/Inner tumor), are represented on volcano plots. Significant proteins are indicated by red dots. **b** Line plots depicting the abundance profiles (log2 of the LFQ intensity) of the proteins selected for the verification step. Shown are the abundances across all MS runs from discovery proteomics analysis of neoplastic island samples. Samples from the ITF are denoted by "F", and samples from inner tumor are denoted by "I". **c** Differentially regulated proteins between the ITF and inner tumor for tumor stroma (see panel **a** for plot details). **d** Protein abundance profiles for tumor stroma samples (see panel **b** for plot details)

cytoplasmic staining was identified for all proteins; however, the marked proteins were more pronounced or more specific to the cell type in which they were identified. For the specificities of the antibodies employed here, please see Supplementary Figure 7.

**Integrating IHC data with clinicopathological parameters**. In the 125- and 96-patient cohorts, IHC combined with clinicopathological characteristics strengthened the prognostic values for the selected proteins from the OSCC samples.

Cross-tabulation and the chi-square test indicated significant associations between the clinicopathological parameters and CSTB, PGK1, COL6A1, and ITGAV expression (Supplementary Data 23−24). Kaplan−Meier survival analysis in the OSCC patients indicated the association of lower abundance of CSTB and NDRG1 in the ITF with local relapse and second primary tumor, respectively, and higher abundance of PGK1 and ITGAV in the ITF with locoregional relapse and lymph node relapse, respectively (Fig. 5). CSTB, PGK1, and ITGAV were significant for the 5-year disease-specific survival and disease-free survival, while NDRG1 was only significant for the 5-year disease-free survival (Table 2).

CSTB was associated with local relapse in Cox multivariate analysis with 3-year disease-free survival rate of 82% for 44 patients, thus confirming this protein as an independent prognostic marker for OSCC patients (Supplementary Data 25). Also, this analysis showed that the lower abundance of CSTB in the ITF was associated to a higher risk of developing local

recurrence (HR 0.1224, 95% CI 0.0153−0.9801, *P* value = 0.0478, Cox multivariate analysis).

**SRM-MS of prioritized proteins in saliva of OSCC patients**. Saliva samples from 40 OSCC patients (Supplementary Data 26) were obtained to monitor peptides derived from CSTB, NDRG1, LTA4H, PGK1, COL6A1, ITGAV, and MB (Supplementary Data 27-28) based on the previously described criteria for protein selection. The samples were divided into two groups: patients without lymph node metastasis (N0) and patients with lymph node metastasis (N+).

Among the seven proteins investigated in saliva, six proteins, including CSTB, LTA4H, PGKI, NDRG1, COL6A1, and ITGAV, showed lower abundances in the saliva of the patients with lymph node metastasis (N+) than the patients without lymph node metastasis (N0) (Mann−Whitney *U* test, with *P* values adjusted for multiple comparisons using the Benjamini−Hochberg FDR method, adj. *P* value < 0.05, Fig. 6a, c).

At the peptide level, we also determined that the majority of peptides showed statistically significant lower abundances in the N+ patients than in the N0 patients (Mann−Whitney *U* test, with *P* values adjusted for multiple comparisons using the Benjamini−Hochberg FDR method, adj. *P* value < 0.05, Fig. 6b, d, Supplementary Data 29−31). The quality of SRM data and unsupervised hierarchical clustering analysis of peptides are illustrated in Supplementary Fig. 8−11.

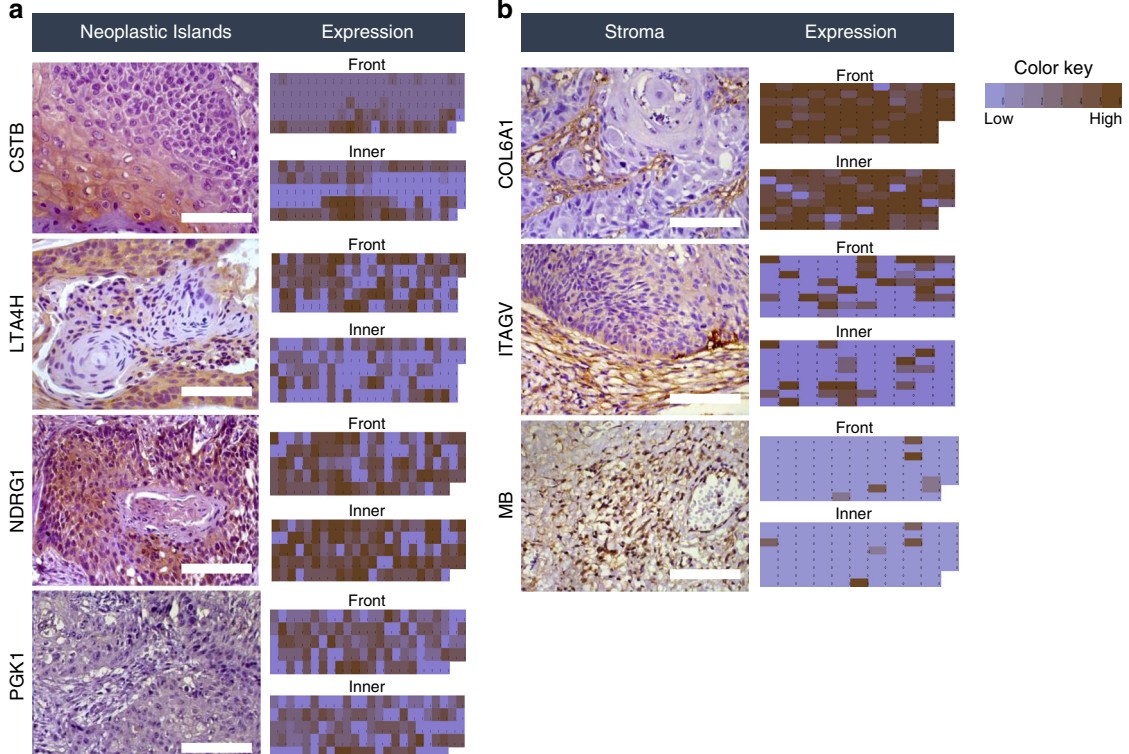

**Fig. 4** Immunohistochemical staining of targeted proteins. Oral SCC tissue samples from a set of 125 cases were used to verify the abundance of **a** CSTB, LTA4H, NDRG1, and PGK1 neoplastic island proteins, and 96 cases were used to verify the abundance of **b** COL6A1, ITGAV, and MB tumor stroma proteins. Among the cases, it was possible to analyze 114 cases for CSTB, 118 cases for LTA4H, 119 cases for NDRG1, 118 cases for PGK1, 93 cases for COL6A1, 80 cases for ITGAV, and 86 cases for MB. The scores that represent the sum of the intensities and percentage of protein staining in the ITF or the inner tumor are shown as a heat map. Differential staining in ITF and inner tumor in both neoplastic islands and tumor stroma was identified, in agreement with the MS discovery analysis however, we also identified negative cases for each protein and cases with gradual staining according to the color key shown at the top right. For proteins selected in neoplastic island, increased CSTB and NDRG1 expression was identified in the inner tumor, according to the MS results, with staining only inside neoplastic cells of OSCC. However, LTA4H and PGK1 presented peripheral staining in neoplastic cells and were also detected in cells from tumor stroma, such as inflammatory cells. For proteins selected in tumor stroma, greater COL6A1 and MB expression was identified in the ITF area, which demonstrates the preferential localization in OSCC regions compared with that observed in the MS discovery analysis. COL6A1, ITGAV, and MB proteins were preferentially present in tumor stroma (Histological images obtained using a ×40 objective, scale bars, 200 μm)

Moreover, we performed an analysis of the relationship between the tissue and saliva based on the two MS techniques employed, DDA and SRM (Fig. 6e). Most evaluated proteins exhibited lower abundance at the site of ITF and in saliva, which correlated with a poor prognosis. However, the overall results indicate that the abundance of proteins in saliva and its association with prognosis (N+ and N0) is not necessarily associated with the proximity of the altered oral epithelium (Fig. 6f).

**Saliva prognostic signatures distinguish regional metastasis.** The low expression of the proteins LTA4H, PGK1, NDRG1, COL6A1, and ITGAV in the saliva samples was associated with lymph node metastasis and advanced clinical staging (cross-tabulation and chi-square test, $P$ value < 0.05, Supplementary Data 32).

Further, through the strategies of machine learning described (Fig. 7a), we evaluated the predictive power of individual and groups of peptides and proteins to distinguish the patient with lymph node metastasis (N+) from the patient without lymph node metastasis (N0) (Fig. 7b–e, Table 3; Supplementary Data 33−37). The groups $S_1$: (Pep8_LTA4H, Pep12_CSTB), $S_2$: (Pep8_LTA4H, Pep9_COL6A1, Pep12_CSTB), $S_3$: (Pep8_LTA4H, Pep9_COL6A1), and $S_4$: (LTA4H) are the most relevant

signatures ($S_i$) considering accuracy and AUC (Fig. 7d; Table 3, Panel 1; Supplementary Data 38).

We determined that the signatures with the highest accuracies did not have the highest AUC values, which may be explained by the class imbalance between N0 and N+, further confirmed through the oversampling analysis. The AUC of the peptide level is considerably higher than that of the protein level, 82.8% ($S_2$) compared with 73.9% ($S_4$). Only the signature $S_4$ (LTA4H) was selected at the protein level, with an AUC of 73.9%, as other signatures have AUCs less than 62.5%. Further, balancing the training subsets also increased the overall prediction performance (Fig. 7b, e, Table 3, Panel 1).

Furthermore, the signatures $S_1$ and $S_2$ at the peptide level and S4 at the protein level are the best candidates for both types of cross-validation, using imbalanced and balanced classes. In addition, the $S_2$ is the best signature to discriminate N0 and N+ of OSCC.

## Discussion

The proteomic profiling of tumors is a promising approach for the discovery of diagnostic and prognostic methods, based on the identification of predictive markers of clinical aggressiveness and treatment outcomes, and the potential of therapeutic monitoring. However, the preparation of tissue samples for LC-MS/MS

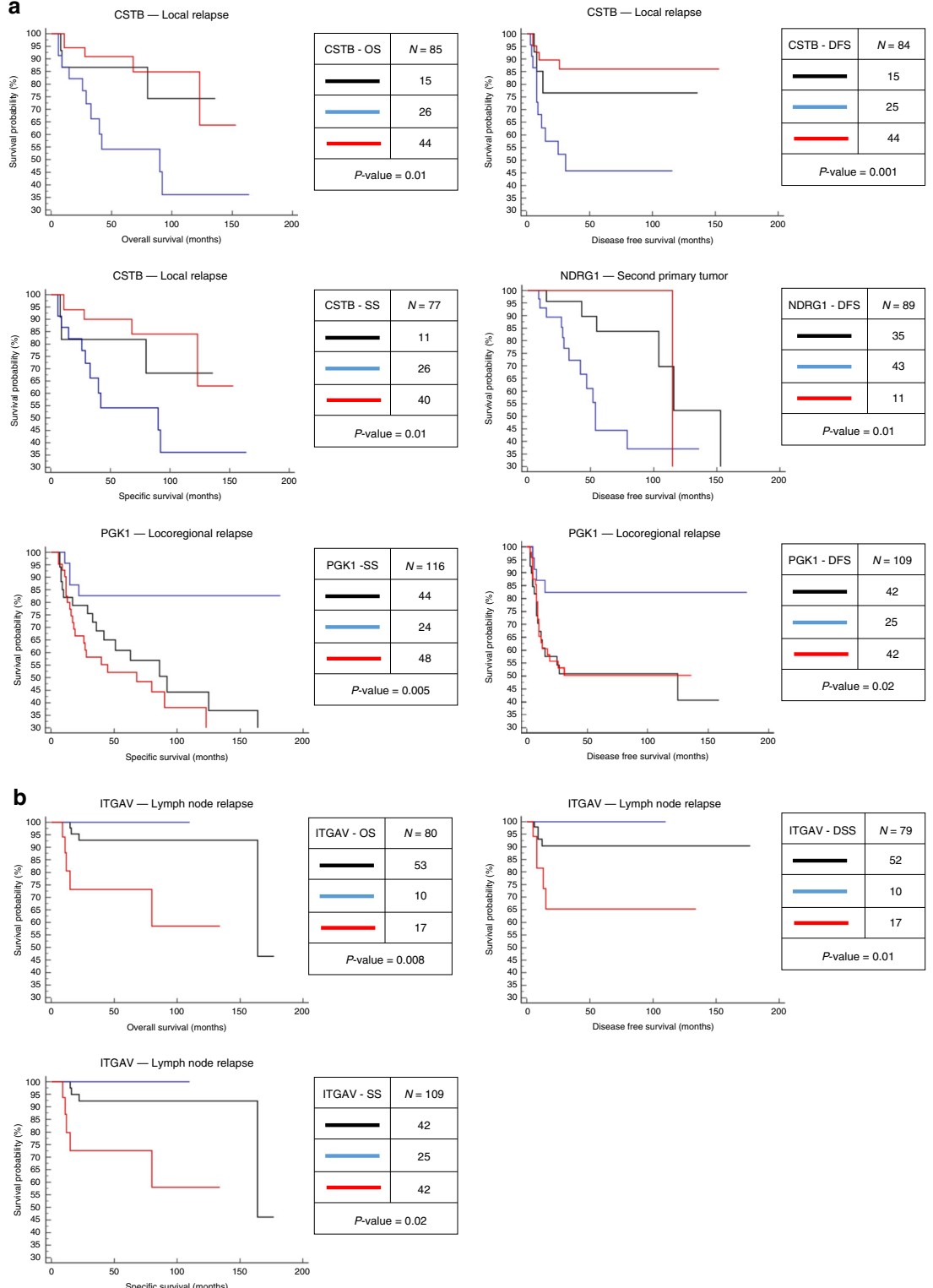

**Fig. 5** Kaplan−Meier survival analysis of IHC and clinical outcomes. Overall survival (OS), disease-free survival (DFS) and specific survival (SS) were available in relation to a second primary tumor, local, locoregional or lymph node relapse. **a** Patients with lower CSTB expression in the ITF had a higher risk of local relapse and worse survival (*P* value < 0.05, log-rank test). In addition, a lower NDRG1 expression in the ITF was associated with a higher risk of the patient presenting a second primary tumor and a worse DFS (*P* value < 0.05, log-rank test). Equal expression between the ITF and the inner tumor or higher CSTB and NDRG1 expression in the ITF did not influence the local relapse or second primary tumor. Patients with higher PGK1 expression in the ITF had a worse survival and early locoregional recurrence (*P* value < 0.05, log-rank test). **b** Patients with higher expression of ITGAV in the ITF have a greater risk to present lymph node metastasis relapse and poor survival (*P* value < 0.05, log-rank test)

**Table 2 Kaplan—Meier survival analysis of OSCC patients according to the levels of protein expression obtained by IHC analysis**

| Parameter | Disease-free survival in 5 years % (n) | HR; CI (P value) | Specific survival in 5 years % (n) | HR; CI (P value) |
|---|---|---|---|---|
| CSTB | | | | |
| Lower expression in ITF | 45% (25) | | 54% (26) | |
| Equal expression | 76% (15) | | 81% (11) | |
| Higher expression in ITF | 86% (44) | 0.21; 0.07−0.60 (0.001) | 90% (40) | 0.28; 0.10−0.79 (0.01) |
| NDRG1 | | | | |
| Lower expression in ITF | 44% (43) | | — | — |
| Equal expression | 83% (35) | | — | — |
| Higher expression in ITF | 100% (11) | 0.31; 0.12−0.79 (0.01) | — | — |
| PGK1 | | | | |
| Lower expression in ITF | 82% (24) | | 82% (25) | |
| Equal expression | 50% (44) | | 61% (42) | |
| Higher expression in ITF | 50% (48) | 3.22; 1.45−7.15 (0.02) | 52% (42) | 3.96; 1.82−8.64 (0.005) |
| ITGAV | | | | |
| Lower expression in ITF | 100% (10) | | 100% (08) | |
| Equal expression | 90% (52) | | 92% (50) | |
| Higher expression in ITF | 65% (17) | 4.02; 0.77−20.76 (0.02) | 72% (16) | 4.99; 0.90−27.68 (0.01) |

*HR* hazard ratio, *CI* confidence interval, *ITF* invasive tumor front

analysis can be challenging as a result of their morphological and molecular complexity[7,12,28].

In this context, a comprehensive understanding of the proteome of the ITF of OSCC may enhance prognostic accuracy and guide appropriate treatment, thus improving the survival of patients with OSCC. The interest in ITF composition has augmented because recent studies using histopathological grading systems have incorporated the analysis of the histopathology of invasion patterns[4,9,29] and IHC analyses of OSCC tissues for clinical decisions[14,15,30]. Besides, the differential protein composition of the neoplastic islands and the stroma has also been identified as prognostic factors considering its association with tumor progression[12,13]. The tumor stroma has a distinct histological origin from the neoplastic cells of OSCC, which is primarily composed of fibroblasts, endothelial cells, and inflammatory cells[31,32]. Altogether, the tumor microenvironment is in fact increasingly complex and mediates the signaling crosstalk of different tumor cell and stromal cell types and tumor regions[12–14,32]. Previous proteomic studies of oral cancer have used LDM to evaluate tumor cells of epithelial origin[29,33–35] and Jensen et al.[17] analyzed the ITF and the central area of the OSCC tumor; however, they did not evaluate the stroma from each region.

With our study, we demonstrated using discovery and targeted/verification phases that not only the canonical approach using morphological tumor heterogeneity, the classic clinicopathological characteristics of patients, including TNM, and the immunolocalization of proteins, but also the knowledge about the proteome map should be implemented in the clinical routine to identify proteins of prognostic value for OSCC.

Here, for the discovery phase, we overcame the histopathological complexity of the tumor tissue samples using histopathological-guided LC-MS/MS to separate small and large neoplastic islands and their surrounding tumor stroma by LMD, followed by mapping the proteome of these distinct tumor areas, which contributes to a better understanding of the protein composition between different areas of the tumor. We identified 32 proteins with significantly different abundances between neoplastic islands from the ITF and the inner tumor and 101 proteins with distinct abundances in the stroma from both tumor regions (Figs. 2, 3 and Supplementary Data 9 and 13). With this approach, distinct overrepresented biological processes were enriched in the neoplastic island proteome, such as cellular metabolic processes, whereas the proteome of the tumor stroma was overrepresented by cellular adhesion and other processes associated with protein and peptide cleavage (Fig. 2g). The enriched terms of neoplastic islands are associated with a hallmark of cancer, as the ability to reprogram the metabolism to meet the bioenergetic, biosynthetic, and redox demands of malignant cells[36,37]. Interestingly, the loss of cell adhesion favoring invasion and metastasis is one of the most characterized alterations in carcinoma[36,37].

Considering the significant proteins from different areas of tumor have a potential prognostic value, we subsequently prioritized them for the next targeted phase, IHC and SRM-MS. Among them, 13 proteins correlated with clinicopathological parameters and of these proteins, eight proteins were identified in neoplastic islands (ACTR2, CSTB, COL1A2, LTA4H, PGK1, NDRG1, S100A8, S100A9) and five proteins were identified in tumor stroma (COL6A1, FSCN1, ITGAV, MB, THBS2) (Table 1). We followed-up with verification steps with the seven potential candidates COL6A1, CSTB, NDRG1, LTA4H, ITGAV, PGK1, and MB, according to the additional criteria for protein prioritization (Supplementary Data 18).

Although elucidating the role of these selected proteins was beyond the aim of this study, it is interesting that the spatial organization of the proteins may reflect different phenotypes when observed inside neoplastic cells, whereas in the periphery of neoplastic cells, such as in the ITF, they may be associated with a highly aggressive phenotype, including invasion and cellular migration mechanisms[9,12,17,38]. All selected proteins have previously been reported in some type of cancer, with limited information regarding their role associated with their localization in the tumor microenvironment. It is interesting that it has previously been reported that low levels of CSTB result in an increase in extracellular matrix (ECM) degradation, migration, and cell invasion[39,40]. Moreover, a recent study indicated that its downregulation may promote the development of gastric cancer by affecting cell proliferation and migration, as well as activation of the PI3/Akt/mTOR signaling pathway[39]. Higher levels of human NDRG1 may act as a tumor suppressor that is involved in cellular differentiation, cell cycle regulation, responses to hormones, nickel and stress, cell adhesion and ECM degradation[41]. NDRG1 has been reported to correlate with metastasis in prostate

cancer, pancreatic cancer, and colorectal cancer[42]. Another protein from the neoplastic islands, PGK1, is a glycolytic enzyme that is involved in the tumor biology, angiogenesis, replication and repair of DNA and metastasis, and its increase has been investigated in breast cancer, pancreatic cancer, gastric cancer, and liver cancer[43]. The protein LTA4H belongs to the family of zinc metalloproteases and has a role in the response to inflammation[44]. Another recent study has indicated that LTA4H regulates

the cell cycle and the knockout of the protein reduced skin cancer development in mouse model[45].

Among the proteins from tumor stroma, the protein ITGAV is a receptor of ECM and serves as a subunit for receptors of integrins, RGD-motif bound containing substrates, such as vitronectin, fibronectin and fibrinogen, which are components of the stroma involved in the EMT transition[46]. Furthermore, high levels of ITGAV have previously been associated with tumor

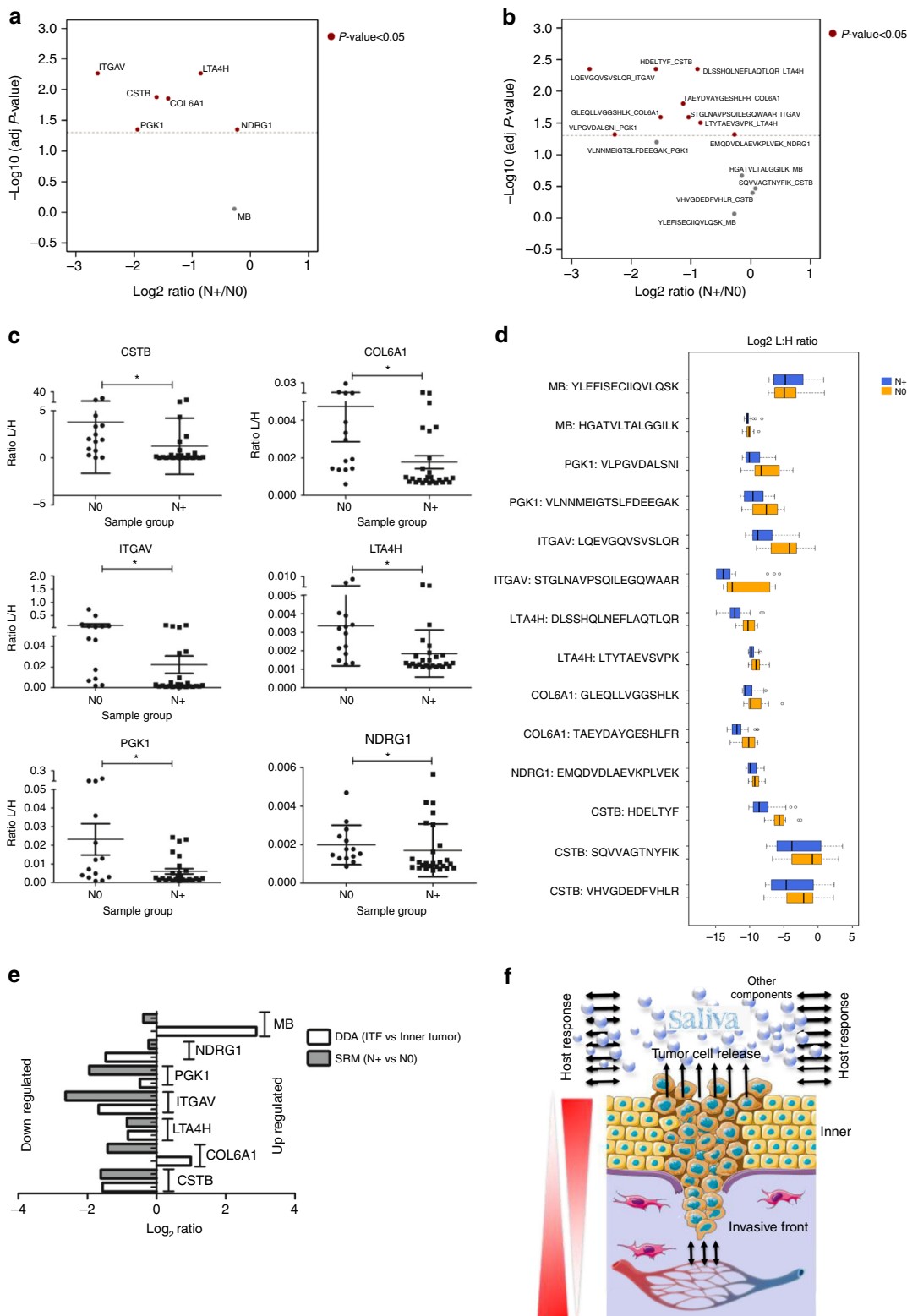

growth regulation and metastasis in tumors of the hypopharynx and larynx[47] and tumor progression in colorectal cancer[48], but not previously in oral cancer. COL6A1 protein is composed of three distinct chains (α1, α2, and α3) secreted in the ECM to form a network of microfilaments, which is associated with the basal membrane of muscular cells and interacts with several components of the ECM[49]. Collagen VI is an important protumorigenic factor that acts in the tumor microenvironment, promoting events of inflammation and angiogenesis[50].

For the targeted phase, several technologies could be used, such as targeted proteomics, IHC, ELISA, and western blotting[17,51]. IHC was the first method of choice because of its broad applications in clinical practice and ability to provide results for patient postoperative management decisions[28]. In addition, histopathology remains the gold standard to determine the post-surgical treatment prognosis of OSCC patients, based on the evaluation of morphological findings, such as invasion by neoplastic islands and perineural invasion that lead to a worse clinical outcome of the disease[4,10].

The seven selected proteins were evaluated by IHC in a larger patient cohort, with 125 and 96 patients for the evaluation of proteins identified in neoplastic islands and tumor stroma, respectively. The clinical characteristics of both patient cohorts used in the discovery and verification phases are considered to be typical of the profiles of patients affected by oral cancer (Supplementary Data 19)[1,29]. The IHC analysis of the neoplastic islands and tumor stroma (Fig. 4) confirmed the proteins' preferential location and abundance identified by MS discovery analysis (Table 1, Supplementary Data 21) and further strengthened the correlation with the patients' outcomes (Supplementary Data 23−24, Fig. 5).

The results indicated that prioritized proteins, CSTB, NDRG1, PGK1, and ITGAV, may play a relevant biological role in tumor progression, since they were potentially associated, directly or indirectly, with patient prognosis (Table 2, Fig. 5). Interestingly, the clinical data indicated a high-power correlation of the low expression of CSTB in the ITF with local recurrence, resulted from aggressive or advanced tumors, as an independent prognostic marker (Supplementary Data 25), using the criteria for the consideration of a marker for cancer prognosis based on the REMARK guidelines[51]. In fact, the lower expression of CSTB in the ITF also observed in discovery phase correlated with patient treatments combining surgery, radiotherapy and chemotherapy, which is usually the therapeutic modality prescribed to patients with advanced/aggressive tumors (Table 1).

A literature review of oral cancer indicated that recurrence episode occurs within 7.5 months after initial treatment[5], whereas our study showed a mean recurrence period of 12 months for

patients with lower CSTB expression in the ITF. It is known that local or regional recurrence of oral cancer represents a clinical challenge for the choice of therapeutic methods and clinical follow-up, considering that failures in postoperative treatment and radiotherapy have been indicated to explain up to 90% of recurrences[52]. Low CSTB levels have previously been correlated with a shorter disease-free survival rate in breast cancer[53]. In contrast, the study by Feldman et al.[54] identified high levels of CSTB as an independent marker for bladder cancer recurrence. Similarly, low levels of NDRG1 in the ITF area revealed to be associated to the presence of secondary primary tumor and low disease-free survival, whereas patients with locoregional recurrence and with lymph node relapse presented higher abundance of PGK1 and ITGAV in the ITF, respectively, associated to disease-free survival (Fig. 5).

To further investigate the prognostic value of the selected proteins in OSCC patients in another targeted phase via SRM-MS analysis, we used saliva, a noninvasive fluid, as a promising source for candidate markers. The lymph node status was the prioritized clinical parameter employed because lymph node metastasis is the most important factor for prognosis in oral cancer[55,56] and its presence is highly associated with a poor prognosis[3]. Moreover, compromised lymph nodes showed significant clinical correlations for some selected proteins, besides associations with clinical stage and disease-free survival (Table 1, Fig. 5 and Supplementary Data 23−24), which are clinical data established to correlate with lymph node metastasis in oral cancer[57−59].

Interestingly, SRM-MS analysis showed CSTB, LTA4H, PGK1, COL6A1, ITGAV, and NDRG1 were significantly downregulated in patients with lymph node metastasis (Fig. 6a, b). Moreover, they were significantly associated with lymph node metastasis, which, in turn, correlated with advanced clinical staging, with exception of CSTB (Supplementary Data 32). Notably, the expression levels of ITGAV and COL6A1 in the tissues were also previously associated with lymph node status (Table 1 and Fig. 5). Future studies could evaluate the CSTB, PGK1, LTA4H levels in saliva according to recurrence and/or second primary tumor, which were relevant clinical data associated with their expression in the tissues (Table 1, Fig. 5) and are also clinical challenges for therapeutic management in OSCC.

Furthermore, to test the strength of the association of these proteins with lymph node status, through multivariate analysis, we evaluated the individual and combined performances of the selected proteins and their peptides, with the aim to identify the best signatures with prognostic value. We performed a robust pipeline (Fig. 7a) for which the groups, N+ and N0, were used to determine the discrimination power of each signature through

**Fig. 6** Targeted proteomics of saliva proteins. **a**, **b** Volcano analyses show log2 ratio of N+/N0 of **a** proteins and **b** peptides according to the adjusted P value. Proteins that met the indicated statistical cut-off criteria (Mann−Whitney U test, with P values adjusted for multiple comparisons using the Benjamini−Hochberg FDR method, adjusted P value < 0.05) are colored in red. **c** The graph demonstrates individually the L/H intensity ratio (not log transformed) of six differentially expressed proteins CSTB, COL6A1, ITGAV, LTA4H, PGK1, and NDRG1 between N+ and N0 saliva samples. *P value < 0.05, Mann−Whitney U test. **d** Peptide relative quantification (log2 L/H ratio) between N+ and N0 saliva samples. For each protein, 2−3 proteotypic peptides were monitored, with exception for NDRG1, only one proteotypic peptide was monitored. The light peptide (corresponding to the endogenous peptide present in saliva) and the heavy peptide (which corresponds to the synthetic peptide spiked-in saliva) were monitored, and the light/heavy ratio for each of the 14 peptides was obtained by Skyline. Box plots represent the median and interquartile range, whiskers represent the 1–99 percentile, and outliers are represented by empty circles. **e** Bar plots represent the relationship between MS Discovery analysis of tissue and SRM-MS of saliva in the identification of potential prognostic signatures. The log2 N+/N0 ratio for saliva samples and log2 ITF/inner ratio for neoplastic islands and tumor stroma from microdissected tissues are represented in the graph. **f** Representative figure illustrates the gradient dynamics of the protein abundance between tissue (ITF and inner tumor) and saliva (N+ and N0), which indicates that the abundance of proteins in saliva and its association with prognosis (N+ and N0) is not necessarily associated with the proximity of the altered oral epithelium. Other components, such as water, electrolytes, DNA, RNA, and microorganisms, were not included. Images in **f** were adapted from files provided by Servier Medical Art (https://smart.servier.com/, licensed under a Creative Commons Attribution 3.0 Unported License)

cross-validation, testing all possible combinations of proteins and peptides. The prognostic value of protein LTA4H was strengthened by the cross-validation performance, being the most accurate signature at the protein level. At the peptide level, the analysis indicated three good candidates, $S_1$ and $S_2$ with the highest accuracy in both balanced and imbalanced cross-validations. Comparing the overall performance among all signatures, we determined that peptide groups are more accurate than protein groups and that $S_2$ is the best signature to

discriminate the stages N0 and N+ of OSCC (Fig.7e, Table 3, Panel 2).

In summary, the discovery phase enabled us to spatially map the proteome of neoplastic islands and their surrounding stroma of OSCC, identifying proteins with potential prognostic value. The targeted phase of IHC and SRM-MS in independent cohorts verified prognostic signature markers that may have applications in routine clinical practice of tissue histopathology and in very promising noninvasive biofluid saliva, driving prognostic

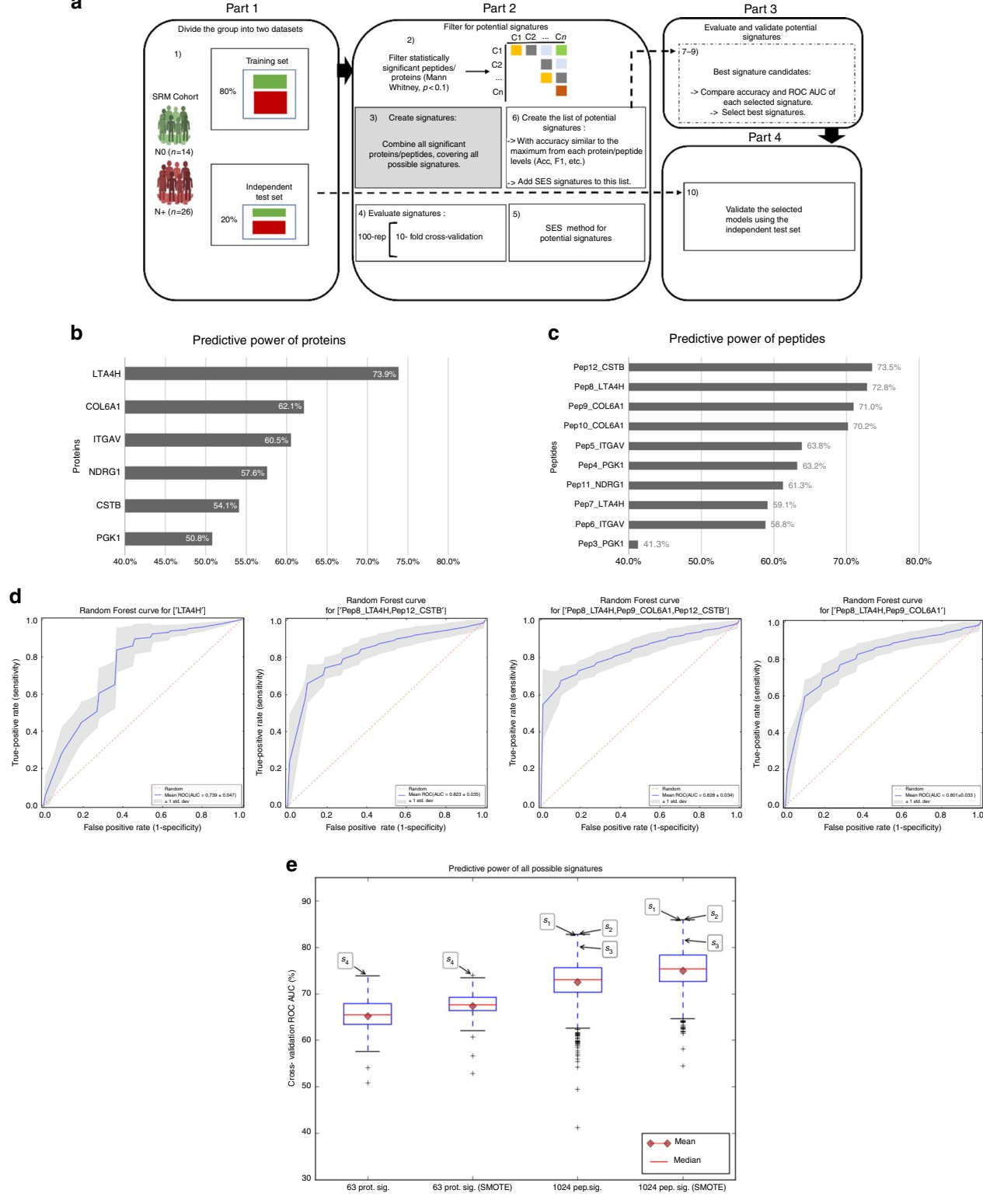

decisions that can contribute to precise treatment protocols and reduction of tumor local relapse or lymph node metastasis. Extensive longitudinal study with large-sized independent patient cohorts is still necessary before clinical implementation. Here we indicate a robust prognostic signature with CSTB, at low protein expression levels in the ITF, as an independent marker for local recurrence; and with the combination of LTA4H-, COL6A1-, and CSTB-specific peptides in saliva, able to distinguish patients with and without lymph node metastasis.

## Methods

**Clinical tissue sample collection**. This retrospective cohort study was approved by the Research Ethics Committee of the Faculty of Medical Sciences at the University of Campinas (Campinas-SP, Brazil) through Plataforma Brasil protocol CAAE 23163113.5.1001.5404 and the Research Ethics Committee of the Piracicaba Dental School through Plataforma Brasil protocol CAAE: 71351517.7.0000.5418. The methods and experimental protocols of the present study were performed in accordance with the approved guidelines and informed consent was obtained from all human participants. Two sets of tissue samples were used: the 20-patient cohort 1 (Supplementary Data 1), with samples for mass spectrometry analysis, and the 125- and 96-patient cohorts (Supplementary Data 19, including approximately 14 cases used in cohort 1 for the discovery MS), with surgical samples for IHC staining. The first set comprised 20 cases of primary tongue squamous cell carcinoma, retrieved from two reference hospitals in Cascavel, Paraná, Brazil, the Oncology Center of Cascavel, CEONC and the UOPECCAN Cancer Hospital, collected over a 20-year period (from 1998 to 2008). The inclusion criteria were as follows: (a) demographic and complete clinicopathological data; (b) location of the tumor in the tongue; (c) type of treatment based on radical surgery with or without postoperative radiotherapy and/or chemotherapy; and (d) availability of surgical specimens in FFPE blocks. The presence of small and large neoplastic islands in the ITF and the inner tumor from hematoxylin and eosin (H&E)-stained histological sections was evaluated and used as inclusion criteria. As proposed by Bryne et al.[16] and Brandwein-Gensler et al.[11], neoplastic islands with more than 15 cells were classified as large islands; neoplastic islands with 15 cells or less were classified as small islands. The ITF region was considered the area that contained the neoplastic islands invading the normal tissues, such as connective tissue, muscle, salivary glands and blood vessels, and the inner part of the tumor was the region close to the neoplastic epithelium (when present), i.e., the most distant region from the invasion. The second set of samples, employed for the IHC assays, was retrieved from reference oncology centers in Brazil and Finland: (1) 96 OSCC surgical specimens from patients diagnosed and treated at the Oncology Center of Cascavel CEONC and the UOPECCAN Cancer Hospital in Cascavel, Paraná, Brazil; and (2) 29 OSCC surgical specimens from patients diagnosed at the initial stage (I and II) of the disease at the University Hospital of Oulu in Finland. The use of OSCC samples from Finland was approved by National Supervisory Authority for Welfare and Health (Valvira, Dnro 7449/06.01.03.01/2013). From this second set of samples, 125 cases were used for the verification of proteins identified in neoplastic islands, and 96 cases were used for the verification of proteins in the tumor stroma.

The information collected from the medical records and follow-up of patients included the following: sex; age; habits, such as smoking and alcohol consumption; tumor location; TNM stage; status of surgical margins; local recurrence and lymph node metastasis; distant metastasis; the presence of second primary tumors; treatment; and survival. After treatment, the patients were monitored for at least 5 years, and disease recurrence was histologically confirmed. Outcomes were classified as specific survival (SS) for the disease, the time from the beginning of treatment until death as a result of OSCC or the last follow-up information when the patient was alive, and DFS, the time from the beginning of treatment until diagnosis of the first recurrence (local, regional or distant) or last follow-up information for patients without recurrence. Stained H&E histological slides were evaluated according to four histopathologic classification systems, the WHO classification system[60], the Malignancy Grading (MG) system[61], the Histologic Risk (HR) model[11] and the Budding and Depth of invasion (BD) model[4], and the slides were calibrated by two pathologists[10].

**Sample preparation and LMD**. Four histological slides were prepared for each of the 20 OSCC samples. Paraffin blocks were cut using a microtome with a thickness of 5 μm; the other three slides had a thickness of 10 μm. Histological sections with a thickness of 5 μm were stained with H&E to guide the LMD. The other three sections of 10 μm thickness were prepared using specific membrane slides (PEN Arcturus® Membrane, Life Technologies, Foster City, CA, USA) for LMD, which were deparaffinized in xylol, hydrated with decreasing concentrations (100, 90, 70,

---

**Table 3 Comparison among the best candidates from cross-validation without (Panel 1) and with oversampling (Panel 2)**

| Si | Accuracy (%) | AUC (%) | F1 (%) | Sensitivity (%) | Specificity (%) | Precision (%) | Signature | | |
|---|---|---|---|---|---|---|---|---|---|
| Panel 1: Cross-validation | | | | | | | | | |
| S1 | 75.9 | 82.3 | 80.5 | 75.6 | 76.6 | 86.2 | Pep8 | Pep12 | |
| S2 | 74.6 | 82.8 | 80.1 | 77.9 | 68.4 | 82.7 | Pep8 | Pep9 | Pep12 |
| S3 | 75.4 | 80.1 | 80.8 | 79.4 | 67.6 | 82.5 | Pep8 | Pep9 | |
| S4 | 76.2 | 73.9 | 82.0 | 83.4 | 62.3 | 80.8 | LTA4H | | |
| Panel 2: Cross-validation (oversampling by SMOTE) | | | | | | | | | |
| S2 | 77.2 | 85.9 | 81.6 | 76.9 | 77.8 | 87.1 | Pep8 | Pep9 | Pep12 |
| S1 | 76.9 | 85.5 | 80.9 | 74.5 | 81.5 | 88.5 | Pep8 | Pep12 | |
| S3 | 75.0 | 83.0 | 80.2 | 77.1 | 70.9 | 83.7 | Pep8 | Pep9 | |
| S4 | 69.3 | 74.0 | 74.18 | 68.05 | 71.55 | 81.95 | LTA4H | | |

Si, Signature; F1, F score; SMOTE, Synthetic Minority Over-sampling Technique. Pep8_ LTA4H: DLSSHQLNEFLAQTLQR; Pep9_COL6A1: GLEQLLVGGSHLK; Pep12_CSTB: HDELTYF

---

**Fig. 7** Prognostic signature in saliva distinguishes OSCC patients. **a** Workflow for machine-learning approach to measure the predictive power of peptides and proteins. **b**, **c** The predictive relevance of individual proteins and peptides to distinguish N0 from N+ patients is represented by a bar chart indicating their cross-validation ROC AUC (100 repetitions of stratified tenfold cross-validation). The most relevant protein and peptide ordered by the AUC is LTA4H and Pep8_LTA4H, respectively. When only the AUCs of the individual signatures (size 1) are considered, the three highest areas at the protein level are LTA4H (73.9%), COL6A1 (62.1%), and ITGAV (60.5%) and at the peptide level are Pep12_CSTB (73.5%), Pep8_LTA4H (72.8%), and Pep9_COL6A1 (71.0%). **d** Cross-validation estimated ROC curves of the best protein and peptide signatures. **e** Box plots representing the AUC of all possibilities of signatures for both imbalanced and balanced (SMOTE) cross-validation. At the peptide level, 1024 signatures were tested. At the protein level, 63 signatures were tested. Signatures formed by peptides from different proteins S1 {Pep8, Pep12} and S2 {Pep8, Pep9, Pep12} have approximately 10.5% higher AUC than the peptide signature formed by LTA4H (S4). S2 peptide signature outperformed both S1 and S4 signatures. The candidate signatures are indicated by labels: S1, S2, S3, and S4. Peptide sequences: Pep1_MB: HGATVLTALGGILK; Pep2_MB: YLEFISECIIQVLQSK; Pep3_PGK1: VLNNMEIGTSLFDEEGAK; Pep4_PGK1: VLPGVDALSNI; Pep5_ITGAV: LQEVGQVSVSLQR; Pep6_ITGAV: STGLNAVPSQILEGQWAAR; Pep7_LTA4H: LTYTAEVSVPK; Pep8_LTA4H: DLSSHQLNEFLAQTLQR; Pep9_COL6A1: GLEQLLVGGSHLK; Pep10_COL6A1: TAEYDVAYGESHLFR; Pep11_NDRG1: EMQDVDLAEVKPLVEK; Pep12_CSTB: HDELTYF; Pep13_CSTB: SQVVAGTNYFIK; and Pep14_CSTB: VHVGDEDFVHLR. Four peptides were not included in the training model because they did not pass the filtering step (step 2 from Part 2 of Fig. 7a; P value < 0.1, Mann−Whitney U test). Box plots represent the median and interquartile range, whiskers represent the 1–99 percentile, and outliers are represented by "+"

and 50%) of ethanol, washed in water and stained with hematoxylin blue for 8 min prior to drying for LMD[62].

Samples were processed using Leica Laser Microdissection Systems. The microdissected areas were as follows: (1) small neoplastic islands from the ITF; (2) large neoplastic islands from the ITF; (3) small neoplastic islands from the inner tumor; (4) large neoplastic islands from the inner tumor; (5) stroma from the ITF; and (6) stroma from the inner tumor (Fig. 1b). Neoplastic islands from the ITF were collected from the farthest island in the invasive surface of the tumor, up to a depth of one millimeter in the histological section, and neoplastic cell islands from the inner tumor were cut from the farthest island inside the tumor, up to a depth of one millimeter (Fig. 1a). Average microdissected tissue areas of 100,000 $\mu m^2$ and 1,000,000 $\mu m^2$ were isolated for small neoplastic islands and large neoplastic islands, respectively, and an average area of 1,000,000 $\mu m^2$ for stroma (Supplementary Fig. 1).

All samples were collected in 600 μL microtubes and stored at −80 °C. LMD was standardized for FFPE tissues with a thickness of 10 μm. The adjustable parameters in the LMD Laser Microdissection Leica software are considered optimum for these samples, and the area ($\mu m^2$) cut for each patient was recorded.

**Protein extraction and trypsin digestion.** For protein extraction and digestion, samples were treated with 8 M urea, followed by protein reduction with dithiothreitol (5 mM for 25 min at 56 °C) and alkylation with iodoacetamide (14 mM for 30 min at room temperature in the dark). For protein digestion, urea was diluted to a final concentration of 1.6 M with 50 mM ammonium bicarbonate, and 1 mM of calcium chloride was added to the samples for trypsin digestion for 16 h at 37 °C (2 μg of trypsin)[63]. The reaction was quenched with 0.4% formic acid, and peptides were desalted with C18 stage tips[64], dried in a vacuum concentrator, reconstituted in 0.1% formic acid and stored at −20 °C for subsequent analysis by LC-MS/MS.

**Mass spectrometry analysis using DDA.** The peptide mixture (4.5 μL) was analyzed using an LTQ Orbitrap Velos (Thermo Fisher Scientific) mass spectrometer coupled to nanoflow liquid chromatography on an EASY-nLC system (Proxeon Biosystems) with a Proxeon nanoelectrospray ion source. Peptides were subsequently separated in a 2–90% acetonitrile gradient in 0.1% formic acid using a PicoFrit analytical column (20 cm × ID75, 5 μm particle size, New Objective) at a flow rate of 300 nL/min over 212 min, in which a gradient of 35% acetonitrile is reached in 175 min. The nanoelectrospray voltage was set to 2.2 kV, and the source temperature was set to 275 °C. The instrument methods employed for LTQ Orbitrap Velos were set up in DDA mode. Full scan MS spectra (m/z 300–1600) were acquired in the Orbitrap analyzer after accumulation to a target value of 1e[6]. Resolution in the Orbitrap was set to $r = 60,000$, and the 20 most intense peptide ions (top 20) with charge states ≥2 were sequentially isolated to a target value of 5000 and fragmented in the high-pressure linear ion trap by CID (collision-induced dissociation) with a normalized collision energy of 35%. Dynamic exclusion was enabled with an exclusion size list of 500 peptides, an exclusion duration of 60 s and a repetition count of 1. An activation Q of 0.25 and an activation time of 10 ms were used. The run order of the samples is described in Supplementary Data 2.

**Proteomic data analysis.** One hundred and twenty LC-MS/MS runs were performed, in which 20 LC-MS/MS runs were conducted for each region (Fig. 1e; Supplementary Data 2). Raw data were processed using MaxQuant v1.3.0.3 software[65], and MS/MS spectra were searched against The Human UniProt database (released January 7, 2015, 89,649 sequences, and 35,609,686 residues) using the Andromeda search engine[66]. As search parameters, a tolerance of 6 ppm was considered for precursor ions (MS search) and 0.5 Da for fragment ions (MS/MS search), with a maximum of two missed cleavages. Carbamidomethylation of cysteine was considered a fixed modification, and oxidation of methionine and protein N-terminal acetylation were considered variable modifications. A maximum of a 1% false discovery rate (FDR) was set for both the protein and peptide identification. Protein quantification was performed using the LFQ algorithm implemented in MaxQuant software, with a minimal ratio count of 2 and a window of 2 min for matching between runs. Statistical analysis was performed with Perseus v1.2.7.4 software[65], which is available in the MaxQuant package. Both the raw files of large and small neoplastic islands from the ITF and the large and small neoplastic islands from the inner tumor were combined in the experimental design of MaxQuant. Identified protein entries were processed, excluding reverse sequences and those identified "only by site" entries. Contaminants were not removed from the dataset because keratin proteins are of interest in the study of squamous tissues. Tumor stroma samples from three patients exhibited low correlation values and low number of quantified proteins and were thus removed from the analysis (Supplementary Fig. 2).

Protein abundance, which was calculated based on the normalized spectrum intensity (LFQ intensity), was log2-transformed, and the dataset was filtered by minimum valid values in at least one group (ten valid values for neoplastic island samples and eight valid values for tumor stroma samples). Missing values for the LFQ intensity were imputed with random numbers from a normal distribution, the mean and standard deviation of which were selected to best simulate low abundance values close to the noise level (imputation width = 0.3, shift = 1.8)[67,68].

Significance was assessed using Student's $t$ test to identify differentially expressed proteins between the ITF and inner areas ($P$ value < 0.05). Exclusive and common proteins from each comparison are presented as a Venn diagram generated using the InteractiVenn tool[69]. For data visualization, heat maps with $z$-score values of log2 LFQ intensities and volcano plots were built using the open-source statistical programming language R.

GO annotation of biological processes for the proteome of neoplastic islands and tumor stroma was performed using the BinGO plugin[70] within Cytoscape[71], with the significance threshold set at $P$ value < 0.05 using Hypergeometric test. The whole human proteome Gene Ontology (GO) annotation file was used as a reference set. Overrepresented GO terms are represented as bar plots according to the $P$ value of the enrichment analysis. The fold enrichment was calculated between the enriched terms for the upregulated proteins and the downregulated proteins in neoplastic islands, and the terms are presented in a volcano plot (Fig. 2g).

**Discovery proteomics and clinicopathological data correlation.** Linear regression analysis was performed using the R code to evaluate the linear relationship between protein expression and the following clinicopathological variables: age (>40 or <40 years old for island samples, >50 or <50 years old for stroma), sex, smoking habits, tumor size, lymph node metastasis at diagnosis, clinical stage, type of treatment (surgery, surgery and radiotherapy, or a combination of surgery, radiation and chemotherapy), disease-free survival, second primary tumors (from different histological origins), local recurrence, lymph node recurrence, presence of the worst pattern of invasion[11], presence of inflammatory infiltrate, perineural invasion, second histological classification[11], presence of tumor buddings according to the BD model[4], and the WHO histopathological grading system[60]. $P$ value < 0.05 was used to define significance. The Pearson product−moment correlation coefficient was also calculated to measure the strength of the association between the previously described variables.

**IHC of OSCC tissues.** Slides of the OSCC cases (125 cases for the evaluation of neoplastic island proteins and 96 cases for tumor stroma) were incubated with anti-PGK1 (SAB1300102, Sigma) diluted 1:50, anti-NDRG1 (HPA006881, Sigma) diluted 1:100, anti-LTA4H (HPA008399, Sigma) diluted 1:100, anti-CSTB (HPA017380, Sigma) diluted 1:250, anti-COL6A1 (HPA019142, Sigma) diluted 1:1000, anti-ITGAV (HPA004856, Sigma) diluted 1:300 and anti-MB (HPA003123, Sigma) diluted 1:75 according to The Human Protein Atlas and the manufacturers' instructions and were assessed using the Envision detection system (Dako). The control reactions were performed by exclusion of the primary antibodies. The specificities of the antibodies employed were shown by western blot with seven cell line extracts.

**Western blot analysis.** Protein extracts were obtained from BJ-5ta (ATCC CRL-4001, a fibroblast immortalized cell line), CAF (a primary oral cancer-associated fibroblasts, α-SMA positive), SCC-9 (ATCC CRL-1629, a tongue cancer cell line), HSC3 (JCRB 0623, a human tongue squamous cell carcinoma cell line, Osaka National Institute of Health Sciences, Japan), SK-MEL-28 (ATCC HTB-72, malignant skin-derived melanoma cell line), MCF7 (ATCC HTB-22, a breast cancer cell line) and A549 (BCRJ 0033, an epithelial lung cancer cell line) and subjected to western blot (Supplementary Fig. 7). The antibodies were anti-CSTB (1:500, Sigma—HPA017380), anti-LTA4H (1:500, Sigma—HPA008399), anti-NDRG1 (1:500 Sigma—HPA006881), anti-PGK1 (1:500, Sigma—SAB1300102), anti-COL6A1 (1:1000, Sigma—HPA019142), anti-ITGAV (1:1000, Sigma—HPA004856), anti-MB (Sigma—HPA003123). Western blots were performed using 40 μg of total lysate protein and analyzed on a 10–15% SDS-PAGE gel. After incubation with secondary antibodies, visualization of the proteins were achieved by chemiluminescence with the ECL kit (Amersham Biosciences). Anti-ACTB (1:2000, Sigma—A1978) antibody was used as loading control. Uncropped scans of all blots are shown in Supplementary Fig. 7 in the Supplementary Information. The cell lines were tested for mycoplasma contamination.

**IHC data analysis.** Histological slides were independently evaluated by three pathologists in a blinded manner. The examiners were instructed to reach a consensus on discordant cases. The ITF and the inner tumor were scanned under a low power field to select the correct tumor area[28]. We used the protocol of da Silva et al.[27] with modifications. Samples were classified according to the intensity and percentage of immunostaining, both for the ITF and the inner tumor separately. Each intensity score was added to the percentage score to generate a combined score, one score for the ITF and one score for the inner OSCC. The ITF score was deducted from the inner score to determine the difference in expression for each area. Values lower than zero were considered to indicate lower expression in the ITF, values greater than zero were considered to indicate higher expression in the ITF, and negative or missing values, with equal values between the ITF and the inner portion, were considered equal (Supplementary Data 20−22).

Correlations between the immunostaining and clinical parameters of the tumors were performed via crosstabulation and the chi-square test. Furthermore, a survival analysis was calculated using Kaplan−Meier methodology and compared with the log-rank test. In the univariate survival analysis, the comparison was performed between the greater expression in the ITF in relation to the lower

expression in the ITF. For the multivariate survival analysis, the Cox proportional hazard model with a stepwise method was used. For this analysis, the equal expression between the ITF and inner was clustered to the low expression data. $P$ value < 0.05 was used to define significance using Cox proportional hazard model. The criteria for the consideration of a marker for cancer prognosis were based on the REMARK guidelines[51].

**Saliva sample collection.** This study was approved by the Ethics Review Board of the Cancer Institute of São Paulo (ICESP), Octavio Frias de Oliveira, ICESP, São Paulo, SP, Brazil, and Plataforma Brasil through protocol CAAE 30658014.1.1001.0065. Informed consent was obtained from all patients. The procedures used for saliva collection[22] and annotation were performed in accordance with the approved guidelines and experimental protocols defined by the ethics committee of the ICESP, and it comprised a 40-independent patient cohort. Saliva samples were voluntarily obtained from OSCC patients with active lesions at the time of saliva collection and were subsequently distributed based on their clinical stage (TNM), including patients without lymph node metastasis (designated as N0, $n = 14$) and patients with lymph node metastasis (designated as N+, $n = 26$). The detailed clinical and pathological information for the 40 patients enrolled in this study are summarized in Supplementary Data 26. Individuals who had not eaten for at least 1 h first rinsed their mouths with 5 mL of drinking water, and saliva was subsequently harvested without stimulation into a glass receptacle. The saliva samples were aliquoted into 15 mL tubes and frozen at −80 °C for long-term storage until use.

**Proteotypic peptides and transition selection.** The proteins CSTB, LTA4H, NDRG1, PGK1, COL6A1, ITGAV, and MB were selected for verification in saliva patients without (N0) and with (N+) lymph node metastasis. Briefly, three proteotypic peptides per protein were selected based on the number of residues, hydrophobicity, and DDA and/or SRMAtlas evidence[72,73]. Seventeen proteotypic peptides were selected and purchased as crude heavy-isotope-labeled peptide standards (Thermo Fisher Scientific). The stable isotope-labeled peptides (SIL) were synthesized with heavy isotopes on lysine, arginine or leucine (+8, +10, or +7 Da, respectively), localized, preferentially, at the C-terminal of the peptide (Thermo Fisher Scientific). Three transitions were monitored for the light and heavy counterparts of each peptide, with a total of 102 monitored transitions (Supplementary Data 27). Eight or nine peptides with their respective 24 or 27 transitions of the internal retention time standard (Pierce™ Peptide Retention Time Calibration Mixture, Thermo Fisher Scientific) were monitored as a control for retention time shifts in liquid chromatography. A detailed description of the peptide and transition selection is provided (Supplementary Data 27) as follows:

Peptide selection: (1) Peptides were selected based on previous DDA data from the same samples that were used in the discovery phase. With our own DDA data, we generated spectral libraries using Skyline software; (2) In the cases we could not retrieve three proteotypic peptides per protein, considering the rules for proteotypic peptide selection[72,73]; we chose peptides informed by SRMAtlas.

Transition selection: (1) All transitions monitored in saliva were initially obtained by spectral libraries built from our own DDA data and Human plasma DDA data (obtained from the PeptideAtlas repository); (2) In the cases of transitions that were not confidently detected, they were excluded from the method and were complemented with transitions informed by the SRMAtlas during the refinement of the SRM method; (3) The transitions were selected based on the rank of intensity identified in the spectral libraries or the SRMAtlas spectra. The three most intense transitions were preferably selected for further SRM analysis.

**Saliva sample preparation for SRM-MS.** The saliva samples were centrifuged at $1500 \times g$ for 5 min at 4 °C to pellet the debris. The resulting supernatant was collected and quantified using the Bradford assay (Bio-Rad, Hercules, CA, USA). A volume that corresponded to 10 μg of total protein was used for sample preparation, and the sample volumes were adjusted. Ten micrograms of total protein were denatured in urea buffer (100 mM Tris-HCl pH 7.5, 8 M urea, 2 M thiourea, 5 mM EDTA, 1 mM PMSF, and 1 mM DTT) that contained Protease Inhibitor Cocktail Complete Mini Tablets (Roche, Auckland New Zealand). The samples were sonicated for 10 min and subsequently centrifuged at $10,000 \times g$ for 5 min. The supernatant was collected, and the proteins were reduced with 5 mM DDT and alkylated with 14 mM iodoacetamide. Prior to the addition of trypsin, all samples were diluted 1:5 in 50 mM ammonium bicarbonate. Proteins were digested overnight at 37 °C using 1.8 μg of trypsin. After digestion, the reaction was terminated by the addition of trifluoroacetic acid. Desalting was performed by solid-phase extraction using Stage-tips C18 resin[22,63] (with modifications). After vacuum drying, the peptides were resolubilized in 0.1% formic acid. SIL peptides and iRT (Pierce) retention time standards (used here as a quality control for retention time shift) were spiked into 10 μg of desalted saliva.

To prevent bias during the measurements, the data collection was blocked and randomized for each group (N0 and N+). Samples from both the N0 and N+ groups were randomized using the R (v3.4.0) environment. Randomization was applied for each set of technical replicates (Supplementary Data 28). Each sample was analyzed in three technical replicates using the same instrument parameters as described below.

**SRM-MS.** Samples were analyzed on a Xevo TQ-XS triple quadrupole mass spectrometer (Waters, Milford, MA, USA) equipped with an electrospray ion source (Ion Key, Waters, Milford, MA, USA) with MassLynx software (version 4.2).

An aliquot that contained 1 μg of saliva peptide mixture was separated on a trap column (Waters Acquity UPLC BEH C18 130A, 5 μm, 300 μm × 50 mm) and a BEH Shield C18 IonKey column (10 cm × 150 μm ID packed with 1.7 μm C18 particles, Waters, USA) heated to 40 °C. Peptides were maintained at 4 °C in sample manager and loaded onto the column from an Acquity UPLC-Class M LC autosampler (Waters, Milford, MA, USA). Chromatographic conditions were as follows: 60-min gradient at a flow rate of 1.2 μL/min starting with 98% A (water), followed by 40% B (ACN) at 45 min with a step increase, followed by a step increase to 85% B until 47 min and 2% B at 60 min. Targeted acquisition of eluting ions was performed using the mass spectrometer operated in SRM-MS mode with Q1 and Q3 analyzers set to 0.7 Th FWMH and a cycle time of 3 s. For all SRM-MS runs, multiple scheduled injections with a 3-min elution window were used, each targeting three transitions per peptide. The optimal collision energy was determined for each peptide by Skyline[74]. The dwell time for all monitored peptides was automatically set in MassLynx software (v.4.2), from 14 ms to 163 ms, with at least ten points per peak. To avoid carryover between the samples, one blank sample using the trap column (90% isocratic gradient, over 15 min) and one blank sample using both the trap and analytical columns (80% B isocratic gradient, over 10 min) were run.

**Quantitative and statistical analysis.** Visualization and inspection of peaks were manually performed in Skyline. Both the light and heavy peptides were checked regarding the quality of the data by observing the alignment of light and heavy peptide elution times, the co-elution of all three transitions, relative intensity correlation with the spectral library (dotp, close to 1), relative intensity correlation of light with heavy transitions (rdotp close to 1), proximity to the predicted retention time and reproducibility in terms of retention time and intensity between technical replicates (Supplementary Fig. 8−9). All monitored peptides were detected and presented a measured retention time close to the predicted retention time among the replicates ($r = 0.9955$ for set 1 and $r = 0.9994$ for set 2, Supplementary Fig. 8).

The reproducibility based on sample group were assessed by Pearson correlation analysis (Supplementary Fig. 10), and visualization of sample grouping was assessed by unsupervised hierarchical clustering (Supplementary Fig. 11). Each peptide was quantified in a sample by dividing the intensity of the light peptide (sum of light transitions) by the intensity of each reference SIL peptide (sum of heavy transitions) to obtain the light-to-heavy peptide ratio.

Comparison of the levels of the monitored peptides between the groups of patients was performed using Mann−Whitney $U$ test (not log transformed data). $P$ values were adjusted for multiple comparisons using the Benjamini−Hochberg FDR method[75]. Adjusted $P$ values less than 0.05 were considered statistically significant.

**Targeted proteomics and clinicopathological data correlation.** Correlations between the protein abundance in SRM and the clinicopathological data of the tumors were performed by crosstabulation and the chi-square test (Supplementary Data 32). $P$ value < 0.05 was used to define significance.

**Machine learning to predict power of prognostic signatures.** To analyze the prediction power of combinations of peptides and proteins to distinguish OSCC patients by the presence of lymph node metastasis, the peptide and protein abundances of 14 patients (N0) and 26 patients (N+) were used (Supplementary Data 33). We split the dataset (40 samples) into a training set (80% of the dataset) and an independent test set (20% of the dataset).

The training set was used to perform filtering of the variables, cross-validation and running of the SES tool method[76]. A preselection of the appropriate features may facilitate the performance of machine learning, particularly when applied to proteomic and transcriptomic data[77]. We filtered proteins/peptides by the Mann−Whitney $U$ test to obtain only variables with significant differences between the groups ($P$ value < 0.10). We further combined all proteins/peptides to create all possible signatures (size 1 to $N$, where $N$ is the total amount of peptides/proteins) to test in the next step.

To define the best classifier to evaluate the predictive power of signatures, we performed a repeated cross-validation (100 rep. of stratified tenfold cross-validation) on the training set and employed seven different types of machine-learning algorithms (Linear SVM, RBF SVM, Decision Tree, Logistic Regression, Random Forest, Perceptron, and Naive Bayes). For both levels (protein and peptide), Random Forest has the highest performance (Supplementary Data 34-35).

The steps used to obtain and validate the signatures are detailed in Fig. 7a. We performed a repeated cross-validation (100 rep. of stratified tenfold cross-validation) on the training set using Random Forest to classify samples in N0 versus N+, thus creating a list of potential signatures with the highest accuracy scores (Fig. 7b–d; Supplementary Data 36–37). Furthermore, we performed the SES method[76] to identify potential statistically equivalent signatures (Supplementary Data 39). We generated ROC curves considering the 100 rep. of

stratified tenfold cross-validation on the selected signatures (Fig. 7d). We compared the AUC of each signature, as well as their accuracy, specificity, sensitivity, and precision, as shown in Supplementary Data 38, to identify the most relevant results regarding the pairs of Accuracy and ROC AUC. Finally, we validated the models using the independent test set (Supplementary Fig. 12−13). Groups of signatures with both the highest accuracy (≥74% for peptides, >68% for proteins) and the highest AUC were selected as the most relevant features. Moreover, we applied a technique to oversample the training subsets and balance the number of samples in each class, using the Synthetic Minority Over-sampling Technique (SMOTE)[78]. For comparisons based on the AUC, oversampling provides more accurate results than undersampling[79] (Supplementary Data 40−41, Fig. 7e). The 1000 training subsets were synthetically oversampled, and the associated 1000 test subsets were composed of only original samples (nonsynthetic).

## Data availability

The mass spectrometry proteomic data have been deposited in the ProteomeXchange Consortium (http://proteomecentral.proteomexchange.org) via the PRIDE partner repository[80] with the dataset identifier PXD007232 (https://www.ebi.ac.uk/pride/archive/projects/PXD007232).

The SRM analyses for the seven measured proteins are available through the Panorama repository at the following link (https://panoramaweb.org/labkey/saliva_SRM.url). Skyline exported data for all quantified peptides are available in Supplementary Data 29−30. All other data supporting the findings of this study are available from the corresponding author on reasonable request.

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

## Acknowledgements

This work was supported by FAPESP Grants 2009/54067-3, 2010/19278-0, 2013/16483-0, 2014/02288-0, 2016/07846-0, CNPq Grants 152619/2015-1, 470268/2013-1, 305432/2014-1. We acknowledge Prof. Dr. Tsai Siu Mui, CENA, USP for the use of the Leica Laser Microdissection Systems (FAPESP 2009/53998-3). We acknowledge Prof. Dr. Débora Lima Pereira for the assistance in histopathological images. We thank Dr. Ana Karina de Oliveira and Jamile de Oliveira Sá for all the assistance in primary cell culture experiments.

## Author contributions

Drafted the manuscript: C.M.C., C.C.S.M., D.C.G., T.D.R. and A.F.P.L.; conceived and designed the experiments: C.M.C., C.C.S.M., S.D.S., M.A.A.-J., R.D.C., A.F.P.L.; contributed the clinical samples: I.S.-C, T.S., T.B.B., A.R.S.-S. M.A.L., G.C.J., W.A.G.-A.; performed the sample preparation and laser microdissection of tumor tissues: C.C.S.M., N.K.C.; performed the tumor tissue sample preparation for mass spectrometry analysis: C.M.C., R.R.D., A.F.B.L.; performed the LC-MS/MS of tumor tissue samples: R.R.D, B.A.P., C.M.C; performed the proteomics data analysis of tumor tissue samples: C.M.C., C.C.S.M. and A.F.P.L.; performed the statistical analysis of discovery-based proteomic data: C.M.C., C.C.S.M., G.V.M., F.A.M.; performed the western blots: S.Y., D.C.G.; performed the IHC assays: C.C.S.M., C.M.C., S.D.S.; performed the IHC analysis: C.C.S.M., C.R., R.D.C., P.C.R., T.S.; selected the proteotypic peptides: D.C.G., T.D.R., C.M.C.; performed the saliva sample collection: A.C.P.R, C.C.S.M., T.B.B., G.C.J., W.A.G.-A.; performed the sample preparation of saliva samples for mass spectrometry analysis: D.C.G. and T.D.R; performed the SRM analysis: D.C.G., T.D.R., B.A.P., A.F.P.L.; performed the statistical analysis of targeted data: D.C.G., T.D.R., C.R., G.P.T, H.H. and R.M; critically revised the manuscript for important intellectual content: C.M.C, C.C.S.M., T.D.R., D.C.G., H.H., G.V.M., A.R.S.-S, P.C.R., A.F.B.L., A.G., F.P., G.C.J, W.A.G.-A, J.W.F, A.F.P.L. All authors have read and agreed with the final version of the manuscript.

## Additional information

**Competing interests:** The authors declare no competing interests.

