## [Peer Review File · Nature Communications]

Reviewers' comments:

Reviewer #1 (Remarks to the Author):

The current manuscript titled "Discovery and targeted proteomics map of prognostic signature proteins in oral cancer" by Paes Leme and colleagues is an interesting example of how annotated tissue samples could be integrated into a pipeline for the discovery of novel molecular biomarkers of prognosis.

Briefly, the authors analyzed 20 oral cancer tissues that had detailed clinical annotation available. While the number seems small, this is a significant amount of work since each sample was pathologically inspected and then laser-capture micro-dissected to obtain 6 individual regions from each patient. This resulted in 120 samples for proteomic analyses, a significant effort and the authors should be complemented on their work. The main comparisons included tumor vs. stroma and inner tumor regions vs. the invasive front (different comparisons were performed here as well). Using a 1D-LC-MS pipeline, the authors identified ~2,000 proteins and then performed various quantitative comparisons to identify potentially prognostic biomarker proteins. While this is not completely novel, the overall package and rigor is definitely novel and a number of potentially interesting observations were made. Several differentially expressed proteins were then "verified" using the clinically standard assay – IHC (of note the dynamic range of MS is significantly superior to IHC, which often uses antibody that nobody knows what they actually recognize). As a result, ~60% of the tested proteins verified in an additional cohort, the other did not (many reasons exist why). This is an expected verification rate and in-line with several previous proteomics studies. Finally, the authors selected several of their differentially identified proteins and developed target proteomics assays (by SRM) using synthetic peptide standards and quantified their expression in ~30 clinically stratified saliva samples as potential liquid biopsy signatures.

Overall, this is extensive and carefully conducted work from a proteomics group that has focused on head and neck cancer proteomics and biomarker discovery. While years from clinical implementation this work represents an excellent example of a rigorously conducted clinical proteomics project, combining discovery shotgun proteomics, data mining, independent IHC verification and targeted proteomics in a related proximal tissue fluid. The authors should be complemented to their amount of work and effort. These data (and the approach) should be of major interest to the head & neck cancer, but also to the clinical proteomics communities.

There are a number of comments that the authors should address to make their paper even stronger.

Comments:

1) Supplementary Figs 2/3: Not sure these scatter plots are the best way to show this. A heatmap with correlation coefficients would be much better.

2) I wasn't clear about the discovery proteomics: Were the samples normalized by peptides loaded or cell number micro-dissected? Is any of this taken into account?

a. Along these lines (page 7; first paragraph): I was not clear how the authors get from 2,049 identified proteins (which I assume were 1% FDR, based on a typical MaxQuant search) to around 700 proteins used for quantitative comparisons?

b. Why add this subjective filter of "present in a number of samples"? Stats will take care of this?

c. How would the data look if all identified proteins were clustered?

d. The authors mention they removed some stromal samples with low protein counts. Was there anything different when these samples were quantified following protein extraction?

i. What is going on with the P11_I sample (Fig 2b)?

e. Page 7; lines 170/171: Were the statistical analyses corrected for multiple testing?

f. Figure 2: Tumor samples seem to cluster much tighter compared to the stroma samples. Could the authors please comment?

g. Figure 2e: Not sure it's best to perform the GO analyses on the unique proteins in the Venn diagram. Wouldn't it be better to display these data as a Volcano plot and use the significantly differential proteins for GO analysis? Also, what was set as background for this GO enrichment?

3) Figure 3b/d: I am not clear about these box-plots. Why are there more than 20 boxes for ITF/inner tumor in Figure 2b for example? The authors only analyzed 20 samples each. If these are individual samples why is there even a distribution? Each protein in a given sample would have a discrete value (and not a range)? In any event, the differences seem very neglectable in the box plots?

4) The 33 saliva samples: Is there any overlap to the discovery cohort?

a. Most of the proteins used for validation in saliva don't seem to be "classically secreted"? Why are they in saliva?

b. It was interesting to see that some of the selected proteins validated in saliva (or were even detectable, considering they are not secreted proteins), although the clinical question (NO vs. N+) is slightly different.

i. Would combination of some of these peptides into a small signature improve the performance?

5) How were the antibodies evaluated prior to IHC evaluation? It is quite common now that cell lines that express the proteins of interest (and kd-versions thereof) are pelleted and added to the analyses. How did the authors quality check these antibodies (the biggest bottleneck in the validation pipeline)?

6) Page 23 (protein extraction): How did the authors deparaffinize their samples prior to proteomics. I couldn't find this in the protocols?

7) Page 28 (SRM section): Peptides were selected based on SRMAtlas. Did the authors not make use of their own discovery data?

8) Page 29 (salvia samples): Not sure adding protease inhibitors to a sample that was already denatured by 8M urea and then immediately processed for proteomics is the best choice. This might actually reduce digestion efficiency?

Minor comments:

1) Paper could benefit from some additional proof reading

2) Some of the figures were of poor resolution. Some of the figures even had visible borders (Figure 2)?

Reviewer #2 (Remarks to the Author):

The manuscript "Discovery and targeted proteomics map of prognostic signature proteins in oral cancer" describes the use of proteomics to identify diagnostic/prognostic markers of oral cancer. They have used IHC of oral cancers and salivary SRM analysis of markers in NO and N+ tumors to indicate that 4 markers of tumors and 3 markers of stroma are useful biomarkers of local recurrence and lymph node metastasis. While the techniques employed are standard and extensive, convincing data is not presented.

Major deficiencies

1. Title and abstract are vague. If the authors believe that cystatin B and other 6 markers are cancer biomarkers, the information should be spelled out in the title and in the abstract. They need to state that loss or reduced expression of cystatin B (CST B) as a diagnostic marker of local recurrence in the title. Similarly, authors need to make a clear statement regarding up or down

regulation of the 7 markers in local recurrence and nodal metastasis in the abstract.

2. It is difficult to read the manuscript with redundant and convoluted sentences throughout the manuscript. For example, lines 249 to 258 talks about the utility of CST B and PGK1 in recurrence and survival. The results seem to suggest that down regulation of CST B and up regulation of PGK1 have a role in local recurrence which is not stated clearly. Lines 251/252 indicating significant 5-year disease free survival is linked with previous line making it difficult to know the real association, up or down regulation. Similarly, lines 257/258 on 3-year DFS is not clear on the reference to lower expression of CST B.

3. Introduction, results and discussion need to be shortened by deleting repetitive and vague sentences and making clear statements on the up or down regulation of identified bio markers.

4. IHC for CST B and MB (figure 4) are of poor quality making it difficult to visualize the staining. Better staining and representative higher magnification figures are required to convince hybridization of these markers to tumor cells or stroma respectively.

5. Authors have not provided justification for the selection of these markers over differential expression of markers with higher significance in figures 3A and 3C. Also, the relevance of the identified markers to cancer development is not stated for all 7 markers.

6. There seems to be only one N0 sample with higher expression for COL6A1 expression in figures 6C-D. To some extent it is also true for ITGAV making it difficult to accept these two as significant markers of N0 over N+ tumors.

7. There are two previous reports by the authors on the identification of tumor specific markers in cancer associated fibroblasts and saliva relating to oral tumors. It is surprising authors have not analyzed these markers in IHC to show their clinical relevance with respect to markers identified in the present investigation.

8. Line 433 to 435 points to a mean recurrence period of 7.5 months in other studies and a mean recurrence of 12 months with lower CST B expression in the present investigation. This contradiction should be clarified instead of a vague statement in line 436 that CST B expression is associated with local relapse of OSCC.

Reviewer #3 (Remarks to the Author):

Summary

This work describes a project aiming for discovery and validation of prognostic biomarkers in OSCC. The discovery part consist of LCM dissection of tumor tissue where neoplastic and stromal regions are analysed seperately whith respect also to regional location (inner vs ITF). Targeted validation of candidate biomarkers was then performed using both IHC and SRM.

Overall

The search for biomarkers to detect cancer, to prognosticate outcome or to predict response to therapy is an important but difficult part of cancer research. The approach here used to identify prognostic biomarkers is sound and should have the potential to identify candidate biomarkers. Especially the multi-layered analysis used, with a discovery in the tumor itself followed by an evaluation of biomarker presence in an easily accessible body-fluid is likely to increase the chance of finding good biomarker candidates.

In deed, the analysis produce a few candidate biomarkers with potential use for prognostication in OSCC. Specifically CSTB is claimed to be an independent marker for local recurrence in tissue, and five proteins are claimed to be specifically associated with lymph node metastasis. Saliva biomarkers of tumor spread should undoubtedly be of interest to others in the community.

Unfortunately, I am not convinced by the performance of the analysis and the interpretation of the results as detailed below. It is not fully proven that the suggested biomarkers have the prognostic

value that is claimed in the manuscript. For these reasons it is not clear that the work is suitable for Nature Communications as presented.

In general, the manuscript would also benefit from more thorough proof-reading. In many cases the readability was hampered by misplaced modifiers or unclear references. Also, in some cases numbers are not matching between different parts of the manuscript. As an example, the text states that four proteins are higher in ITF stroma and one protein is higher in inner stroma (line 195-200). The text also refers to Table 1, but in table 1 only two proteins are higher in ITF stroma (COL6A1 and MB) while three proteins are higher in inner stroma. Several more similar examples are present and collectively the reader has to spend a lot of time trying to understand what the authors mean.

Discovery Proteomics

The discovery part of this work was performed using label free quantification (LFQ) of proteins across 120 different samples. A major problem with LFQ is that the identification overlap between samples often limits the number of proteins that can be compared across samples. In the current work this is illustrated by the total number of identified proteins (e.g. 2049 in neoplastic cells) compared to the number of proteins used in the statistical analysis after filtering away proteins with low overlap between samples (799). This is also illustrated by the fact that six samples from the stroma set were discarded due to low number of identified proteins in three of the samples. Such filtering is commonly used in label free experiments.

More problematic is that it is unclear how the decision was made to exclude the six stroma samples, while still including neoplasm samples that are also low in number of identified proteins (Supplementary Fig 2 and Supplementary Table 5, e.g. neoplasm 11_I, 3_F and 9_F are missing 87, 66 and 62% respectively of the 799 proteins evaluated). This becomes even more problematic since the authors use imputation to replace missing values with random numerical values corresponding to a low abundance measurement. A missing value in MS based proteomics does not necessarily mean that the protein is missing or very low, just that the MS-instrument did not measure it. In addition, there is a skewness in the distribution of missing values between the ITF and inner neoplasm samples so that ITF samples have more missing values than inner samples in general (number of NaNs per sample in Supplementary Table 5, median 128 for ITF vs 83 for inner samples). Such imbalance will affect the quantitative and statistical analysis, and could potentially have caused the asymmetrical distribution of values in the volcano plot in Figure 3a.

The statistical analysis of proteins that are differentially expressed between ITF and inner samples in neoplasm or stroma is based on a paired t-tests for the 799 and 704 overlapping proteins that passed the filtering criteria. From what I understand of the materials and methods, no correction for multiple testing has been done and consequently many of the 32 and 101 proteins identified as differentially expressed will be false positives. With a p-value threshold of 0.05 we would expect to find $799 * 0.05 = 40$ proteins to be significant just by chance.

IHC validation

The selection of proteins for the IHC validation was based on the association between the candidate biomarkers (32 and 101) and clinicopathological parameters. To me this choice was not very transparent, and in some cases I did not find the underlying data. As an example, Table S1 holds the clinicopathological data for the discovery cohort, but it does not contain information about treatment or second primary tumor. Still, Table 1 indicate that there is a statistically significant association between e.g. treatment and CSTB or PGK1 and second primary tumor. The authors state that the IHC analysis partially confirmed the results from the discovery cohort. I would have liked to see an additional representation of this data since it is difficult to evaluate their claims based only on the heatmaps in Figure 4. A summary of the quantifications as well as a statistical analysis would help.

Further, the authors keep LTA4H, PGK1 and ITGAV for further analysis even though the IHC

validation was not in concordance with the discovery results for these proteins. The rationale being that the IHC results were unreliable since the proteins are expressed also in the stroma (lines 235 and 394) or that the antibody is non-specific (line 393). These are potential explanations, but a more obvious one would be that the proteins were not validated because the assumptions were not true. Also, with IHC analysis it should be possible to distinguish the expression of proteins between neoplasm and stroma. Further, if the antibodies were non-specific then the results would have little clinical value.

To evaluate the survival analysis (Figure 5) it would be good if the group sizes were indicated for each group in each Kaplan-Meier plot.

The strongest finding from the IHC validation was an association between local recurrence and low expression of CSTB in the ITF demonstrated in the multivariate analysis ($p=0.0478$). Even though this result is close to the significance threshold, it is fully in line with the result from the discovery cohort.

SRM validation

It is not entirely clear to me how the targeted SRM analysis relates to the discovery and IHC analysis. In the two latter, regional differences in protein expression is considered and used for correlation to clinical parameters. In the SRM analysis saliva from node positive cancer patients is compared to saliva from node negative patients. As I understand from Figure 6f, the authors suggest that proteins that are higher in the inner part of the tumor (and therefore closer to the oral epithelium) are more likely to end up in the saliva. This may be true, but it would not explain why there would be a difference in the abundance of these proteins between node positive and node negative patients. Irrespective of the presence of tumor cells in the lymph nodes all patients will have inner tumor cells that are closer to the oral epithelium and able to release proteins into the saliva.

It is also difficult to evaluate the results from the SRM analysis for several reasons. In Figure 6a I am missing error bars that would help the reader to estimate variation in the measurements. In Figure 6c the scale is so compressed that it is difficult to evaluate the results. Also, for LTA4H and COL6A1 the signal of the peptides are much smaller than the spike in (Ratio L/H <0.01). It would be better to titrate the spike-in standard to more closely resemble the concentration of the peptides in the real samples. As it looks now it is difficult to evaluate the reliability of the quantification or if the signal generated is close to the analytical noise. At least a few example spectra could be added as supplementary figures to convince the reader.

Further, for the examples LTA4H and COL6A1 (Figure 6d), even if the ROC and Sens./Spec. curves indicate some prognostic value for the two proteins, the distribution graphs does not really support a good resolution of node positive and node negative patients based on the levels of the proteins in saliva.

Reviewer(s)' Comments to Author:

Reviewer #1:

Remarks to the Author:

The current manuscript titled “Discovery and targeted proteomics map of prognostic signature proteins in oral cancer” by Paes Leme and colleagues is an interesting example of how annotated tissue samples could be integrated into a pipeline for the discovery of novel molecular biomarkers of prognosis.

Briefly, the authors analyzed 20 oral cancer tissues that had detailed clinical annotation available. While the number seems small, this is a significant amount of work since each sample was pathologically inspected and then laser-capture micro-dissected to obtain 6 individual regions from each patient. This resulted in 120 samples for proteomic analyses, a significant effort and the authors should be complemented on their work. The main comparisons included tumor vs. stroma and inner tumor regions vs. the invasive front (different comparisons were performed here as well). Using a 1D-LC-MS pipeline, the authors identified ~2,000 proteins and then performed various quantitative comparisons to identify potentially prognostic biomarker proteins. While this is not completely novel, the overall package and rigor is definitely novel and a number of potentially interesting observations were made. Several differentially expressed proteins were then “verified” using the clinically standard assay – IHC (of note the dynamic range of MS is significantly superior to IHC, which often uses antibody that nobody knows what they actually recognize). As a result, ~60% of the tested proteins verified in an additional cohort, the other did not (many reasons exist why). This is an expected verification rate and in-line with several previous proteomics studies. Finally, the authors selected several of their differentially identified proteins and developed target proteomics assays (by SRM) using synthetic peptide standards and quantified their expression in ~30 clinically stratified saliva samples as potential liquid biopsy signatures.

Overall, this is extensive and carefully conducted work from a proteomics group that has focused on head and neck cancer proteomics and biomarker discovery. While years from clinical implementation this work represents an excellent example of a rigorously conducted clinical proteomics project, combining discovery shotgun proteomics, data mining, independent IHC verification and targeted proteomics in a related proximal tissue fluid. The authors should be complemented to their amount of work and effort. These data (and the approach) should be of major interest to the head & neck cancer, but also to the clinical proteomics communities.

Response: We would like to thank the reviewer for the positive comments concerning our manuscript.

There are a number of comments that the authors should address to make their paper even stronger.

Comments:

1) Supplementary Figs 2/3: Not sure these scatter plots are the best way to show this. A heatmap with correlation coefficients would be much better.

Response: We would like to thank the reviewer for this suggestion. We agree with the reviewer about this representation, and as suggested, we included a heat map to show the correlation between the samples. These figures are included in the main text as Figures 2e and 2f in the revised version. We also kept the correlation plot of the original version, which is now shown in Supplementary Figures 2 and 3, which show the exact value of correlation between each pair of samples. In addition, it is important to note that we corrected the correlation plot of the neoplastic island samples, and now all the values include Pearson correlation coefficients (R).

2) I wasn't clear about the discovery proteomics: Were the samples normalized by peptides loaded or cell number micro-dissected? Is any of this taken into account?

Response: We would like to thank the reviewer for the opportunity to clarify this point. To normalize the quantity of tissue collected, we isolated approximately $1 \times 10^5 \mu\text{m}^2$ of tissue for small neoplastic islands and $1 \times 10^6 \mu\text{m}^2$ for large neoplastic islands and tumor stroma, and $10 \mu\text{m}$ tissue sections of each sample were prepared using laser microdissection.

The only additional data normalization used here was protein quantification using the LFQ intensity parameter from MaxQuant software, which normalizes the global differences in sample loading. According to the algorithms described by Cox et al. (2014), the LFQ intensity allows relative protein quantification across all samples (Tyanova et al., 2016). The advantages of the MaxLFQ algorithm were demonstrated in a study by Cox et al. (2014) comparing quantification using spectral counts, summed intensities, and MaxLFQ; MaxLFQ resulted in better quantification accuracy. The MaxLFQ algorithm produced better results for FFPE samples compared with analyses based on intensity or MS count values. In fact, we identified our peptide list with intensity values and without MS counts or without intensity values and with MS counts, but in both cases, LFQ intensity values were present.

It is important to emphasize that we made every effort to quantify the proteins and peptides before injection onto the LC-MS/MS, but because of the type of sample (FFPE material from oral cancer patients), it was very challenging to increase the efficiency of extractions and to obtain more material from each paraffin block. Therefore, we normalized the samples according to the microdissected area, as indicated previously.

We would like to share some of our protocols used to evaluate the yield and reproducibility of peptide quantification using the Pierce™ Quantitative Colorimetric Peptide Assay Kit, Thermo Fisher Scientific:

- We prepared two 6-point standard curves (0.31 to 10 μg) with the standard peptide mixture provided with the kit (Curve 1 and Curve 2). We obtained R^2 values of 0.9898 and 0.9917 for Curves 1 and 2, respectively.
- A microdissected FFPE tissue sample extracted from a 3 mm^2 segment from a 10 μm tissue section was trypsin-digested according to the protocol described in the manuscript, and after the desalting step, the sample was resuspended in 45 μl of water. The peptide concentration was 0.096 $\mu\text{g}/\mu\text{l}$, and thus, 4.32 μg of peptide mixture was isolated, which represents 1.44 μg of peptide per 1 mm^2 of tissue. Considering that we isolated an average microdissected tissue

area of 0.1 mm², 1 mm², and 1 mm² for small neoplastic islands, large neoplastic islands, and stroma, respectively, there was insufficient material to perform quantification. Considering the limits of sensitivity of the kit (<0.03125 µg/µl in the standard curve did not produce any readout), at least 10 µl of sample would be needed for quantification, and thus, we did not have sufficient material to perform both this quantification and LC-MS/MS analysis.

- Considering these calculations and that we resuspended all 120 samples in 15 µl of 0.1% formic acid and injected 4.5 µl, we injected approximately 400 ng onto the column.
- Based on the tests performed, we concluded that the quantification was reliable for microdissected samples, but for the limited quantities of samples used in this study, this method lacked sufficient sensitivity and was not suitable due to the quantities needed for quantification. We also tested other methods, such as the BCA Protein Assay Kit (Thermo Fisher Scientific), microBCA Protein Assay Kit (Thermo Fisher Scientific), and Bradford (BioRad) methods, but quantification of low concentrations of peptides was not successful.

- a. **Along these lines (page 7; first paragraph): I was not clear how the authors get from 2,049 identified proteins (which I assume were 1% FDR, based on a typical MaxQuant search) to around 700 proteins used for quantitative comparisons?**

Response: We would like to thank the reviewer for the opportunity to clarify this point. The reviewer is correct; the total number of proteins identified in the neoplastic island samples corresponds to 2,049 entries (after excluding reverse sequences and those identified “only by site”, Supplementary Table 4), with a 1% FDR at the peptide and protein levels for MaxQuant search. However, we applied a filter for valid values, i.e., considering the 20 samples of neoplastic islands in each group (invasive tumor front – ITF, or inner tumor); a protein was considered quantified when it had a valid value in at least 10 samples of at least one group (Supplementary Table 5), which reduced the number to 799 proteins.

The same approach was applied for the stroma tumor samples, in which 1,733 proteins were identified, and after filtering the entries of the reverse database and those identified “only by site” (Supplementary Table 9) with a 1% FDR at the peptide and protein levels and requiring 8 valid values in at least one group, the number of proteins was reduced to 704 (Supplementary Table 10). The number of valid values can be assumed to be 1/3 to 1/2 of the total number of samples in a group.

- b. **Why add this subjective filter of “present in a number of samples”? Stats will take care of this?**

Response: We are not entirely sure to which filter the reviewer is referring when indicating “present in a number of samples”. If the reviewer is referring to the filtering of valid values, as we mentioned in the previous answer (2-a), we applied this filter to remove proteins that were only quantified in a few samples, which may represent poor identification and quantification. This filter improves the reliability of protein identification and quantification across samples.

The number of valid values used in our study ranges from one-quarter to half of the number of samples in a group; thus, the protein would have quantification values for at least 50-75% of the samples, which is in accord with the findings in previously published manuscripts (Cox et al., 2014; Tyanova et al., 2016; Räschele et al., 2015). To avoid the removal of a large number of entries, we decided to make the cutoff for valid values to require presence in half of the number of patient samples used in each dataset, i.e., eight valid values in at least one group of tumor stromal samples (composed of 17 patients) and 10 valid values

in at least one group of neoplastic islands samples (composed by 20 patients). To provide the reviewer with information regarding the effect of applying the valid value filter and imputation on statistical analysis, we also performed the two tests without filtering and with/without imputation and we observed the following:

Test 1: Without applying the filter of valid values:

Tumor Stromal samples:

- **Without applying the filter of valid values followed by the replacement of missing values (imputation)**, we verified 115 significant proteins. However, THBS2 was not in the list, whereas the other proteins (COL6A1, FSCN1, ITGAV, and MB) were significant (P -value <0.05 , Student's t -test);
- **Without applying the filter of valid values and not replacing the missing values (imputation)**, we found 76 significant proteins, including COL6A1, ITGAV, FSCN1, MB, and THBS2.

Neoplastic island samples:

- **Without applying the filter of valid values followed by the replacement of missing values (imputation)**, we verified 81 significant proteins. However, LTA4H was not in the list, whereas the other proteins (CSTB, NDRG1, and PGK1) were significant (P -value <0.05 , Student's t -test);
- **Without applying the filter of valid values and not replacing the missing values (imputation)**, we found 74 significant proteins. However, LTA4H was not in the list, whereas the other proteins (CSTB, NDRG1, and PGK1) were significant (P -value <0.05 , Student's t -test).

Test 2: applying the filter of valid values:

Tumor Stromal samples (8 valid values):

- **With imputation** (the analysis we showed in the main text), we had 101 differentially abundant proteins;
- **Without imputation**, this value decreased to 71 differentially abundant proteins. However, there was no effect on the statistical significance of the selected tumor stromal proteins, COL6A1, ITGAV, MB, FSCN1, and THBS2.

Neoplastic island samples (10 valid values):

- **With imputation** (the analysis we showed in the main text), we had 32 differentially abundant proteins;
- **Without imputation**, this value increased to 64 differentially abundant proteins. However, in this analysis, only LTA4H was not considered significant (P -value <0.05 , Student's t -test).

Therefore, statistical analysis will not necessarily solve this problem, and the decision whether to consider valid values or to apply imputation was guided by the requirement for all values to achieve clinical correlation. Moreover, we considered that in this discovery phase, as previously mentioned, it was critical to evaluate the clinicopathological parameters to select the proteins for evaluation in the next steps of verification. Considering that some statistical

methods, such as linear regression analysis for correlation with clinical data, require all values to be present, the replacement of missing values by data imputation may be necessary after filtering proteins based on valid values; we imputed the missing values using the Perseus software, which chooses random values from a distribution mean to simulate expression below the detection limit (Tynova et al., 2016; Hubner et al., 2010; Marakalal et al., 2016; Robles, et al., 2014).

Another point that may also be important to highlight is that although missing values are common in shotgun experiments and especially in this study, we believe that the missing values can be associated with several factors. For instance, missing values can be related to the fixation process used for FFPE samples, which induces molecular modifications, such as crosslinking and methylene bridges between amino acid residues. Additionally, the period of time that the samples have been stored, constraints of sample processing, resistance to protein extraction, and accessibility to cleaving sites can all influence missing values (Gustafsson et al., 2015; Magdeldin and Yamamoto, 2012; Lemaire et al., 2007; Sprung et al., 2009; Craven et al., 2013; Nirmalan et al., 2008; Bernard Metz et al., 2004).

c. How would the data look if all identified proteins were clustered?

Response: We included in the supplemental material a heat map showing all the proteins identified for the neoplastic island samples and all the proteins identified in the stroma tumor samples (Supplementary Figure 5). Clustering analysis of all identified proteins was performed using Z-scored log₂ LFQ intensity values in the R environment; the Euclidean distance and complete linkage was used for neoplastic island samples, and the Euclidean distance and average linkage was used for tumor stroma samples. In this analysis, one can observe that some proteins were identified and quantified in a small number of samples, as highlighted by the “gap” present in the central area of the heat map. As mentioned before, these entries were removed by filtering for valid values in at least 8 samples for stroma and in at least 10 samples for neoplastic islands, as previously described (in question 2a).

d. The authors mention they removed some stromal samples with low protein counts. Was there anything different when these samples were quantified following protein extraction? i. What is going on with the P11_I sample (Fig 2b)?

Response: We would like to thank the reviewer for pointing this out and giving us the opportunity to explain in detail some of the decisions that were made regarding the data analysis.

All FFPE tissue samples were subjected to the same protocols.

Regarding stroma samples:

The decision to exclude P11F, P19F, P19I, and P20I tumor stroma samples was made for the following reasons:

- We observed low-intensity spectra, resulting in a low number of protein identifications. Only 106 proteins for the P11F sample, 107 proteins for P19F, 246 proteins for P19I, and 289 proteins for P20I were identified after applying the filters (reverse, “only identified by site”, valid values) (Response Figure 1A-B). We also observed, as mentioned in the original version

of the manuscript, through histogram analysis (Supplementary Figure 4), a higher number of missing values in these samples (Response Figure 1C).

- In addition, we observed correlation values (Pearson correlation) ranging from 0.455 to 0.762 for P11F, 0.277 to 0.775 for P19F, 0.443 to 0.775 for P19I, and 0.524 to 0.825 for P20I. These samples exhibited the highest numbers of missing values before and after the application of all filters, as shown in the figure below (Response Figure 1A-C).
- It is important to state that although the internal quality control used in our group showed that the trypsin autolysis peaks presented the expected retention time across most of the samples (m/z 523.2855, +2, retention time 22.8 min; m/z 421.7584, +2, retention time 28.7 min; m/z 737.7062, +3, retention time 83.6 min), we found that samples 20 and 11 presented higher shifts in terms of predicted retention time. The removal of these samples from the analysis improved the coefficient of variation for the trypsin peptides (Response Figure 1D).

D

Trypsin peptides	Mean (min)	SD	CV % Before removing P11	CV % After removing P11F	CV % After removing P11I
Peptide 1	21.76	2.04	9.40	7.46	9.48
Peptide 2	17.30	1.20	6.93	6.2	6.91
Peptide 3	79.50	2.24	2.82	2.73	2.77

Response Figure 1: Quality control of the DDA runs (Tumor stroma samples). The bar plots show the total number of identified and quantified proteins (A) and box plots (B) show the LFQ intensity values found in each tumor stromal sample. The number of missing values in each stroma sample is presented as bar plots (C). All the graphs were performed after all filters (reverse, ‘only identified by site’ and at least 8 valid values per group) were applied. (D) Trypsin autolysis peaks presented the expected retention time across most of the samples (m/z 523.2855, +2, retention time 17.30 min; m/z 421.7584, +2, retention time 21.76 min; m/z 737.7062, +3, retention time 79.50 min), except for P11E and P20F (where E corresponds to the stroma from ITF and F to the stroma from inner tumor).

Regarding neoplastic islands samples:

Regarding the previous figure 2b (now figure 2c), sample P11_I.

- P11_I is composed of samples from the inner tumor area of (i) small neoplastic islands and (ii) large neoplastic islands and contain 32 and 85 quantified proteins, respectively. The paired P11_F is composed of samples from the ITF area of (i) small neoplastic islands and (ii) large neoplastic islands with 93 and 619 quantified proteins, respectively (excluding the reverse database and identified “only” by site entries).
- Thus, excluding the P11 samples, we would discard four samples, including samples that had an acceptable number of identified and quantified proteins.
- Similarly, P3_I is composed of samples from the inner tumor area of (i) small neoplastic islands and (ii) large neoplastic islands, with 253 and 961 quantified proteins, respectively. The paired P3_F is composed of samples from the ITF area of (i) small neoplastic islands and (ii) large neoplastic islands, with 175 and 98 quantified proteins, respectively (excluding the reverse database and identified “only” by site entries). Additionally, P9_I is composed of samples from the inner tumor area of (i) small neoplastic islands and (ii) large neoplastic islands, with 54 and 792 quantified proteins, respectively. The paired P9_F is composed of samples from the ITF area of (i) small neoplastic islands and (ii) large neoplastic islands, with 75 and 297 quantified proteins, respectively (excluding the reverse database and identified “only” by site entries).
- Removal of these samples from the analysis did not improve the coefficient of variation (Response Figure 2D).
- In addition, considering the internal quality control used in our group, the trypsin autolysis peaks presented the expected retention times across most of the samples (m/z 523.2855, +2, retention time 22.8 min; m/z 421.7584, +2, retention time 28.7 min; m/z 737.7062, +3, retention time 83.6 min) (Response Figure 2).

D

Trypsin peptides	Mean (min)	SD	Sample number				
			CV % Before removing P11	CV % After removing P11A	CV % After removing P11B	CV % After removing P11C	CV % After removing P11D
Peptide 1	22.67	2.38	10.51	10.57	10.57	10.51	10.41
Peptide 2	17.61	1.37	7.78	7.80	7.81	7.77	7.79
Peptide 3	80.59	3.88	4.82	4.83	4.84	4.83	4.83

Response Figure 2: Quality control of the DDA runs (neoplastic island samples). The bar plots show the total number of identified and quantified proteins (A) and box plots (B) show the LFQ intensity values found in each neoplastic island samples. The number of missing values in each neoplastic island sample is presented as bar plots (C). All the graphs were performed after all filters (reverse, ‘only identified by site’ and at least 10 valid values) were applied. (D) Trypsin autolysis peaks presented the expected retention time across most of the samples (m/z 523.2855, +2, retention time 17.30 min; m/z 421.7584, +2, retention time 21.76 min; m/z 737.7062, +3, retention time 79.50 min), including trypsin peptides of P_11 samples (visualized inside red boxes), except for P1, P2 and P3 samples.

e. Page 7; lines 170/171: Were the statistical analyses corrected for multiple testing?

Response: Statistical analysis for the discovery phase was not corrected for multiple testing. As a requirement in the proteomics field, we applied Benjamini-Hochberg using the Perseus software for multiple testing corrections; however, this correction resulted in no proteins with significant difference between the compared groups (front and inner tumor).

We also tested the multiple hypothesis corrections using Storey’s test (Storey et al., 2004). The q-values were calculated considering a maximum FDR of 5%, which resulted in no significant proteins from the neoplastic islands and 13 significant proteins from the stroma dataset (Response Table 1).

Response Table 1: Storey’s test for multiple test correction of tumor stroma dataset.

P-value (Student's t-test)	q-value	Gene name
0.000483	0.040439	ITGAV
0.001107	0.049698	LDHB
0.000573	0.041084	ANXA2
0.000446	0.040439	RNASE2
0.001144	0.049698	LCP1
0.00102	0.049698	FLNA
0.000272	0.034091	TGM2
5.73E-05	0.01076	COL14A1
0.001247	0.049698	PLEC
6.43E-05	0.01076	PTGIS
0.000926	0.049698	FSCN1
4.84E-05	0.01076	RCN3
0.001287	0.049698	MXRA5

Considering that the list of statistically significant proteins using Student's t-test is not the endpoint of this study, we first chose to filter out proteins that did not correlate with the clinicopathological parameters instead of adjusting the P-values. Additionally, we performed several layers of downstream analysis, including IHC verification, and targeted proteomics to verify the proteins. In the case of the SRM data, in the revised version, the statistical analyses were corrected for multiple testing (Figure 6A, main text), reinforcing the verification of the six candidates (CSTB, ITGAV, NDRG1, LTA4H, COL6A1, PGK1) obtained from DDA data.

f. Figure 2: Tumor samples seem to cluster much tighter compared to the stroma samples.

Could the authors please comment?

Response: At first glance, the neoplastic island samples seem to cluster more tightly if you evaluate the protein expression patterns in the clustering rows. However, one might notice in the dendrogram analyses (Figures 2c and 2d, main text), which show the groups of samples according to their patterns of similarity, that the two main clusters are composed of samples from both neoplastic island areas (ITF and the inner tumor) that were not clustered together in Figure 2c (main text). In fact, the heat map with the stroma tumor samples exhibited a better grouping of samples compared to the ITF or the inner tumor, as observed by the dendrogram and the different colored lines shown underneath (Figure 2d).

- dendrogram of neoplastic island samples, grouped by the Euclidean distance and complete ligation:

- dendrogram of tumor stroma, grouped by the Euclidean distance and average ligation:

Regarding the biological aspect of this effect, the reduced clustering ability observed in neoplastic islands compared to the stroma tumor area is associated with different efficiencies of protein extraction from FFPE samples and with the intrinsic heterogeneity of OSCC tumors. This effect is also expected because oral squamous cell carcinomas are known for their biological variability, which produces specific clinical behaviors, e.g., it has been observed that tumors at the same stage present different clinical outcomes (Noguti et al., 2012; da Silva et al., 2012; Almangush et al., 2015).

- g. **Figure 2e: Not sure it's best to perform the GO analyses on the unique proteins in the Venn diagram. Wouldn't it be better to display these data as a Volcano plot and use the significantly differential proteins for GO analysis? Also, what was set as background for this GO enrichment?**

Response: We would like to thank the reviewer for the suggestion. Enrichment analysis of the original manuscript was performed using the whole human proteome Gene Ontology (GO) annotation file as a reference set, and in the revised manuscript, this information is indicated for exclusive proteins in the Materials and Methods section of the main text.

As suggested by the reviewer, in the revised version, we also performed an enrichment analysis for the 601 differentially expressed proteins (P -value <0.05 , Student's t -test) between neoplastic islands and tumor stroma. In this analysis, we used the BinGO plugin within Cytoscape with the whole human proteome Gene Ontology (GO) annotation file as a reference set. The biological processes according to the fold enrichments are represented as a volcano plot. After BinGO exported the numbers of genes associated with each GO term and the number of genes from our dataset found in the term, the fold enrichment was calculated for the up-regulated proteins and the down-regulated proteins in neoplastic islands. The new Figure 2g is included in the revised manuscript with specific comments about the new results (on page 8).

- 3) **Figure 3b/d: I am not clear about these box-plots. Why are there more than 20 boxes for ITF/inner tumor in Figure 2b for example? The authors only analyzed 20 samples each. If these are individual samples why is there even a distribution? Each protein in a given sample would have a discrete value (and not a range)? In any event, the differences seem very neglectable in the box plots?**

Response: We would like to thank the reviewer for the opportunity to improve this result. The previous Figures 3b/d were replaced in the revised version to better represent the results.

Laser microdissection was used to isolate the tumor stroma and neoplastic cells of small or large islands both in the ITF and the inner tumor area of the 20 samples from patients with OSCC. Thus,

we had a total of 80 samples of neoplastic islands. These 80 samples were represented in the box plots of previous Figure 3b:

- 20 samples of small neoplastic islands from the ITF,
- 20 samples of large neoplastic islands from the ITF,
- 20 samples of small neoplastic islands from inner tumor,
- 20 samples of large neoplastic islands from the inner tumor.

The microdissection of the tumor stroma area from the 20 samples of patients with OSCC resulted in a total of 40 samples that were represented in the box plots of previous Figure 3d:

- 20 samples of stroma from the ITF,
- 20 samples of stroma from the inner tumor

We agree with the reviewer that the graphs in the original version do not adequately represent the data, and thus, we have replaced the box plots with line graphs (Figures 3b and 3d); the y-axis shows the \log_2 LFQ intensity values of each of the 7 selected proteins, and the x-axis shows the samples according to the different runs. Thus, the new figure presents 40 points for the neoplastic island samples after combining data from the small and large islands in MaxQuant (Figure 3b).

The box plots with the peptide intensity values were manually reanalyzed in the R environment to build box plots using the raw intensity values of peptides from the MaxQuant results instead of from each raw file, as was performed in the original version by MSstats. Additionally, these “neglectable differences” are attributed to the normalization performed by MSstats, which equalized the medians. We agree that this approach is not desired in this analysis, and as explained, we show in the main text the LFQ intensity of the 7 selected proteins (Figures 3b and 3d). Additionally, in the box plots we show the abundance of peptides without equalizing the medians; this analysis was moved to the Supplemental Material (Supplementary Figure 6) of the revised manuscript.

4) The 33 saliva samples: Is there any overlap to the discovery cohort?

Response: We strongly appreciate the comments by the reviewer and the opportunity to answer these questions. We have approximately 14 overlapped cases between DDA and IHC patient cohorts. For SRM analysis, we have an independent patient cohort. Fortunately, we were also able to include 4 new N0 patients and 3 new N+ patients from whom we collected saliva and performed analyses during manuscript revision. We have improved these details in the Materials and Methods and Supplementary Table 17 and 19 of the revised version.

CSTB: 13 cases are shared between IHC and DDA.

LTA4H: 14 cases are shared between IHC and DDA.

NDRG1: 14 cases are shared between IHC and DDA.

PGK1: 14 cases are shared between IHC and DDA.

COL6A1: 14 cases are shared between IHC and DDA.

ITGAV: 7 cases are shared between IHC and DDA.

MB: 8 cases are shared between IHC and DDA.

a. Most of the proteins used for validation in saliva don't seem to be “classically secreted”? Why are they in saliva?

Response: This is a very interesting issue, and we thank the reviewer for this comment. Saliva is a very complex fluid composed mostly of water (97–99.5%), and the other components are electrolytes, proteins, peptides, polynucleotides, hormones, enzymes, cytokines, antibodies, and other substances (Ship and Fisher, 1997; Carpenter, 2013). These components originate from major salivary glands

and minor glands (Humphrey and Williamson, 2001). Other known components include epithelial cells (Dawes, 2003), resident microbial communities (Takeshita et al., 2016), exosomes, microvesicles, and apoptotic bodies (Xiao and Wong, 2012, Mengxi et al., 2017). The composition of saliva is very dynamic, and the best understood mechanism of transport into saliva is from serum into the salivary glands by active transport, passive diffusion, or simple filtration; acinar cells also pump sodium ions into the duct (Lee and Wong, 2012). Most compounds found in blood are also present in saliva, and therefore, saliva can host proteins from diverse sources.

Specifically, the proteins CSTB, LTA4H, NDRG1, PGK1, COL6A1, ITGAV, and MB were verified by SRM in saliva; however, there is no consensus regarding either the localization of these proteins or whether they are classically or non-classically secreted. Evidence in the literature and in previous studies of our group has shown their detection either in extracellular milieu (Kawahara et al., 2015) or in extracellular vesicles (Winck et al., 2015). We have also described below information about the selected proteins using Uniprot, SignalP, SecretomeP, and THMN predictions and Vesiclepedia and Exocarta query results. We included a statement concerning these predictions in the revised version of the manuscript.

First, we present Supplemental Table 16 to show the details about CSTB, LTA4H, NDRG1, PGK1, COL6A1, ITGAV and MB in terms of the availability of information in the Uniprot database. We also performed some predictions using bioinformatics tools, including TMHMM 2.0, Signal P 4.1 (Petersen et al., 2011), SecretomeP 2.0 (Bendtsen et al., 2004), and MultiLoc2 (Blum et al., 2009), to predict localization. Supplemental Table 16 shows the studies that found these proteins in extracellular vesicles or in saliva. The only verified protein predicted to have a transmembrane region and to be non-classically secreted is ITGAV. The proteotypic peptides selected for verification localize in the extracellular milieu.

We can add that the protocol used to prepare the saliva samples is the “gold standard” protocol used in the literature (Bastos-Amador et al., 2012 and Hansen et al., 2014) and in our EV studies (Winck et al., 2015); however, it does not include ultracentrifugation, and thus it is possible that extracellular vesicles are present in the protein digest preparation. Moreover, because we included urea and sonication, it is possible that the extracellular vesicles were lysed in the process and that their content was released and susceptible to digestion.

b. It was interesting to see that some of the selected proteins validated in saliva (or were even detectable, considering they are not secreted proteins), although the clinical question (N0 vs. N+) is slightly different.

Response: We thank the reviewer for this pertinent observation. The main reason to evaluate the selected proteins in saliva was to determine whether non-invasive saliva collection could be used to reflect the relevant proteins in the tissue, in which case saliva could also be a very promising source of prognostic markers. As mentioned previously, saliva can host proteins from diverse origins.

The lymph node status was selected as the clinical parameter for two main reasons:

1) One reason is that lymph node metastasis is the most important factor for prognosis in oral cancer (Larsen et al., 2009; Ho et al., 2017), and its presence is highly associated with worse prognosis (Scully & Bagan, 2009).

2) Another reason is that a compromised lymph node shows significant clinical correlation using linear regression with ITGAV and COL6A1, and other proteins show association with clinical stage (MB) and disease-free survival (PGK1) (Table 1 main text), which are known to correlate with lymph node metastasis in oral cancer (Fan et al., 2011; Arduino et al., 2008; Greenberg et al., 2003).

i. Would combination of some of these peptides into a small signature improve the performance?

Response: Thank you for this suggestion. We evaluated the signature at the peptide and at protein levels. This evaluation is described in the revised manuscript on the Material and Methods session (page 38) and Results session (page 15) and shown in Supplemental Tables 31-39. Through strategies of machine learning, we evaluated the predictive individual and group importance of peptides/proteins to distinguish patients with lymph node metastasis (N+) and those without lymph node metastasis (N0). To identify the most relevant signatures, we measured the classification power for single peptides/proteins and for all possible combinations of peptides/proteins (Figures 7a-e). The figure added to the revised manuscript (Figure 7a) demonstrates the methodology used here to select the best signatures for oral cancer prognosis. The steps used to obtain and validate the signatures are divided into 4 subsequent parts (Parts 1- 4 of Figure 7a) that are detailed below and in the revised manuscript:

Part 1:

1- Split the 40 samples data into training set (80%) and independent test set (20%). The training set is used in parts 2 and 3 to perform filtering of variables, cross-validation and run the SES tool;

Part 2:

2- Filter proteins/peptides by Mann Whitney U test to obtain only variables with statistical difference between the groups (P-value<0.10);

3- Combine all proteins/peptides to create all possible signatures (size 1 to N, where N equals all peptides/proteins) to test in the next steps;

4- Perform a repeated cross-validation (100 rep. of stratified 10-fold cross-validation) employing Random Forests;

5- Perform SES method (Lagani et al., 2017) to seek for potential statistically equivalent signatures from step 4;

6- Create the list of potential signatures based on the scores of step 4; equivalent signatures (step 5) found by SES are added to the potential signatures list;

Part 3:

7-8- Compare the ROC curves and AUC of selected signatures, considering the cross-validated Random Forests from Part 2.

9- Compare the AUC values of each selected signature and their related accuracies. Check if the scores are consistent and define the best candidate markers list.

Part 4:

10- Validate the best models using the independent test set.

We found 9 **peptide** signatures through cross-validation with an accuracy $\geq 74.0\%$ and an SES of 4; we found 17 **protein** signatures through cross-validation with accuracy $> 68\%$ and an SES of 4. The peptide signatures found by SES had accuracy less than 62.0% and thus were not considered. Most of the proteins signatures identified by SES also had low accuracy ($<65.5\%$) and low AUC ($<62.2\%$); except for LTA4H, which had an accuracy of 76.2% and an AUC of 73.9%. The signature lists with highest performance values are described in Supplementary Table 36 and ordered by the highest AUC. As observed in the Supplementary Table 36, the combination of some peptides into a small signature can improve the performance in terms of AUC compared to the combination of proteins.

Comparing the signatures' cross-validation accuracy and ROC_AUC, we found that the signatures with highest accuracy do not have the highest AUC values. The most relevant signatures (S_i) from each level to discriminate N0 vs. N+ are identified in Table 3 (Panel 1). The highest accuracy found is 75.9%, which is 76.2% for the protein level (S_4) and 75.9% for the peptide level (S_1). Although they have similar accuracy, the highest AUC for the peptide level is considerably higher than that for the protein level—82.8% (S_2) vs. 73.9% (S_4). Only one protein signature was

chosen because the other signatures have lower performance with less than 71.3% accuracy compared to 76.2% for S_4 .

At the peptide level, we selected three signatures as options (S_1 , S_2 , and S_3). Among these three, we consider S_1 to be the most reasonable signature because it has an accuracy of nearly 76% and an AUC of 82.3%. Although S_2 has the best AUC, its accuracy is only 74.6%. S_1 has 1.3% higher accuracy than S_2 (74.6% vs. 75.9%) and only a small 0.5% decrease of AUC (82.8% vs. 82.3%); in contrast, the accuracy and AUC of S_3 are lower than those of S_1 . Considering that the class $N+$ has approximately twice as many samples as $N0$, S_1 has the most balanced prediction scores between both classes ($N0$, $N+$). Despite having lower sensitivity (75.6%), it has the highest specificity (76.6%) and the highest precision (86.2%). When only the AUC of individual signatures (size 1) are considered, the highest areas are from Pep12 (73.5%) and Pep8 (72.8%); Pep9 (71.0%) follows with approximately 2% difference.

The best candidates were further validated with the independent test set as shown in Supplementary Table 37. The peptide models had consistent prediction for the independent test, with accuracy and AUC as follows: 75.0% and 80.0% for S_1 , 75.0% and 86.7% for S_2 , and 75.0% and 86.7% for S_3 . Only S_4 showed decreased performance on the independent test with an AUC of 70.0% and accuracy of 62.5%, a difference of 13.7% from its cross-validation accuracy.

To compare the protein and peptide groups' performance, we compared the AUC values for Supplementary Table 36, as shown in Response Figure 3. Despite the higher accuracy of LTHA4, the peptide level has the best AUC and accuracy scores. This result could be explained by the class imbalance between $N0$ and $N+$. In support of this possibility, the specificity score is lower (62.3%) than the sensitivity score (83.4%). For this reason, it is important not only to evaluate the accuracy but also the probabilities through ROC curve analysis.

Response Figure 3. Comparison between protein and peptide signature. The highest ordered accuracy values for the potential peptide (in green) and for protein (in orange) signature are plotted in the graph.

Additionally, because decision trees and random forests are sensitive to class imbalance, we adopted an approach that consists of over-sampling the training subsets for each iteration of the 100 repetitions of stratified 10-fold cross-validation with the synthetic minority over-sampling technique (SMOTE) [Chawla et al., 2002, Han et al., 2005]. For the comparison using AUC, over-sampling provides more accurate results compared with under-sampling [Batista et al., 2004]. Only the 1,000 training subsets were over-sampled, and the associated 1,000 test subsets were composed of only

original samples. Thus, no synthetic sample was used to test the cross-validation models. When SMOTE was applied in the cross-validation, the training subsets had their size increased from ~27 to ~38 samples, with 50% N0 and 50% N+. The adopted method introduced changes at the top of the signature accuracy rankings. As shown in Table 3 (Panel 2), S4 has the highest AUC (74.0%) in the protein level and the second highest accuracy (69.3% vs. 70.4%), whereas S2 has the highest accuracy (77.2%) and the highest AUC (85.5%) among all protein and peptide signatures. For the peptide level, signature S1 has the third highest AUC (85.5%), the second highest accuracy (76.9%), and a difference of only 0.4% for AUC and 0.3% for accuracy compared to S2 (highest AUC and accuracy). In contrast, S3 had its performance decreased, moving from position 2 to 71 when ranking based on accuracy; S3 had 74.1% accuracy and 81.5% AUC. Although its performance is not greatly different from the imbalanced cross-validation, other signatures could have had their performance decreased by class imbalance; however, S1 and S2 are in the first positions of the rank. In addition to balancing the classes, the SMOTE introduced changes on the cross-validation training subsets (40% more samples than in the original training subsets). Therefore, the signatures S1 and S2 on the peptide level and S4 on the protein level are the best candidates for both types of cross-validation, using original imbalanced classes and using the balanced classes (over-sampled). Balancing the training subsets also increased the overall performance of the cross-validation schemes, as shown in Figure 7e. Thus, in the search for a signature that reflects the abundance of a combination of proteins or peptides in saliva for distinguishing different stages (N0 vs. N+) of OSCC patients, the best choice identified here is to consider the peptides in an exclusive manner without considering protein abundance.

5) How were the antibodies evaluated prior to IHC evaluation? It is quite common now that cell lines that express the proteins of interest (and kd-versions thereof) are pelleted and added to the analyses. How did the authors quality check these antibodies (the biggest bottleneck in the validation pipeline)?

Response: Thank you for the question and for the opportunity to describe in detail how we selected and evaluated the antibodies. To select the antibodies, we describe in detail the following four steps: (1) datasheet information analysis, (2) The Human Protein Atlas information analysis, (3) our Western blotting experiments, and (4) the standardization of IHC.

1. Datasheet and literature information: All antibodies were purchased from Sigma-Aldrich, and according to their datasheets, most were validated by Western blotting against recombinant proteins (CSTB, NDRG1, LTA4H, PGK1). Exceptions include anti-ITGAV (no bands were detected using a standard panel of samples, but it was validated by other techniques), anti-COL6A1 (validated by other techniques), and anti-MB (validated by other techniques).

2. Human Protein Atlas information: Each antibody was validated by the Human Protein Atlas Project (www.proteinatlas.org) in normal and tumor samples from different organs (please also see below).

- Antibody: anti-CSTB (HPA017380, Sigma)
Protein Atlas results and comments: general – selective cytoplasmic and nuclear expression in squamous epithelia and urothelium, head and neck cancer – cytoplasmic/membranous nuclear expression.
- Antibody: anti-LTA4H (HPA008399, Sigma)
Protein Atlas results and comments: general - nuclear and cytoplasmic expression in most tissues, head and neck cancer – strong nuclear expression.

- *Antibody: anti-NDRG1 (HPA006881, Sigma)*
Protein Atlas results and comments: general – cytoplasmic and nuclear expression, head and neck cancer – cytoplasmic/membranous nuclear expression.
- *Antibody: anti-PGK1 (SAB1300102, Sigma)*
PGK1 in human hepatocarcinoma tissue with Anti-PGK1 (Center S320) Antibody at 40 µg/mL. When this antibody was selected, it was presented in The Human Protein Atlas, but for some reason it was removed from the webpage, and we retrieved this information from Sigma. The image below is from The Human Protein Atlas at the time it was selected.
- *Antibody: anti-COL6A1 (HPA019142, Sigma)*
Protein Atlas results and comments: general – distinct positivity in connective tissue and extracellular matrix, head and neck cancer – cytoplasmic and membranous expression.
- *Antibody: anti-ITGAV (HPA004856, Sigma)*
Protein Atlas results and comments: general – cytoplasmic and membranous expression in most tissues, head and neck cancer – cytoplasmic and membranous expression in stroma and neoplastic cells.
- *Antibody: anti-MB (HPA003123, Sigma)*
Protein Atlas results and comments: general – highly specific cytoplasmic expression in striated muscle, head and neck cancer – no expression. In this case, we observed a faint staining of this antibody.

Response Figure 4. Comparison between the immunohistochemical staining of head and neck cancer from The Human Protein Atlas and labeling with the same antibodies in the samples from this study. Note the difference in contrast attributed to the type of hematoxylin used (eg. Harris, Mayer, Carrazi, etc.).

3. Evaluation of specificity and selectivity. We also performed some experiments to evaluate the specificity and selectivity of the antibodies for OSCC by Western blotting; the methods and results are shown in Supplementary Figure 8. We tested cells that could represent neoplastic islands (SCC-9 ATCC CRL-1629, a tongue cancer cell line; HSC3, a tongue cancer cell line) and stroma (BJ-5TA ATCC CRL-4001, a fibroblast immortalized cell line; CAF, primary cell of oral cancer-associated fibroblasts). We also tested other non-specific cell lines (SKMEL, a skin melanoma cell line; MCF7 ATCC HTB-22, a breast cancer cell line; and A549, an epithelial lung cancer cell line). We observed that most of the antibodies recognized the protein in all cell lines with the exception of anti-COL6A1, which recognized COL6A1 primarily in stroma cells. We also observed some “non-specific” bands for some antibodies, especially for anti-PGK1; these bands were not seen for LTA4H and COL6A1 (Response Table 2).

The PGK1 antibody was the only antibody that recognizes both higher and lower apparent molecular masses compared to the expected PGK1 molecular mass of 41.0 kDa. It is interesting that most of the bands are shared among the cell lines, with minor variations. We can speculate that several post-translational modifications, including ubiquitination, phosphorylation, acetylation, methylation, and others (PhosphoSitePlus, phosphor.Elm, Phosida) affect the mobility of PGK1 and associated apparent molecular weight. In addition, we can also speculate that the lower molecular masses result from cleavage by other proteinases (Merops). However, these are only hypotheses; a complete understanding is beyond the scope of this study.

Response Table 2: Summary of Western blot performed with extracts of seven different cell lines:

Gene name	Neoplastic island		Stroma		Other cell lines			Unspecific bands
	SCC-9	HSC3	BJ-5ta	CAF	SK-MEL-28	MCF7	A549	
CSTB	+	+	+	+	++	+	+	**
LTA4AH	-	+	+	-	-	-	++	-
NDRG1	-	+	+	++	++	-	+	*
PGK1	+	+	+	+	++	+	++	**
COL6A1	+	-	+	++	-	-	-	-
ITGAV	+	+	+	++	+	+	+	**
MB	-	+	-	-	++	+	++	**

Low (+) to high (++) intensity within the cell lines.

* one unspecific band; **more than one unspecific bands.

4- Standardization for IHC: Prior to the immunohistochemical reactions, we standardized the immunohistochemical tests in 5 independent samples of oral SCC using 3 different concentrations of antibodies and 2 distinct protocols of antigen retrieval to verify the label patterns; tests were performed with either citric acid or TRIS-EDTA (Response Table 3).

Response Table 3: Standardization of antigenic recovery for IHC:

Gene name	Concentrations of antibody tested in IHC	Concentrations of antibody with the best results in IHC	Antigenic recovery for IHC
CSTB	1:1000; 1:500	1:250	citric acid
LTA4AH	1:200; 1:150	1:100	citric acid
NDRG1	1:200; 1:150	1:100	citric acid

PGK1	1:200; 1:100	1:50	citric acid
COL6A1	1:3000; 1:1000	1:1000	citric acid
ITGAV	1:300; 1:75	1:300	citric acid
MB	1:500; 1:100	1:75	citric acid

For antigen recovery, the protocol with Tris-EDTA was tested, but the best results were obtained with citric acid.

In summary, we believe that these antibodies were appropriate for IHC analysis. It is important to highlight that the IHC results were subjected to a rigorous histology analysis by two pathologists, who observed that each antibody was specific to the area where the proteins were identified, with the exceptions of LTA4H and PGK1, which in addition to presenting peripheral staining in neoplastic cells were also detected in cells from tumor stroma, such as inflammatory cells. ITGAV staining was predominant in the tumor stroma with some staining observed in neoplastic cells.

In addition, we demonstrated that the IHC performed with the cases used for discovery proteomics revealed a similar protein expression pattern as the MS results (Supplementary Table 19), despite the superior dynamic range of MS relative to IHC as indicated by the reviewer. Finally, our results suggest that the antibodies are of sufficient quality for use in IHC analysis in combination with other analyses in the pipeline.

6) Page 23 (protein extraction): How did the authors deparaffinize their samples prior to proteomics. I couldn't find this in the protocols?

Response: *As requested, we have improved the description of the protocol used for deparaffinizing the samples prior to proteomics in the revised version of the manuscript.*

The following sentence was incorporated into the revised manuscript in Materials and Methods in the "Sample preparation and LMD" section: "...The other three sections of 10 µm thickness were prepared using specific membrane slides (PEN Arcturus® Membrane, Life Technologies, Foster City, CA, USA) for LMD; these slides were deparaffinized in xylol, hydrated with decreasing concentrations (100%, 90%, 70%, and 50%) of ethanol, washed in water, and stained with hematoxylin blue for 8 minutes before drying (Flores et al., 2016)."

7) Page 28 (SRM section): Peptides were selected based on SRMAtlas. Did the authors not make use of their own discovery data?

Response: *We would like to thank the reviewer for pointing this out. We have improved our description of peptide and transition selection below and in the revised manuscript in the Materials and Methods "Proteotypic peptides and transition selection" section. We also included Supplemental Table 25, which describes all the peptides (heavy and light counterparts) from the 7 target proteins monitored in saliva and the source used for method development (spectral libraries built up from our own DDA data and from human plasma DDA data present in the Peptide Atlas repository or SRMAtlas).*

8) Page 29 (salvia samples): Not sure adding protease inhibitors to a sample that was already denatured by 8M urea and then immediately processed for proteomics is the best choice. This might actually reduce digestion efficiency?

Response: *We would like to thank the reviewer for pointing out this technical issue. The protocol used for saliva sample preparation was previously modified by our group for shotgun and targeted studies (Winck et al., 2014, Winck et al., 2015, Kawahara et al., 2015 and 2016) based on a previously*

published protocol of the Gygi group (Villen and Gygi, 2008). We described the protocol and the concentrations in detail both below and in the revised version.

Response Table 4: Initial and final concentration of reagents used for saliva digestion:

Reagents	Initial concentration	Sample + buffer	Before adding trypsin
Urea	8M	6.1M	1.52M
Thiourea	2M	1.5M	0.37M
PMSF	1mM	0.76mM	0.19mM
EDTA	5mM	3.8mM	0.95mM
DTT	1mM	0.76mM	0.19mM
Tris HCl pH7.5	100mM	76.2mM	19mM
Complete Roche Inhibitor	1X	0.76X	0.19x

We agree that it may be excessive to include inhibitors in this stage of preparation, but considering that in the beginning of the protocol the concentration of urea is 6 M, some residual endogenous protease activity may persist. Previous studies have shown the recovery of protease activity even after treatment with 8 M urea both in snake venom (Paes Leme et al., 2009) and skin (Paes Leme et al., 2012). However, the effect of inhibitors on digestion was not an issue because before adding trypsin, all samples were diluted 1:5 in 50 mM ammonium bicarbonate.

To further demonstrate that the protease inhibitors did not affect the efficiency of digestion, we performed the same protocol with and without the inhibitors EDTA; 1 mM PMSF; and cComplete™, Mini, EDTA-free Protease Inhibitor Cocktail using SRM experiments with two technical replicates. We are presenting the results of saliva from a healthy subject due to limited patient samples. We are showing the results of two proteotypic peptides of CSTB and COL6A1; the other light (endogenous) peptides were not detected with a confident signal. Therefore, we confirmed that the digestion protocol with or without protease inhibitors did not interfere with the peptide yield (Response Figure 5).

(a)

CSTB K.SQVVAGTNYFIK.V [44, 55]

Light peptide

Heavy peptide

(b)

COL6A1

K.GLEQLLVGGSHLK.E [135, 147]

Light peptide

Heavy peptide

Response Figure 5. Light and Heavy peptide chromatograms for (a) COL6A1 and (b) CSTB proteotypic peptides in saliva samples digested with and without protease inhibitors and the respective bar plots showing the comparison between presence and absence of inhibitors. Two technical replicates were performed for each condition.

Minor comments:

1) Paper could benefit from some additional proof reading.

Response: We apologize for submitting a manuscript with some grammatical errors, despite that the original version of the manuscript was revised and edited by Springer Nature Author Services. A native English speaker has carefully revised the new version of the manuscript and we also revised the new version in the Springer Nature Author Services.

2) Some of the figures were of poor resolution. Some of the figures even had visible borders (Figure 2)?

Response: We improved all the figures in the manuscript.

References used to answer the reviewer 1:

1. Almangush, A. et al. A simple novel prognostic model for early stage oral tongue cancer.(2015). *Int. J. Oral Maxillofac. Surg.*, 44, 143–150.
2. Arduino PG. Clinical and histopathologic independent prognostic factors in oral squamous cell carcinoma: a retrospective study of 334 cases. (2008). *J Oral Maxillofac Surg.*, 66(8):1570-9.
3. Bastos-Amador P, Pérez-Cabezas B, Izquierdo-Useros N, Puertas MC, Martínez-Picado J, Pujol-Borrell R, Naranjo-Gómez M, Borràs FE. Capture of cell-derived microvesicles

- (exosomes and apoptotic bodies) by human plasmacytoid dendritic cells. (2012). *J. Leukoc. Biol.*, 91(5):751-8.
4. Batista GEAPA, Prati RC, and Monard MC. A study of the behavior of several methods for balancing machine learning training data. (2004). *SIGKDD Explor. Newsl.* 6, 1, 20-29.
 5. Bendtsen JD, Jensen LJ, Blom N, Von Heijne G, Brunak S. Feature-based prediction of non-classical and leaderless protein secretion. (2004). *Protein Eng Des Sel.*,17(4):349-56.
 6. Blum T, Briesemeister S, Kohlbacher O. MultiLoc2: integrating phylogeny and Gene Ontology terms improves subcellular protein localization prediction. (2009). *BMC Bioinformatics* 10:274.
 7. Carpenter GH. The secretion, components, and properties of saliva. (2013). *Annu Rev Food Sci Technol*, 4, pp. 267-276.
 8. Chang X, Xu X, Ma J, Xue X, Li Z, Deng P, Zhang S, Zhi Y, Chen J, Dai D. (2014). *NDRG1* expression is related to the progression and prognosis of gastric cancer patients through modulating proliferation, invasion and cell cycle of gastric cancer cells. *Mol Biol Rep.*, 41(9):6215-23.
 9. Chawla NV, Bowyer KW, Hall LO, Kegelmeyer WP. SMOTE: synthetic minority over-sampling technique. (2002). *Journal of artificial intelligence research*, 321-357.
 10. Chiang WF, Wu CC, Yu JS, Tsai CH, Liang KH, Chang YS, Wu M, and Yang WTO. Development of a Multiplexed Liquid Chromatography Multiple-Reaction-Monitoring Mass Spectrometry (LC-MRM/MS) Method for Evaluation of Salivary Proteins as Oral Cancer Biomarkers. (2017). *Molecular and Cellular Proteomics* 16.5, 799-811.
 11. Cox J, Hein MY, Luber CA, Paron I, Nagaraj N, Mann M. Accurate proteome-wide label-free quantification by delayed normalization and maximal peptide ratio extraction, termed MaxLFQ. (2014). *Mol Cell Proteomics*. 2014 Sep;13(9):2513-26.
 12. Craven RA, Cairns DA, Zougman A, Harnden P, Selby PJ, Banks RE. Proteomic analysis of formalin-fixed paraffin-embedded renal tissue samples by label-free MS: assessment of overall technical variability and the impact of block age. (2013). *Proteomics Clin Appl.* 7(3-4):273-82.
 13. da Silva, S. D., Hier, M., Mlynarek, A., Kowalski, L. P. & Alaoui-Jamali, M. A. Recurrent oral cancer: Current and emerging therapeutic approaches. (2012). *Front. Pharmacol.* 3: 149.
 14. Dawes C. Estimates, from salivary analyses, of the turnover time of the oral mucosal epithelium in humans and the number of bacteria in an edentulous mouth. (2003). *Arch Oral Biol.*, 48(5):329-36.
 15. Fan S, Tang QL, Lin YJ, Chen WL, Li JS, Huang ZQ, Yang ZH, Wang YY, Zhang DM, Wang HJ, Dias-Ribeiro E, Cai Q, Wang L. A review of clinical and histological parameters associated with contralateral neck metastases in oral squamous cell carcinoma. (2011). *International Journal of Oral Science*, 3: 180-191.
 16. Flores I, Kawahara R, Miguel MC, Granato DC, Domingues RR, Macedo CC, Carnielli CM, Yokoo S, Rodrigues PC, Monteiro BV, Oliveira CE, Salmon CR, Nociti FH Jr, Lopes MA, Santos-Silva A, Winck FV, Coletta RD, Paes Leme AF. *EEF1D* modulates proliferation and epithelial-to-mesenchymal transition in oral squamous cell carcinoma. (2016). *Clin Sci (Lond)*. 130(10):785-99.
 17. Gallien, S., Duriez, E. & Domon, B. Selected reaction monitoring applied to proteomics. *Journal of Mass Spectrometry* **46**, 298–312 (2011).
 18. Greenberg JS, El Naggar AK, Mo V, Roberts D, Myers JN. Disparity in Pathologic and Clinical Lymph Node Staging in Oral Tongue Carcinoma. (2003). *Cancer* 98 (3): 508-515.
 19. Gustafsson OJ, Arentz G, Hoffmann P. Proteomic developments in the analysis of formalin-fixed tissue. (2015). *Biochim Biophys Acta* 1854(6):559-80.
 20. Han H, Wen-Yuan W, Bing-Huan M. Borderline-SMOTE: a new over-sampling method in imbalanced data sets learning. (2005). *Advances in intelligent computing*, 878-887.
 21. Hansen HP, Engels HM, Dams M, Paes Leme AF, Pauletti BA, Simhadri VL, Dürkop H,

- Reiners KS, Barnert S, Engert A, Schubert R, Quondamatteo F, Hallek M, Pogge von Strandmann E. Protrusion-guided extracellular vesicles mediate CD30 trans-signalling in the microenvironment of Hodgkin's lymphoma. (2014). *J. Pathol.*, 232(4):405-14.
22. Ho AS, Kim S, Tighiouart M, Gudino C, Mita A, Scher KS, Laury A, Prasad R, Shiao SL, Van Eyk JE, Zumsteg ZS. Metastatic Lymph Node Burden and Survival in Oral Cavity Cancer. (2017). *J Clin Oncol.*, 35(31):3601-3609.
 23. Hubner, N. C., Cox J, Spettstoesser B, Badilla P, Poser I, Hyman A, Mann M et al. Quantitative proteomics combined with BAC TransgeneOmics reveals in vivo protein interactions. (2010). *J. Cell Biol.* 189, 739–754.
 24. Humphrey SP, Williamson RT. A review of saliva: Normal composition, flow, and function. (2001). *J Prosthet Dent.*, 85:162–169.
 25. Kawahara R, Meirelles GV, Heberle H, Domingues RR, Granato DC, Yokoo S, Canevarolo RR, Winck FV, Ribeiro ACP, Brandão TB, Filgueiras PR, Cruz KSP, Barbuto JA, Poppi RJ, Minghim R, Telles GP, Fonseca FP, Fox JW, Santos-Silva AR, Coletta RD, Sherman NE, Paes Leme AF. Integrative analysis to select cancer candidate biomarkers to targeted validation. (2015). *Oncotarget* 6, 41, 43635-43652.
 26. Vinzenz Lange, Paola Picotti, Bruno Domon, and Ruedi Aebersold. Selected reaction monitoring for quantitative proteomics: a tutorial. (2008). *Mol Syst Biol.* 4: 222.
 27. Vincenzo Lagani, Giorgos Athineou, Alessio Farcomeni, Michail Tsagris, Ioannis Tsamardinos. Feature Selection with the R Package MXM: Discovering Statistically Equivalent Feature Subsets. (2017). *Journal of statistical software*, 80 (7).
 28. Lagani. Feature Selection with the R Package MXM: Discovering Statistically Equivalent Feature Subsets. (2017). *Journal of statistical software.* 80 (7).
 29. Larsen SR, J. Johansen, JA, Krogh SA. The prognostic significance of histological features in oral squamous cell carcinoma. (2009). *J Oral Pathol Med.*, 38: 657–662.
 30. Lee YH and Wong DT. Saliva: An emerging biofluid for early detection of diseases. (2010). *Am J Dent.*, 22(4): 241–248.
 31. Lemaire R, Desmons A, Tabet JC, Day R, Salzet M, Fournier I. Direct analysis and MALDI imaging of formalin-fixed, paraffin-embedded tissue sections. (2007). *J Proteome Res.* 6(4):1295-305.
 32. Magdeldin S, Yamamoto T. Toward deciphering proteomes of formalin-fixed paraffin-embedded (FFPE) tissues.(2012). *Proteomics.*12(7):1045-58.
 33. Marakalala MJ, Raju RM, Sharma K, Zhang YJ, Eugenin EA, Prideaux B, Daudelin IB, Chen PY, Booty MG, Kim JH, Eum SY, Via LE, Behar SM, Barry CE 3rd, Mann M, Dartois V, Rubin EJ. Inflammatory signaling in human tuberculosis granulomas is spatially organized. (2016). *Nat Med.* 22(5):531-8.
 34. Metz B, Kersten GFA, Hoogerhout P, Brugghe HF, Timmermans HAM, Meiring H, Hove JT, Hennink WE, Crommelin DJA and Jiskoot W. J. Biol. Chem. 2004). Identification of Formaldehyde-induced Modifications in Proteins. (2004). *Journal of Biochemistry*, 279, 6235-6243.
 35. Nirmalan NJ, Harnden P, Selby PJ, Banks RE. Mining the archival formalin-fixed paraffin-embedded tissue proteome: opportunities and challenges. (2008). *Mol Biosyst.* 4(7):712-20.
 36. Noguti J, De Moura CF, De Jesus GP, Da Silva VH, Hossaka TA, Oshima CT, Ribeiro DA. Metastasis from oral cancer: an overview. (2012). *Cancer genomics & proteomics* 9(5):329-335.
 37. Paes Leme AF, Sherman NE, Smalley DM, Sizukusa LO, Oliveira AK, Menezes MC, Fox JW, Serrano SM. Hemorrhagic activity of HF3, a snake venom metalloproteinase: insights from the proteomic analysis of mouse skin and blood plasma. (2012). *J Proteome Res.* 11(1):279-91.

38. Paes Leme AF, Kitano ES, Furtado MF, Valente RH, Camargo AC, Ho PL, Fox JW, Serrano SM. Analysis of the subproteomes of proteinases and heparin-binding toxins of eight *Bothrops* venoms. (2009). *Proteomics* 9(3):733-45.
39. Petersen TN, Brunak S, von Heijne G, Nielsen H. SignalP 4.0: discriminating signal peptides from transmembrane regions. (2011). *Nat Methods* 8(10):785-6.
40. Räschle M, Smeenk G, Hansen RK, Temu T, Oka Y, Hein MY, Nagaraj N, Long DT, Walter JC, Hofmann K, Storchova Z, Cox J, Bekker-Jensen S, Mailand N, and Mann M. Proteomics reveals dynamic assembly of repair complexes during bypass of DNA cross-links. (2015). *Science* 348(6234): 1253671.
41. Robles MS, Cox J, Mann M. In-Vivo Quantitative Proteomics Reveals a Key Contribution of Post-Transcriptional Mechanisms to the Circadian Regulation of Liver Metabolism. (2014). *PLOS Genetics*, v.10, issue 1.
42. Ship JA, Fischer DJ. The relationship between dehydration and parotid salivary gland function in young and older healthy adults. (1997). *J Gerontol*, 52: 310–319.
43. Sprung, Jr. RW, Brock JWC, Tanksley JP, Li M, Washington MK, Slebos RJC and Liebler DC. Equivalence of Protein Inventories Obtained from Formalin-fixed Paraffin-embedded and Frozen Tissue in Multidimensional Liquid Chromatography-Tandem Mass Spectrometry Shotgun Proteomic Analysis. (2009). *Mol Cell Proteomics* 8(8): 1988–1998.
44. Storey J.D., Strong control, conservative point estimation and simultaneous conservative consistency of false discovery rates: a unified approach. (2004). *Journal of the Royal Statistical Society, B* 66, 187-205.
45. Takeshita, T. et al. Bacterial diversity in saliva and oral health-related conditions: the Hisayama Study. (2016). *Sci Rep* 6, 22164.
46. Tyanova S, Temu T, Sinitcyn P, Carlson A, Hein MY, Geiger T, Mann M, Cox J. The Perseus computational platform for comprehensive analysis of (prote)omics data. (2016). *Nat Methods* 13(9):731-40.
47. Winck FV, Ribeiro ACP, Domingues RR, Ling LY, Riaño-Pachón DM, Rivera C, Brandão TB, Gouvea AF, Santos-Silva AR, Coletta RD, Paes Leme AF. Insights into immune responses in oral cancer through proteomic analysis of saliva and salivary extracellular vesicles. (2015). *Scientific Reports* 5, 16305.
48. Wu M, Ouyang Y, Wang Z, Zhang R, Huang PH, Chen C, Li H, Li P, Quinn D, Dao M, Suresh S, Sadosky Y and Huang TJ. Isolation of exosomes from whole blood by integrating acoustics and microfluidics. (2017). *PNAS* 114 (40) 10584-10589
49. Xiao, H. & Wong, D. T. Proteomic analysis of microvesicles in human saliva by gel electrophoresis with liquid chromatography-mass spectrometry. (2012). *Anal Chim Acta* 723, 61–67.

Reviewer #2:

Remarks to the Author:

The manuscript " Discovery and targeted proteomics map of prognostic signature proteins in oral cancer" describes the use of proteomics to identify diagnostic/prognostic markers of oral cancer. They have used IHC of oral cancers and salivary SRM analysis of markers in N0 and N+ tumors to indicate that 4 markers of tumors and 3 markers of stroma are useful biomarkers of local recurrence and lymph node metastasis. While the techniques employed are standard and extensive, convincing data is not presented.

Major deficiencies

1. Title and abstract are vague. If the authors believe that cystatin B and other 6 markers are cancer biomarkers, the information should be spelled out in the title and in the abstract. They need to state that loss or reduced expression of cystatin B (CST B) as a diagnostic marker of local recurrence in the title. Similarly, authors need to make a clear statement regarding up or down regulation of the 7 markers in local recurrence and nodal metastasis in the abstract.

Response: We thank you for your opinion and suggestion. We partially agree with the reviewer that the title is vague because one of the most important results of this study is that we were able to identify a panel of prognostic signature candidates going from discovery to targeted phases and using several layers of verification. The word “map” indicates the meticulous analysis we performed in the discovery phase using laser microdissection of different areas of tumors, and the verification was performed for seven candidate proteins in the next phases using both IHC and SRM. We agree that the evidence that cystatin B (CSTB) is an independent prognostic marker of local recurrence is a very strong finding, but it is only part of the data. We also believe the results showing that the selected proteins in tissues could be significantly associated with lymph node metastasis and detected in saliva are original and promising.

We improved our description of the correlation with clinical data for the other proteins from Discovery, IHC, and SRM data, and finally, we also included a new analysis of the SRM to indicate the best protein and peptide signatures. All the changes that resulted from the suggestions of the three reviewers clearly improved the manuscript and clarified the reasons that the title needs to reflect the prognostic signature and not only one potential candidate.

We have also included information regarding the expression of the 7 selected candidates in the abstract as suggested.

2. It is difficult to read the manuscript with redundant and convoluted sentences throughout the manuscript. For example, lines 249 to 258 talks about the utility of CST B and PGK1 in recurrence and survival. The results seem to suggest that down regulation of CST B and up regulation of PGK1 have a role in local recurrence which is not stated clearly. Lines 251/252 indicating significant 5-year disease free survival is linked with previous line making it difficult to know the real association, up or down regulation. Similarly, lines 257/258 on 3-year DFS is not clear on the reference to lower expression of CSTB.

Response: Thank you for this important observation. We have revised the entire manuscript, and this particular section has been rewritten.

3. Introduction, results and discussion need to be shortened by deleting repetitive and vague sentences and making clear statements on the up or down regulation of identified bio markers.

Response: We thank the reviewer for the suggestions. We have changed the description in the results and discussion sections in an attempt to make this clearer for the reader.

4. IHC for CST B and MB (figure 4) are of poor quality making it difficult to visualize the staining. Better staining and representative higher magnification figures are required to convince hybridization of these markers to tumor cells or stroma respectively.

Response: As requested, we have included representative immunohistochemical images for CSTB and MB with better quality and high power (40x). Thus, Figure 4 was replaced in the revised version, and the figures with lower magnification are included in Supplemental Figure 7.

5. Authors have not provided justification for the selection of these markers over differential expression of markers with higher significance in figures 3A and 3C. Also, the relevance of the identified markers to cancer development is not stated for all 7 markers.

Response: We would like to thank the reviewer for highlighting this important step of the study. The selection of targeted proteins started from the discovery phase and followed steps that we justified in the revised version in a separate section of Results and also in the Supplementary Table 16: (1) only proteins with different protein abundances in between the ITF and the inner tumor in the discovery phase (Student's *t*-test, P -value <0.05); (2) only proteins that present a significant association with clinical characteristics of patients (Linear regression, P -value <0.05 , $R < -0.7$ or $0.7 < R$ and $R^2 > 0.4$) (Table 1); (3) only proteins with positive staining of squamous cell carcinoma in HNSCC in The Human Protein Atlas (<https://www.proteinatlas.org/>); and (4) only proteins not cited or cited only in limited studies related to oral cancer (Supplementary Table 16), with exception of COL1A2 and ACTR2 that are planned to be evaluated in future study. In the Discussion session of the revised version, we also included some relevant information regarding the involvement of the seven identified markers in cancer development, extending the discussion for the 6 verified proteins in saliva.

6. There seems to be only one N0 sample with higher expression for COL6A1 expression in figures 6C-D. To some extent it is also true for ITGAV making it difficult to accept these two as significant markers of N0 over N+ tumors.

Response: We thank the reviewer for these observations. To address this comment, please consider the two points below:

- 1) We improved the graph visualizations in Figure 6 of the revised version to provide clear evidence of the differences between N0 and N+. Fortunately, we were also able to include samples from 4 new N0 patients and 3 new N+ patients that we collected and analyzed during manuscript revision.
- 2) We agree that the data have ordinary variations expected in human samples, but we feel that this variation reflects the intrinsic differences among patients with OSCC, a type of cancer known to present distinct behavior even at a similar developmental stage. This is one of the reasons that the most used clinical TNM system is unable to accurately predict patient outcome (Almangush et al. 2015; da Silva et al., 2012).

In addition to literature data showing the variation in OSCC, we have also observed this variation in our previous studies using saliva analysis in discovery proteomics (Winck et al., Scientific Reports, 2015), targeted proteomics by pseudo-SRM (Kawahara et al., Oncotarget, 2015), and SRM (Kawahara et al., Proteomics, 2016).

In our first shotgun study analyzing saliva, Winck et al. (2015) evaluated the protein abundance in saliva in the presence or absence of active tumor lesions for OSCC patients. Among the 507 proteins identified, 181 remained after applying filters (removal of contaminants, reverse database, only identified by site, and at least 5 valid values per group),

and 11 proteins were differentially expressed between the “active lesion” ($n = 12$) and “no lesion” (removed by surgery, $n = 9$) groups (Response Figure 1A). We also noticed that the number of missing values varies among the samples (Response Figure 1B). Here, we also performed an unsupervised PCA, which indicated a great heterogeneity of proteins in saliva (Response Figure 1C). In other words, we also observed the inter-patient variability in previous studies when using shotgun proteomics in our group.

Response Figure 1. Inter-patient variation of saliva samples from patients with oral cancer analyzed by shotgun proteomics (Winck et al., 2015). (a) Number of identified proteins after applying the filters varies among samples extracted from saliva. (b) Number of missing values after applying the filters varies among protein samples extracted from saliva. (c) The unsupervised PCA analysis demonstrates that proteomic data from saliva of oral cancer patients with and without active lesion do not separate well in two distinct classes. All these data indicate that such degree of heterogeneity of patients occurs and reflects the OSCC behavior.

In addition, in other studies we used pseudo-PRM and SRM (Kawahara et al., 2015 and 2016, respectively). We have plotted these results using plots (Response Figure 2) similar to those we used

in the original and revised versions of the manuscript. In these studies, we have analyzed saliva samples from patients with or without active lesions, and we also observed inter-patient variability within the groups.

Response Figure 2 (modified from our previous study Kawahara et al., 2015 and 2016). CFB and C3 peptides showed higher normalized intensities in “Lesion” saliva samples than in “No lesion” saliva samples. (a-b) Pseudo-PRM analytical approach for peptides of C3 (precursor m/z 631.05, +3; 735.89, +2) and CFB (precursor m/z 638.33, +2; 939.13, +3) normalized with 5 fmol/ μ l of angiotensin (m/z 432.89, +3) as an internal reference peptide. These data represent two technical replicates of saliva samples from patients who undergone surgical resection of OSCC (named no lesion, $n = 7$) and saliva samples from patients with active OSCC lesion without any treatment (named lesion, $n = 10$) (Student’s t -test, P -value <0.05). The normalization to the internal reference peptide was performed for each run. (c) Box plot representation of Light/Heavy ratio peptide expression levels within saliva samples by SRM analytical approach (shown as \log_2 ratios of endogenous “Light” divided by spike-in standard “Heavy” peak).

7. There are two previous reports by the authors on the identification of tumor specific markers in cancer associated fibroblasts and saliva relating to oral tumors. It is surprising authors have not analyzed these markers in IHC to show their clinical relevance with respect to markers identified in the present investigation.

Response: We thank the reviewer for pointing this out. Although several markers for oral cancer have been reported in our studies and others, none have shown diagnostic or prognostic value with high specificity and accuracy in oral tumors, and none are used in clinical practice. A recent systematic review by our group (Rivera et al., *Oral Oncology*, 2017) emphasizes this point and shows that there are several individual studies claiming to have identified novel biomarkers, but interestingly none of these biomarkers are in the clinic yet due to a lack of follow-up studies. Thus, the present study is the first one in the field of Neck and Head Cancer that has performed several layers of verification using rigorous discovery and targeted phases, and this report may encourage new future directions for assisting in the prognosis of OSCC patients in clinical routine. We plan in the near future to combine all the candidates from our previous discovery shotgun and targeted studies, in addition to the potential targets selected from the systematic review, and evaluate them in additional cohorts; this evaluation will most likely involve saliva collection, considering all the advantages of this non-invasive method.

8. Line 433 to 435 points to a mean recurrence period of 7.5 months in other studies and a mean recurrence of 12 months with lower CST B expression in the present investigation. This contradiction should be clarified instead of a vague statement in line 436 that CST B expression is associated with local relapse of OSCC.

Response: We have altered this ambiguous sentence in the revised manuscript. We intended to note that recurrence in patients with lower expression levels of CSTB in the ITF occurred within a mean time of 12 months after initial treatment, whereas the literature reports a mean of 7.5 months.

References used to respond to the reviewer 2:

1. Almangush, A. et al. A simple novel prognostic model for early stage oral tongue cancer. (2015). *Int. J. Oral Maxillofac. Surg.*, 44, 143–150.
2. Brandwein-Gensler, M. et al. Validation of the Histologic Risk Model in a New Cohort of Patients with Head and Neck Squamous Cell Carcinoma. (2010). *Am. J. Surg. Pathol.* 34, 1.
3. Chung, L.-C. et al. L-Mimosine blocks cell proliferation via upregulation of B-cell translocation gene 2 and N-myc downstream regulated gene 1 in prostate carcinoma cells. (2012). *AJP Cell Physiol.* 302, C676–C685.
4. Costa, L. C. M. C. et al. Expression of epithelial-mesenchymal transition markers at the invasive front of oral squamous cell carcinoma. (2015). *J. Appl. Oral Sci.* 23, 169–178.
5. da Silva, S. D., Hier, M., Mlynarek, A., Kowalski, L. P. & Alaoui-Jamali, M. A. Recurrent oral cancer: Current and emerging therapeutic approaches. (2012). *Front. Pharmacol.*, 30;3:149.
6. de Carvalho, H.F., Taboga, S.R. & Vilamaior, P.S. Collagen type VI is a component of the extracellular matrix microfibril network of the prostatic stroma. (1997). *Tissue & cell*, 29(2), pp.163–70.
7. Hwang, T.-L., Liang, Y., Chien, K.-Y. & Yu, J.-S. Overexpression and elevated serum levels of phosphoglycerate kinase 1 in pancreatic ductal adenocarcinoma. (2006). *Proteomics* 6, 2259–72 (2006).
- 7- Jensen, D. et al. Molecular profiling of tumour budding implicates TGFβ-mediated epithelial-mesenchymal transition as a therapeutic target in oral squamous cell carcinoma. (2015). *J. Pathol.* 236, 505–516.
- 8- Kawahara R, Meirelles GV, Heberle H, Domingues RR, Granato DC, Yokoo S, Canevarolo RR, Winck FV, Ribeiro ACP, Brandão TB, Filgueiras PR, Cruz KSP, Barbuto JA, Poppi RJ, Minghim R, Telles GP, Fonseca FP, Fox JW, Santos-Silva AR, Coletta RD, Sherman NE, Paes

- Leme AF. Integrative analysis to select cancer candidate biomarkers to targeted validation. (2015). *Oncotarget* 6, 41, 43635-43652.
- 9- Kawahara R, Bollinger JG, Rivera C, Ribeiro ACP, Brandão TB, Paes Leme AF, MacCoss MJ. A targeted proteomics strategy for the measurement of oral cancer candidate biomarkers in human saliva. (2016) *Proteomics*, 16 (1):159-173.
 8. Maruyama, Y. et al. Tumor growth suppression in pancreatic cancer by a putative metastasis suppressor gene *Cap43/NDRG1/Drg-1* through modulation of angiogenesis.(2006). *Cancer Res.* 66, 6233–6242.
 9. Oi N., Yamamoto H, Langfald A, Bai R, Lee MH, Bode AM, Dong Z. *LTA4H* regulates cell cycle and skin carcinogenesis. (2017). *Carcinogenesis*, 38(7):728-737.
 10. Rivera C , Oliveira AK, Costa R.A.P, De Rossi T, Paes Leme AF. Prognostic biomarkers in oral squamous cell carcinoma: A systematic review. (2017). *Oral Oncol.*, 72:38-47.
 11. Sabatelli, P. et al., 2001. Collagen VI deficiency affects the organization of fibronectin in the extracellular matrix of cultured fibroblasts. *Matrix Biology*, 20(7), pp.475–486.
 12. Sharma, M., Sah, P., Sharma, S. & Radhakrishnan, R. Molecular changes in invasive front of oral cancer. (2013). *J. Oral Maxillofac. Pathol.* 17, 240.
 13. Winck FV, Ribeiro ACP, Domingues RR, Ling LY, Riaño-Pachón Diego Mauricio, Rivera C, Brandão TB, Gouvea AF, Santos-Silva AR, Ricardo D. Coletta RD, Paes Leme AF. Insights into immune responses in oral cancer through proteomic analysis of saliva and salivary extracellular vesicles. (2015). *Scientific Reports* 5, 16305.
 14. Yang, X. et al. *N-myc Downstream-regulated Gene 1* Promotes Oxaliplatin-triggered Apoptosis in Colorectal Cancer Cells via Enhancing the Ubiquitination of *Bcl-2*. (2017). *Oncotarget* 8(29):47709-47724.
 15. Zieker, D. et al. Phosphoglycerate kinase 1 a promoting enzyme for peritoneal dissemination in gastric cancer. (2010). *Int. J. Cancer* 126, 1513–1520.

Reviewer #3

Remarks to the Author:

Summary

This work describes a project aiming for discovery and validation of prognostic biomarkers in OSCC. The discovery part consist of LCM dissection of tumor tissue where neoplastic and stromal regions are analysed seperately whith respect also to regional location (inner vs ITF). Targeted validation of candidate biomarkers was then performed using both IHC and SRM.

Overall

The search for biomarkers to detect cancer, to prognosticate outcome or to predict response to therapy is an important but difficult part of cancer research. The approach here used to identify prognostic biomarkers is sound and should have the potential to identify candidate biomarkers. Especially the multi-layered analysis used, with a discovery in the tumor

itself followed by an evaluation of biomarker presence in an easily accessible body-fluid is likely to increase the chance of finding good biomarker candidates.

Indeed, the analysis produce a few candidate biomarkers with potential use for prognostication in OSCC. Specifically CSTB is claimed to be an independent marker for local recurrence in tissue, and five proteins are claimed to be specifically associated with lymph node metastasis. Saliva biomarkers of tumor spread should undoubtedly be of interest to others in the community.

Unfortunately, I am not convinced by the performance of the analysis and the interpretation of the results as detailed below. It is not fully proven that the suggested biomarkers have the prognostic value that is claimed in the manuscript. For these reasons it is not clear that the work is suitable for Nature Communications as presented.

Response: We improved the analysis and interpretation as suggested by all reviewers, thus strengthening the evidence that these candidates could be useful candidates as prognostic signatures.

In general, the manuscript would also benefit from more thorough proof-reading.

Response: We apologize for submitting a manuscript with some grammatical errors, despite that the original version of the manuscript was revised and edited by Springer Nature Author Services. A native English speaker has carefully revised the new version of the manuscript and we also revised the new version in the Springer Nature Author Services.

In many cases the readability was hampered by misplaced modifiers or unclear references. Also, in some cases numbers are not matching between different parts of the manuscript. As an example, the text states that four proteins are higher in ITF stroma and one protein is higher in inner stroma (line 195-200). The text also refers to Table 1, but in table 1 only two proteins are higher in ITF stroma (COL6A1 and MB) while three proteins are higher in inner stroma. Several more similar examples are present and collectively the reader has to spend a lot of time trying to understand what the authors mean.

Response: Thank you for the corrections. We improved the description of the results throughout the revised manuscript and specifically improved the description in the clinicopathological section for the samples evaluated in the discovery phase.

Discovery Proteomics

The discovery part of this work was performed using label free quantification (LFQ) of proteins across 120 different samples. A major problem with LFQ is that the identification overlap between samples often limits the number of proteins that can be compared across samples. In the current work this is illustrated by the total number of identified proteins (e.g. 2049 in neoplastic cells) compared to the number of proteins used in the statistical analysis after

filtering away proteins with low overlap between samples (799). This is also illustrated by the fact that six samples from the stroma set were discarded due to low number of identified proteins in three of the samples. Such filtering is commonly used in label free experiments. More problematic is that it is unclear how the decision was made to exclude the six stroma samples, while still including neoplasm samples that are also low in number of identified proteins (Supplementary Fig 2 and Supplementary Table 5, e.g. neoplasm 11_I, 3_F and 9_F are missing 87, 66 and 62% respectively of the 799 proteins evaluated).

Response: We would like to thank the reviewer for highlighting the difficulty of quantitation in label-free samples. Although we agree that the LFQ algorithm can influence the number of proteins identified, the MaxQuant package was chosen to analyze these data in the discovery phase because it can quantify label-free protein samples and has been applied in several biological projects in the proteomics community.

In addition, we also agree that these data are not easy to analyze, and some decisions were made during analysis to produce the most reliable and high-confidence results in the discovery phase. To analyze these types of samples, we had to balance limitations in terms of the quantities of extracted proteins and peptides, take into account the quality of extraction, implement a high-rigor analysis procedure during the pathological inspection of the tissue areas and during the microdissection of the six areas from FFPE samples, and avoid the exclusion of samples to avoid the loss of valuable information regarding clinical characteristics and outcomes. Unfortunately, we did not have enough samples to repeat the laser microdissection, protein extraction, and digestion.

Therefore, we would like to justify the decisions made regarding data analysis for both stromal and neoplastic samples, as described below:

Regarding stroma samples:

The decision to exclude P11F, P19F, P19I, and P20I tumor stroma samples was made for the following reasons:

- We observed low-intensity spectra, resulting in a low number of protein identifications. Only 106 proteins for the P11F sample, 107 proteins for P19F, 246 proteins for P19I, and 289 proteins for P20I were identified after applying the filters (reverse, “only identified by site”, valid values) (Response Figure 1A-B). We also observed, as mentioned in the original version of the manuscript, through histogram analysis (Supplementary Figure 4), a higher number of missing values in these samples (Response Figure 1C).*
- In addition, we observed correlation values (Pearson correlation) ranging from 0.455 to 0.762 for P11F, 0.277 to 0.775 for P19F, 0.443 to 0.775 for P19I, and 0.524 to 0.825 for P20I. These samples exhibited the highest numbers of missing values before and after the application of all filters, as shown in the figure below (Response Figure 1A-C).*
- It is important to state that although the internal quality control used in our group showed that the trypsin autolysis peaks presented the expected retention time across most of the samples (m/z 523.2855, +2, retention time 22.8 min; m/z 421.7584, +2, retention time 28.7 min; m/z 737.7062, +3, retention time 83.6 min), we found that samples 20 and 11 presented higher shifts in terms of predicted retention time. The removal of these samples from the analysis improved the coefficient of variation for the trypsin peptides (Response Figure 1D).*

D

Trypsin peptides	Mean (min)	SD	CV % Before removing P11	CV % After removing P11F	CV % After removing P11I
Peptide 1	21.76	2.04	9.40	7.46	9.48
Peptide 2	17.30	1.20	6.93	6.2	6.91
Peptide 3	79.50	2.24	2.82	2.73	2.77

Response Figure 1: Quality control of the DDA runs (Tumor stromal samples). The bar plots show the total number of identified and quantified proteins (A) and box plots (B) show the LFQ intensity values found in each tumor stromal sample. The number of missing values in each stroma sample is presented as bar plots (C). All the graphs were performed after all filters (reverse, ‘only identified by site’ and at least 8 valid values per group) were applied. (D) Trypsin autolysis peaks presented the expected retention time across most of the samples (m/z 523.2855, +2, retention time 17.30 min; m/z 421.7584, +2, retention time 21.76 min; m/z 737.7062, +3, retention time 79.50 min), except for P11E and P20F (where E corresponds to the stroma from the invasive tumor front – ITF, and F to the stroma from inner tumor).

Regarding neoplastic islands samples:

Regarding the previous figure 2b (now figure 2c), sample P11_I.

- P11_I is composed of samples from the inner tumor area of (i) small neoplastic islands and (ii) large neoplastic islands and contain 32 and 85 quantified proteins, respectively. The paired P11_F is composed of samples from the invasive tumor front (ITF) area of (i) small neoplastic islands and (ii) large neoplastic islands with 93 and 619 quantified proteins, respectively (excluding the reverse database and identified “only” by site entries).
- Thus, excluding the P11 samples, we discarded four samples, including samples that had an acceptable number of identified and quantified proteins.
- Similarly, P3_I is composed of samples from the inner tumor area of (i) small neoplastic islands and (ii) large neoplastic islands, with 253 and 961 quantified proteins, respectively. The paired P3_F is composed of samples from the ITF area of (i) small neoplastic islands and (ii) large neoplastic islands, with 175 and 98 quantified proteins, respectively (excluding the reverse database and identified “only” by site entries). Additionally, P9_I is composed of samples from the inner tumor area of (i) small neoplastic islands and (ii) large neoplastic islands, with 54 and 792 quantified proteins, respectively. The paired P9_F is composed of samples from the ITF area of (i) small neoplastic islands and (ii) large neoplastic islands, with 75 and 297 quantified proteins, respectively (excluding the reverse database and identified “only” by site entries).
- Removal of these samples from the analysis did not improve the coefficient of variation (Response Figure 2D).
- In addition, considering the internal quality control used in our group, the trypsin autolysis peaks presented the expected retention times across most of the samples (m/z 523.2855, +2, retention time 22.8 min; m/z 421.7584, +2, retention time 28.7 min; m/z 737.7062, +3, retention time 83.6 min) (Response Figure 2).

D

Trypsin peptides	Mean (min)	SD	CV % Before removing P11	CV % After removing P11A	CV % After removing P11B	CV % After removing P11C	CV % After removing P11D
Peptide 1	22.67	2.38	10.51	10.57	10.57	10.51	10.41
Peptide 2	17.61	1.37	7.78	7.80	7.81	7.77	7.79
Peptide 3	80.59	3.88	4.82	4.83	4.84	4.83	4.83

Response Figure 2: Quality control of the DDA runs (neoplastic island samples). The bar plots show the total number of identified and quantified proteins (A) and box plots (B) show the LFQ intensity values found in each neoplastic island samples. The number of missing values in each neoplastic island sample is presented as bar plots (C). All the graphs were performed after all filters (reverse, ‘only identified by site’ and at least 10 valid values) were applied. (D) Trypsin autolysis peaks presented the expected retention time across most of the samples (m/z 523.2855, +2, retention time 17.30 min; m/z 421.7584, +2, retention time 21.76 min; m/z 737.7062, +3, retention time 79.50 min), including trypsin peptides of P_11 samples (visualized inside red boxes), except for P1, P2 and P3 samples.

This becomes even more problematic since the authors use imputation to replace missing values with random numerical values corresponding to a low abundance measurement. A missing value in MS based proteomics does not necessarily mean that the protein is missing or very low, just that the MS-instrument did not measure it. In addition, there is a skewness in the distribution of missing values between the ITF and inner neoplasm samples so that ITF samples have more missing values than inner samples in general (number of NaNs per sample in Supplementary Table 5, median 128 for ITF vs 83 for inner samples). Such imbalance will affect the quantitative and statistical analysis, and could potentially have caused the assymetrical distribution of values in the volcano plot in Figure 3a.

Response: Thank you for the valuable observations. The missing values for the LFQ intensities were imputed with random numbers from a normal distribution using low abundance values close to the noise level (imputation width=0.3, shift=1.8), which is similar to methods used in other manuscripts (Hubner, N. C. et al., 2010; Marakalala, M. J. et al., 2016). It is worth considering that the absence of LFQ intensity values does not indicate the absence of a protein; it may have intensity values or even spectral counts for the protein, and therefore we also agree with the reviewer that a “missing value does not necessarily mean that the protein is missing or in very low abundance, but just that the MS-instrument did not measure it”. In addition, the imputation also helps avoid the apparent exclusivity of the proteins in each sample or group.

Missing values are a common problem with large-scale data and especially with challenging FFPE samples. However, to be able to apply statistical methods to correlate the clinical pathological characteristics of patients, we could not handle missing information and imputation before the analysis was required. Thus, we used Perseus software to select random values from a distribution meant to simulate protein abundance below the detection limit (Tynova et al., 2016; Wiśniewski et al., 2012).

The detailed workflow for discovery data analysis is as follows:

- 1) A search was performed using MaxQuant at 1% FDR for peptide and protein levels;
- 2) Using Perseus software, we removed reverse entries and “only by site” entries;
- 3) LFQ intensity values were Log₂ normalized;
- 4) Categorical annotation of the groups was performed;
- 5) Correlation analysis using Pearson correlation within Perseus software was performed;
- 6) A filter for valid values was applied (10 valid values in at least one group for neoplastic island samples; 8 valid values in at least one group for stromal samples);
- 7) Missing values were replaced by imputation with random numbers from a normal distribution using low-abundance values close to the noise level (imputation width=0.3, shift=1.8);

- 8) *Statistical analysis using Student's t-test was performed;*
- 9) *Linear correlation analysis with clinicopathological parameters was performed.*

To provide the reviewer with information regarding the effects of applying the filter for valid values and associated imputation, we also performed the two tests without filtering and with/without imputation and we observed the following:

Test 1: Without applying the filtering of valid values:

Tumor Stromal samples:

- ***Without applying the filter of valid values followed by the replacement of missing values (imputation)***, we verified 115 significant proteins. However, THBS2 was not in the list, whereas the other proteins (COL6A1, FSCN1, ITGAV, and MB) were significant (P -value <0.05 , Student's t -test);
- ***Without applying the filter of valid values and not replacing the missing values (imputation)***, we found 76 significant proteins, including COL6A1, ITGAV, FSCN1, MB, and THBS2.

Neoplastic island samples:

- ***Without applying the filter of valid values followed by the replacement of missing values (imputation)***, we verified 81 significant proteins. However, LTA4H was not in the list, whereas the other proteins (CSTB, NDRG1, and PGK1) were significant (P -value <0.05 , Student's t -test);
- ***Without applying the filter of valid values and not replacing the missing values (imputation)***, we found 74 significant proteins. However, LTA4H was not in the list, whereas the other proteins (CSTB, NDRG1, and PGK1) were significant (P -value <0.05 , Student's t -test).

Test 2: applying the filter of valid values:

Tumor Stromal samples (8 valid values):

- ***With imputation*** (the analysis we showed in the main text), we had 101 differentially abundant proteins;
- ***Without imputation***, this value decreased to 71 differentially abundant proteins. However, there was no effect on the statistical significance of the selected tumor stromal proteins, COL6A1, ITGAV, MB, FSCN1, and THBS2.

Neoplastic island samples (10 valid values):

- ***With imputation*** (the analysis we showed in the main text), we had 32 differentially abundant proteins;
- ***Without imputation***, this value increased to 64 differentially abundant proteins. However, in this analysis, only LTA4H was not considered significant (P -value <0.05 , Student's t -test).

In the case of “skewness in the distribution of missing values”:

- We created volcano plots with and without the imputation of missing values, both after the filter of valid values (10 valid values for neoplastic islands, 8 valid values for tumor stroma), and we did not observe evident differences as a result of imputation (Response Figure 3 A-B).
- We also performed (Response Figure 3 C-D) an analysis of missing values for the ITF and inner neoplasm, and we observed that the number of missing values between the tumor front and the inner portion of both stroma and neoplastic islands is not statistically significant (Wilcoxon rank sum test with continuity correction, P -value = 0.8497 for stroma and p -0.1368 for neoplastic island), but the P -value is lower in the neoplastic island than in stroma, which can result in the asymmetrical distribution of the volcano plot.

Response Figure 3. Differentially regulated proteins between the ITF and the inner tumor of neoplastic islands (A) and tumor stroma (B) without imputation of missing values, both after filtering for valid values, as determined by plotting Student’s t -test (P -value < 0.05) significance versus the \log_2 ratio of the LFQ intensity (volcano plot); significant values are indicated by red dots. The numbers of missing values in each neoplastic island sample (C) and in tumor stroma (D) are plotted as bar graphs.

The statistical analysis of proteins that are differentially expressed between ITF and inner samples in neoplasm or stroma is based on a paired t-tests for the 799 and 704 overlapping proteins that passed the filtering criteria. From what I understand of the materials and methods, no correction for multiple testing has been done and consequently many of the 32 and 101 proteins identified as differentially expressed will be false positives. With a p-value threshold of 0.05 we would expect to find $799 \times 0.05 = 40$ proteins to be significant just by chance.

Response: Statistical analysis for the discovery phase was not corrected for multiple testing. As a requirement in the proteomics field, we applied Benjamini-Hochberg using the Perseus software for multiple testing corrections; however, this correction resulted in no proteins with significant difference between the compared groups (front and inner tumor).

We also tested the multiple hypothesis corrections using Storey's test (Storey et al., 2004). The q-values were calculated considering a maximum FDR of 5%, which resulted in no significant proteins from the neoplastic islands and 13 significant proteins from the stroma dataset (Response Table 1).

Response Table 1: Storey's test for multiple test correction of tumor stroma dataset.

P-value (Student's t-test)	q-value	Gene name
0.000483	0.040439	ITGAV
0.001107	0.049698	LDHB
0.000573	0.041084	ANXA2
0.000446	0.040439	RNASE2
0.001144	0.049698	LCP1
0.00102	0.049698	FLNA
0.000272	0.034091	TGM2
5.73E-05	0.01076	COL14A1
0.001247	0.049698	PLEC
6.43E-05	0.01076	PTGIS
0.000926	0.049698	FSCN1
4.84E-05	0.01076	RCN3
0.001287	0.049698	MXRA5

Considering that the list of statistically significant proteins using Student's t-test is not the endpoint of this study, we first chose to filter out proteins that did not correlate with the clinicopathological parameters instead of adjusting the P-values. Additionally, we performed several layers of downstream analysis, including IHC verification, and targeted proteomics to verify the proteins. In the case of the SRM data, in the revised version, the statistical analyses were corrected for multiple testing (Figure 6A, main text), reinforcing the verification of the six candidates (CSTB, ITGAV, NDRG1, LTA4H, COL6A1, PGK1) obtained from DDA data.

IHC validation.

The selection of proteins for the IHC validation was based on the association between the candidate biomarkers (32 and 101) and clinicopathological parameters. To me this choice was not very transparent, and in some cases I did not find the underlying data. As an example, Table S1 holds the clinicopathological data for the discovery cohort, but it does not contain information about treatment or second primary tumor. Still, Table 1 indicate that there is a statistically significant association between e.g. treatment and CSTB or PGK1 and second primary tumor.

Response: Thank you for correction. As requested, the missing information was added to Supplementary Table 1 and 17. We included a new section in Results to explain the prioritization of proteins: (1) only proteins with different protein abundances in between the ITF and the inner tumor in the discovery phase (Student's t-test, P -value <0.05); (2) only proteins that present a significant association with clinical characteristics of patients (Linear regression, P -value <0.05 , $R<-0.7$ or $0.7<R$ and $R^2>0.4$) (Table 1); (3) only proteins with positive staining of squamous cell carcinoma in HNSCC in The Human Protein Atlas (<https://www.proteinatlas.org/>); and (4) only proteins not cited or cited only in limited studies related to oral cancer (Supplementary Table 16), with exception of COL1A2 and ACTR2 that are planned to be evaluated in future study.

The authors state that the IHC analysis partially confirmed the results from the discovery cohort. I would have liked to see an additional representation of this data since it is difficult to evaluate their claims based only on the heat maps in Figure 4. A summary of the quantifications as well as a statistical analysis would help.

Response: We appreciate the suggestion, which allowed us to improve our presentation of IHC results for better understanding. The table summarizing the scoring results of IHC staining for each protein in the invasive tumor front (ITF) and the inner tumor (IT) is now included in Supplementary Tables 19-20 of the revised version. The scoring method is described in Supplementary Table 18. The scores for each patient per protein used to build the heat maps are included in Supplementary Table 19 (Panel 2). In Supplementary Table 19 (Panel 1), we also separately included the results of IHC for the cases used for MS analysis. In addition, the data used for the statistical analysis of the IHC were added to Supplementary Table 20.

Further, the authors keep LTA4H, PGK1 and ITGAV for further analysis even though the IHC validation was not in concordance with the discovery results for these proteins. The rational being that the IHC results was unreliable since the proteins are expressed also in the stroma (lines 235 and 394) or that the antibody is non-specific (line 393). These are potential explanations, but a more obvious one would be that the proteins were not validated because the assumptions were not true. Also, with IHC analysis it should be possible to distinguish the expression of proteins between neoplasm and stroma. Further, if the antibodies were non-specific than the results would have little clinical value.

Response: The reviewer is correct; our interpretation regarding the non-specificity of antibodies was not correct. Thank you for the opportunity to clarify these results and our statements because we were

mistaken to state in the original version that there was “opposite expression” between MS and IHC. The revised text was corrected accordingly, and we would like to explain it below:

1. MS was performed in 20 cases (in total 120 LC-MS/MS runs) of OSCC, whereas IHC was performed in 125 cases of OSCC for each protein in neoplastic islands or in 96 cases of OSCC for each protein in the tumor stroma (including 14 cases used for MS); in total, this involved 788 IHC experiments.
2. To elucidate this question regarding the specificity of the antibodies, we are showing below only the immunohistochemistry of 14 of the 20 cases used in discovery proteomics (unfortunately, we did not have tissues from all 20 of the cases used for MS to be analyzed by IHC).
3. Notably, as demonstrated in Response Table 2, the IHC results revealed a similar protein expression pattern in the cases analyzed by discovery proteomics (we also included it in Supplementary Table 19, panel 1). In this manner, we demonstrated that the IHC results agree with the discovery experiments, suggesting that the antibodies are specific but that the MS and IHC techniques have distinct dynamic ranges for quantitation.

Table 2 (Response). Scoring on a scale of 0 to 6 (low to high protein staining) to evaluate the abundance of proteins by IHC in 14 invasive tumor front (ITF) and inner tumor (IT) cases used in discovery analysis.

Protein	Invasive Tumor Front (ITF)								Inner Tumor (IT)								Discovery Phase Log2 LFQ ITF/Inner Tumor ratio
	Score (Intensity + % of immunostaining)								Score (Intensity + % of immunostaining)								
	0	1	2	3	4	5	6	N (total)	0	1	2	3	4	5	6	N (total)	
CSTB	-	-	12	1	-	-	-	13	-	-	9	1	3	-	-	13	-1.55
LTA4H	-	-	12	-	2	-	-	14	-	-	7	1	6	-	-	14	-0.83
NDRG1	-	-	3	2	9	-	-	14	-	-	-	4	10	-	-	14	-1.47
PGK1	-	-	10	1	3	-	-	14	-	-	8	1	5	-	-	14	-0.48
COL6A1	-	-	-	-	14	-	-	14	1	-	-	-	13	-	-	14	0.534
ITGAV	-	-	6	-	1	-	-	7	-	-	-	-	2	5	-	7	-1.674
MB	3	-	3	-	2	-	-	8	6	-	1	1	-	-	-	8	2.901

4. The number of cases was increased in the targeted phase (IHC); 125 cases of neoplastic island and 96 cases of tumor stroma were evaluated, and most of the results are similar to those for the original approximately 14 cases. However, for LTA4H, PGK1, and ITGAV staining, we observed slight variation within the lower and higher scores in each region, whether ITF or IT.
5. The proteins selected from stroma and neoplastic islands specifically stained the areas where they were identified by MS, but unlike the other protein staining, LTA4H and PGK1 also presented peripheral staining in neoplastic cells and were detected in cells from the tumor stroma, such as inflammatory cells. In addition, ITGAV staining was predominant in the tumor stroma with some staining in neoplastic cells. Therefore, LTA4H, PGK1 and ITGAV staining variation does not have any association with antibody selectivity or specificity but this variation is associated with the increase in the number of cases.
6. We thank the reviewer for highlighting this issue because the confirmation of antibody specificity led us to include the information for ITGAV in Figure 5b and Table 2 of the revised manuscript.

We would like to add that the antibodies were carefully selected using the following four steps: (1) datasheet information analysis, (2) The Human Protein Atlas information analysis, (3) our Western blot experiments, and (4) IHC standardization.

1. Datasheet and literature information: All antibodies were purchased from Sigma-Aldrich, and according to their datasheets, most were validated by Western blotting against recombinant proteins (CSTB, NDRG1, LTA4H, PGK1). Exceptions include anti-ITGAV (no bands were detected using a

standard panel of samples, but it was validated by other techniques), anti-COL6A1 (validated by other techniques), and anti-MB (validated by other techniques).

2. Human Protein Atlas information: Each antibody was validated by the Human Protein Atlas Project (www.proteinatlas.org) in normal and tumor samples from different organs (please also see below).

- **Antibody: anti-CSTB (HPA017380, Sigma)**
Protein Atlas results and comments: general – selective cytoplasmic and nuclear expression in squamous epithelia and urothelium, head and neck cancer – cytoplasmic/membranous nuclear expression.
- **Antibody: anti-LTA4H (HPA008399, Sigma)**
Protein Atlas results and comments: general - nuclear and cytoplasmic expression in most tissues, head and neck cancer – strong nuclear expression.
- **Antibody: anti-NDRG1 (HPA006881, Sigma)**
Protein Atlas results and comments: general – cytoplasmic and nuclear expression, head and neck cancer – cytoplasmic/membranous nuclear expression.
- **Antibody: anti-PGK1 (SAB1300102, Sigma)**
PGK1 in human hepatocarcinoma tissue with Anti-PGK1 (Center S320) Antibody at 40 µg/mL. When this antibody was selected, it was presented in The Human Protein Atlas, but for some reason it was removed from the webpage, and we retrieved this information from Sigma. The image below is from The Human Protein Atlas at the time it was selected.
- **Antibody: anti-COL6A1 (HPA019142, Sigma)**
Protein Atlas results and comments: general – distinct positivity in connective tissue and extracellular matrix, head and neck cancer – cytoplasmic and membranous expression.
- **Antibody: anti-ITGAV (HPA004856, Sigma)**
Protein Atlas results and comments: general – cytoplasmic and membranous expression in most tissues, head and neck cancer – cytoplasmic and membranous expression in stroma and neoplastic cells.
- **Antibody: anti-MB (HPA003123, Sigma)**
Protein Atlas results and comments: general – highly specific cytoplasmic expression in striated muscle, head and neck cancer – no expression. In this case, we observed a faint staining of this antibody.

Response Figure 4. Comparison between the immunohistochemical staining of head and neck cancer from *The Human Protein Atlas* and labeling with the same antibodies in the samples from this study. Note the difference in contrast attributed to the type of hematoxylin used (eg. Harris, Mayer, Carrazi, etc.).

3. Evaluation of specificity and selectivity. We also performed some experiments to evaluate the specificity and selectivity of the antibodies for OSCC by Western blotting; the methods and results are shown in Supplementary Figure 8. We tested cells that could represent neoplastic islands (SCC-9 ATCC CRL-1629, a tongue cancer cell line; HSC3, a tongue cancer cell line) and stroma (BJ-5TA ATCC CRL-4001, a fibroblast immortalized cell line; CAF, primary cell of oral cancer-associated fibroblasts). We also tested other non-specific cell lines (SKMEL, a skin melanoma cell line; MCF7 ATCC HTB-22, a breast cancer cell line; and A549, an epithelial lung cancer cell line). We observed that most of the antibodies recognized the protein in all cell lines with the exception of anti-COL6A1, which recognized COL6A1 primarily in stroma cells. We also observed some “non-specific” bands for some antibodies, especially for anti-PGK1; these bands were not seen for LTA4AH and COL6A1 (Response Table 3).

The PGK1 antibody was the only antibody that recognizes both higher and lower apparent molecular masses compared to the expected PGK1 molecular mass of 41.0 kDa. It is interesting that most of the bands are shared among the cell lines, with minor variations. We can speculate that several post-translational modifications, including ubiquitination, phosphorylation, acetylation, methylation, and others (PhosphoSitePlus, phosphor.Elm, Phosida) affect the mobility of PGK1 and associated apparent molecular weight. In addition, we can also speculate that the lower molecular masses result from cleavage by other proteinases (Merops). However, these are only hypotheses; a complete understanding is beyond the scope of this study.

Response Table 3: Summary of Western blot performed with extracts of seven different cell lines:

Gene name	Neoplastic island		Stroma		Other cell lines			Unspecific bands
	SCC-9	HSC3	BJ-5ta	CAF	SK-MEL-28	MCF7	A549	
CSTB	+	+	+	+	++	+	+	**
LTA4AH	-	+	+	-	-	-	++	-
NDRG1	-	+	+	++	++	-	+	*
PGK1	+	+	+	+	++	+	++	**
COL6A1	+	-	+	++	-	-	-	-
ITGAV	+	+	+	++	+	+	+	**
MB	-	+	-	-	++	+	++	**

*Low (+) to high (++) intensity within the cell lines.
* one unspecific band; **more than one unspecific bands.*

4- Standardization for IHC: Prior to the immunohistochemical reactions, we standardized the immunohistochemical tests in 5 independent samples of oral SCC using 3 different concentrations of antibodies and 2 distinct protocols of antigen retrieval to verify the label patterns; tests were performed with either citric acid or TRIS-EDTA (Response Table 4).

Response Table 4: Standardization of antigenic recovery for IHC:

Gene name	Concentrations of antibody tested in IHC	Concentrations of antibody with the best results in IHC	Antigenic recovery for IHC
CSTB	1:1000; 1:500	1:250	citric acid
LTA4AH	1:200; 1:150	1:100	citric acid
NDRG1	1:200; 1:150	1:100	citric acid
PGK1	1:200; 1:100	1:50	citric acid
COL6A1	1:3000; 1:1000	1:1000	citric acid
ITGAV	1:300; 1:75	1:300	citric acid
MB	1:500; 1:100	1:75	citric acid

For antigen recovery, the protocol with Tris-EDTA was tested, but the best results were obtained with citric acid.

In summary, we believe that these antibodies were appropriate for IHC analysis. It is important to highlight that the IHC results were subjected to a rigorous histology analysis by two pathologists, who observed that each antibody was specific to the area where the proteins were identified, with the exceptions of LTA4H and PGK1, which in addition to presenting peripheral staining in neoplastic cells were also detected in cells from tumor stroma, such as inflammatory cells. ITGAV staining was predominant in the tumor stroma with some staining observed in neoplastic cells.

In addition, we demonstrated that the IHC performed with the cases used for discovery proteomics revealed a similar protein expression pattern as the MS results (Supplementary Table 19), despite the distinct dynamic range of MS relative to IHC. Finally, our results suggest that the antibodies are of sufficient quality for use in IHC analysis in combination with other analyses in the pipeline.

To evaluate the survival analysis (Figure 5) it would be good if the group sizes were indicated for each group in each Kaplan-Meier plot.

Response: Thank you for the opportunity to improve Figure 5. As requested, the missing information was added to Figure 5 in the revised version.

The strongest finding from the IHC validation was an association between local recurrence and low expression of CSTB in the ITF demonstrated in the multivariate analysis (p=0.0478). Even though this result is close to the significance threshold, it is fully in line with the result from the discovery cohort.

Response: Thank you for your comment.

SRM validation

It is not entirely clear to me how the targeted SRM analysis relates to the discovery and IHC analysis. In the two latter, regional differences in protein expression is considered and used for correlation to clinical parameters. In the SRM analysis saliva from node positive cancer patients is compared to saliva from node negative patients.

Response: We thank the reviewer for highlighting this very interesting topic that is important to discuss. For this study, we planned the following workflow:

First, we used histopathology-based knowledge. Histopathology is the gold standard routine to determine the prognosis of patients with OSCC based on the evaluation of morphological findings, and we used this information to spatially distinguish the protein abundance in different tumor areas using discovery proteomics.

Second, we verified the selected proteins in a higher number of samples using IHC.

Third, the related proximal tissue fluid, saliva, was assessed for its ability to reflect the relevant proteins of the tissue, which would make saliva a very promising source of prognostic markers. As mentioned before, saliva can host proteins from diverse origins.

The lymph node status was selected as the clinical parameter for two main reasons:

1) One reason is that lymph node metastasis is the most important factor for prognosis in oral cancer (Larsen et al., 2009; Ho et al., 2017), and its presence is highly associated with a worse prognosis (Scully & Bagan, 2009).

2) Another reason is that compromised lymph nodes show significant clinical correlation using linear regression with ITGAV and COL6A1, and other proteins show association with clinical stage (MB) and disease-free survival (PGK1) (Table 1 main text), which are clinical data known to correlate with lymph node metastasis in oral cancer (Fan et al., 2011; Arduino et al., 2008; Greenberg et al., 2003).

As I understand from Figure 6f, the authors suggest that proteins that are higher in the inner part of the tumor (and therefore closer to the oral epithelium) are more likely to end up in the saliva. This may be true, but it would not explain why there would be a difference in the abundance of these proteins between node positive and node negative patients. Irrespective of the presence of tumor cells in the lymph nodes all patients will have inner tumor cells that are closer to the oral epithelium and able to release proteins into the saliva.

Response: Our original text may have given the wrong impression of what we were trying to accomplish. Figure 6f shows that the tissues can contribute to the composition of saliva, but we found that the tissues closest to the oral epithelium do not necessarily contribute more. Instead, the protein composition is driven by a combination of other factors associated with the host response and microenvironment. Moreover, it is important to remember that saliva is a complex fluid composed mostly of water (97–99.5%); other components include electrolytes, proteins, peptides, polynucleotides, hormones, enzymes, cytokines, antibodies, and other substances (Ship and Fisher, 1997; Carpenter, 2013), and these components originate from major salivary glands and minor glands (Humphrey and Williamson, 2001). Other known components include epithelial cells (Dawes, 2003), resident microbial communities (Takeshita et al., 2016), exosomes, microvesicles, and apoptotic bodies (Xiao and Wong, 2012, Mengxi et al., 2017). The composition of saliva is very dynamic, and the most well described mechanism of transport into saliva is from serum into salivary glands by active transport, passive diffusion, or simple filtration; acinar cells also pump sodium ions into the duct (Lee and Wong, 2012). Most compounds found in blood are also present in saliva, and therefore, saliva can host proteins from diverse sources.

Although it is beyond the scope of this study, the gradient that influences the protein concentration in saliva is not yet known, but we noticed that the proteins can have specific abundance independently of their abundance in the adjacent tissues. This is why we designed the last figure of this manuscript (Figure 6f), which highlights the complexity of these dynamic events. We have improved the description of Figures 6e and 6f (on page 14).

It is also difficult to evaluate the results from the SRM analysis for several reasons. In Figure 6a I am missing error bars that would help the reader to estimate variation in the measurements.

Response: We thank the reviewer for this observation. We improved the graph visualizations in Figure 6 of the revised version to provide clear evidence of the differences between N0 and N+. Fortunately, we were also able to include samples from 4 new N0 patients and 3 new N+ patients that we collected and analyzed during manuscript revision. We reanalyzed the data and included details about data analysis in the Material and Methods section. Briefly, the SRM data did not have a normal distribution (data were not log transformed), and the comparison of the levels of the monitored peptides and proteins between groups of patients was performed using the nonparametric Mann-Whitney U test, according to previous publications (Martinez-Garcia et al., 2017; Demeure et al., 2016, Yang et al., 2015). P-values were adjusted for multiple comparisons using the Benjamini-Hochberg FDR method (Benjamini and Hochberg, 1995). Adjusted P-values less than 0.05 were considered statistically significant. In addition, as suggested by reviewer 1, “Would combination of some of these peptides into a small signature improve the performance”, we performed additional analysis and included it in the revised version (Figure 7).

In Figure 6c the scale is so compressed that it is difficult to evaluate the results.

Response: We have also improved the y-axis of previous Figure 6c and replaced it in the revised manuscript.

Also, for LTA4H and COL6A1 the signal of the peptides are much smaller than the spike in (Ratio L/H <0.01). It would be better to titrate the spike-in standard to more closely resemble the concentration of the peptides in the real samples. As it looks now it is difficult to evaluate

the reliability of the quantification or if the signal generated is close to the analytical noise. At least a few example spectra could be added as supplementary figures to convince the reader.

Response: We thank the reviewer for asking this question. It is important to take into consideration that before establishing the final MRM method, we went through different stages of optimization to identify the best conditions to allow the detection of crude heavy-labeled peptide spiked into saliva. We have described the steps that were performed for the optimization of peptide spike-in concentration, the evaluation of matrix interference, retention time prediction, and collisional energy optimization

- 1- The lyophilized mixture containing peptides was prepared based on LC-MS detection. To resuspend peptides, we added 50% acetonitrile containing 0.1% formic acid to obtain a final nominal concentration for each peptide of 2,000 pmol/ μ L in the stock solution, which was diluted to 50 pmol/ μ L in a daughter solution. SIL peptides and iRT were added to the saliva mixture at final nominal concentrations of 800 fmol/ μ L-5 pmol/ μ L and 120 fmol/ μ L, respectively.
- 2- First, this mixture (SIL+iRT) was evaluated without matrix, and then we titrated the spike-in standard in the saliva matrix to guarantee that we could detect the heavy peptides in the saliva matrix with confidence and with as little interference as possible. This mixture was mixed with our samples in a 1:1 ratio.
- 3- Some of the proteotypic peptides, such as those for LTA4H and COL6A1, were detected in the matrix only at higher concentrations. Moreover, for those peptides that were not easily detected, which was the case for LTA4H and COL6A1 peptides, we also added a CE optimization step. After all the optimizations and refinements, we generated the scheduled method with an acquisition window of 3 min.

As requested by the reviewer, we have added a few example chromatograms for LTA4H and COL6A1 to the supplemental figure (Supplemental Figure 10) to better illustrate that even though we had a low L/H ratio for many peptides, the signal is not close to the analytical noise and can be well distinguished using Skyline software. We also included the complete data in Panorama (https://panoramaweb.org/labkey/saliva_SRM.url). Although the data are not public yet, the Skyline team created the login: panorama+paes-leme2@proteinms.net and password: Xdb_Y3PB for the referees. It is important to emphasize that we carefully and manually inspected the peak picking, and for correct peak picking, we have considered different features given by Skyline software such as: dotp and rdotp close to 1.0, co-elution of light and heavy counterpart peptide, the proximity to the predicted retention time, and the reproducibility among peptides of the same proteins and replicates.

The additional quality controls performed for SRM analysis included the following: i) all monitored peptides were detected and presented a measured retention time close to the predicted retention time among the replicates ($r=0.9955$ for set 1 and $r=0.9994$ for set 2) in Supplementary Figure 9; ii) all the light and heavy peptides (Response Figure 5) were detected with reliable abundance, except for peptide TASGSSVTSLDGTR_NDRG1, for which the light form is barely detectable (please see details below); iii) we have also optimized washing steps between runs to avoid carryover between samples by running one blank sample using the trap column and one blank sample using both trap and analytical columns.

Response Figure 5: Peak area for all the 17 proteotypic peptides monitored by SRM (set 1 and set 2) in the 40 saliva samples analyzed in technical replicates are plotted by Skyline for peptide comparison. Three transitions were monitored for each peptide and they are represented in different colors (blue, purple and red).

To better evaluate the reliability of the quantification and whether the signal generated is close to analytical noise, we describe in detail how the analysis was performed:

- We used the information of peptide abundance, dot product values, RT prediction, and manual inspection of elution profiles.
- We evaluated all the spectra in Skyline. For LTA4H and COL6A1, we added a few examples of spectra to the supplementary material (Supplementary Figure 10).

- We observed that most peptides had a L/H ratio > 0.01, and the only peptides that presented in all samples with a L/H ratio < 0.01 were HGA_MB, DLSS_LKH4A, LTY_LKHA4, LSI_COL6A1, and TAEY_COL6A1. As stated previously, the increase of the heavy counterpart was necessary during optimization.
- We considered that even though the L/H ratio is < 0.01 for these five peptides, the reliability of the quantification is not compromised because we spiked in the same concentration of heavy peptides to the matrix for all samples.
- Based on the reviewer concerns and considering that all the proteotypic peptides were reliably identified in at least 38 samples, three peptides (LSIIATDHTYR_COL6A1, TASGSSVTSLDGTR_NDRG1, IYIGDDNPLTLIVK_ITGAV) were excluded in this revised version due to the criteria described below and were not considered for quantification and statistical analysis.

Regarding COL6A1 (LSIIATDHTYR_COL6A1), although we have performed all the necessary optimizations, in the case of the proteotypic peptide LSIIATDHTYR, we revised the results; the light peptide is detected in several samples with a very low intensity and with a chromatogram similar to sample S45 shown in Response Figure 6, and the light peptide was detected with confidence in only one sample (W31) of the N0 group (detected in two out of the three technical replicates) (please see two examples below). Thus, we agree with the reviewer concerning the low intensity of this proteotypic peptide of COL6A1, but the other peptides of COL6A1 (GLEQLLVGGSHLK and TAEYDVAYGESHLFR) are well detected in all samples. With this information in mind, peptide LSIIATDHTYR was excluded, and we have repeated the calculations. The quantification of COL6A1 is now represented by two proteotypic peptides, but the statistical significance of the protein did not change.

Response Figure 6: Intensity of LSIIATDHTYR_COL6A1, which was observed with confidence in only two replicates of the sample W31 of the N0 group.

Regarding NDRG1 (TASGSSVTSLDGTR_NDRG1), the revision of the SRM data indicates that one proteotypic peptide of NDRG1 (TASGSSVTSLDGTR) was also detected in only 2 samples of

N+ (observed in only one out of the three technical replicates) and 4 samples of *N0* (observed in 1 or 2 technical replicates), although the optimizations were performed (please see Response Figure 5, second panel). In the beginning of optimization (not shown in the original version), *NDRG1* was represented by three proteotypic peptides, but the light counterpart of one of them (*MADCGGLPQISQPAK*) was not detected after optimization steps.

With the exclusion of *TASGSSVTSLDGTR_NDRG1*, we recognize that *NDRG1* is now represented by only one unique peptide (*EMQDVDLAEVKPLVEK*) that met the criteria described above. We observed a great impact on the statistics by excluding this peptide that lacked reliable quantification; *NDRG1* protein now showed statistical significance in the final analysis.

Regarding *ITGAV*, we also had an issue, which was mentioned in the original version, and the proteotypic peptide *IYIGDDNPLTLIVK* did not have a similar pattern of abundance compared to the other two selected peptides for *ITGAV* (*LQEVGQVSVSLQR* and *STGLNAVPSQILEGQWAAR*), showing an opposite *N+/N0* ratio. Because in the revised version we included saliva from new patients and we revised the SRM data, we studied the behavior of this *ITGAV* peptide, *IYIGDDNPLTLIVK*, more deeply. First, this peptide was selected as a proteotypic peptide because it passed all the criteria of proteotypic peptides, as described in the Materials and Methods in the revised version of the manuscript, and it was selected based on DDA findings as reported in Supplementary Table 25. However, revising the DDA analysis, we noticed that this peptide was not detected in any of the 17 ITF samples and was detected in only 7 IT samples. We noticed that residue ⁶⁴⁵K (⁶⁴⁵KK.IYIGDDNPLTLIVK) can be ubiquitinated (PhosphositePlus, Kim et al., 2015), and we speculated that ubiquitination can restrict the following residue's accessibility to trypsin and lead to limited proteolysis. However, we reassessed our DDA data including ubiquitination as a variable modification, but we could not find any modification in this peptide. Therefore, to avoid the negative interference of this peptide with an unexpected behavior compared to the other two peptides of the *ITGAV* protein, we excluded this peptide from all analyses. Quantification of *ITGAV* is now represented by two proteotypic peptides, but the statistical significance of the protein did not change.

After revising the SRM data, we also improved data visualization according to referee requests, and we also included the following:

- Pearson's correlation to analyze the reproducibility among replicates for all peptides analyzed in saliva samples, stratified in *N0* and *N+*. We observed great reproducibility among the replicates (Supplementary Figure 11).
- the dendrogram shown in Supplemental Figure 12, and we observed that samples from *N0* and *N+* are clustered within the group.

Further, for the examples LTA4H and COL6A1 (Figure 6d), even if the ROC and Sens./Spec. curves indicate some prognostic value for the two proteins, the distribution graphs does not really support a good resolution of node positive and node negative patients based on the levels of the proteins in saliva.

Response: Considering the suggestion of reviewer 1, we reanalyzed the data using machine learning methods to build ROC curves, and we replaced the analysis in Figure 6 with the new analysis in Figure 7 in the revised manuscript. We also improved Figure 6c to demonstrate more clearly the differences between *N+* and *N0* patients.

References used to respond to the reviewer 3:

- 1- Arduino PA, Carrozzo M, Chiecchio A, Broccoletti R, Tirone F, Borra E, Bertolusso G, Gandolfo S. *Clinical and Histopathologic Independent Prognostic Factors in Oral Squamous Cell Carcinoma: A Retrospective Study of 334 Cases.* (2008). *American Association of Oral and Maxillofacial Surgeons.*
- 2- Benjamini Y, Hochberg Y. *Controlling the False Discovery Rate: A Practical and Powerful Approach to Multiple Testing.* (1995). *J R Stat Soc Ser B Methodol.*, 57:289–300.
- 3- Demeure K, Fack F, Duriez E, Tiemann K, Bernard A, Golebiewska A, Bougnaud S, Bjerkgvig R, Domon B and Niclou SP. *Targeted proteomics to assess the response to anti-angiogenic treatment in human glioblastoma (GBM).* (2016). *Molecular & Cellular Proteomics*, 15, 481-492.
- 4- Fan NJ, Gao Chun-Fang, and Wang XL. *Identification of Regional Lymph Node Involvement of Colorectal Cancer by Serum SELDI Proteomic Patterns.* (2011). *Gastroenterology Research and Practice.*
- 7- Greenberg JS, Fowler R, Gomez J, Mo V, Roberts D, El Naggar AK, Myers JN. *Extent of extracapsular spread: a critical prognosticator in oral tongue cancer.* (2003) *Cancer* 97(6):1464-70.
- 8- Ho AS, Kim S, Tighiouart M, Gudino C, Mita A, Scher KS, Laury A, Prasad R, Shiao SL, Van Eyk JE, Zumsteg ZS. *Metastatic Lymph Node Burden and Survival in Oral Cavity Cancer.*(2017). *J Clin Oncol.*,35(31):3601-3609.
- 9- Hubner, N. C., Cox J, Spettstoesser B, Badilla P, Poser I, Hyman A, Mann M et al. *Quantitative proteomics combined with BAC TransgeneOmics reveals in vivo protein interactions.*(2010). *J. Cell Biol.* 189, 739–754.
- 10- Vinay K. Kartha , Lukasz Stawski , Rong Han, Paul Haines, George Gallagher, Vikki Noonan, Maria Kukuruzinska, Stefano Monti , Maria Trojanowska. *PDGFR β Is a Novel Marker of Stromal Activation in Oral Squamous Cell Carcinomas.* (2016). *PLoS ONE* 11(4): e0154645.
- 11- Kim WI, Bennett EJ, Huttlin EL, Guo A, Li J, Possemato A, Sowa ME, Rad R, Rush J, Comb MJ, Harper JW, Gygi SP. *Systematic and quantitative assessment of the ubiquitin-modified proteome.* (2011) *Mol Cell* 44, 325-40.
- 12- Larsen SR, J. Johansen, JA, Krogdahl SA. *The prognostic significance of histological features in oral squamous cell carcinoma.* (2009). *J Oral Pathol Med.*, 38: 657–662.
- 13- Marakalala MJ, Raju RM, Sharma K, Zhang YJ, Eugenin EA, Prideaux B, Daudelin IB, Chen PY, Booty MG, Kim JH, Eum SY, Via LE, Behar SM, Barry CE 3rd, Mann M, Dartois V, Rubin EJ. *Inflammatory signaling in human tuberculosis granulomas is spatially organized.* (2016). *Nat Med.*(5):531-8.
- 14- Martinez-Garcia E, Lesur A, Devis L, Cabrera S, Matias- Guiu X, Hirschfeld M, Asberger J, van Oostrum J, de los Angeles Casares de Cal M, Gómez-Tato A, Reventos J, Domon B, Colas E, Gil-Moreno A. *Targeted proteomics identifies proteomic signatures in liquid-biopsies of the endometrium to diagnose endometrial cancer and assist in the prediction of the optimal surgical treatment.*(2017). *Clin Cancer Res.* 23(21):6458-6467.
- 7- Rivera C , Oliveira AK, Costa R.A.P, De Rossi T, Paes Leme AF. *Prognostic biomarkers in oral squamous cell carcinoma: A systematic review.* (2017). *Oral Oncol.*, 72:38-47.
- 8- Rodrigues PC, Sawazaki-Calone I, de Oliveira, CE, Macedo CCS, Dourado MR, Cervigne NK, Miguel MC, do Carmo AF, Lambert DW, Graner E, da Silva SD, Alaoui-Jamali MA, Paes Leme AF, Salo TA, and Coletta RD . *Fascin promotes migration and invasion and is a prognostic marker for oral squamous cell carcinoma.* *Oncotarget.* (2017). 8(43): 74736–74754.
- 9- Scully C and Bagan J. *Oral squamous cell carcinoma overview.* (2009). *Oral Oncology* 45:301–308.
- 15- Storey J.D., *Strong control, conservative point estimation and simultaneous conservative consistency of false discovery rates: a unified approach.* (2004). *Journal of the Royal Statistical Society, B* 66, 187-205.
- 10- Tyanova S, Temu T, Sinitcyn P, Carlson A, Hein MY, Geiger T, Mann M, Cox J. *The Perseus computational platform for comprehensive analysis of (prote)omics data.* (2016). *Nat Methods* 13(9):731-40.

- 11- Wiśniewski JR, Ostasiewicz P, Duś K, Zielińska DF, Gnad F, Mann M. Extensive quantitative remodeling of the proteome between normal colon tissue and adenocarcinoma. (2012). *Mol Syst Biol.*, 8:611.
- 12- Yang T, Xu F, Fang D, Chen Y. Targeted Proteomics Enables Simultaneous Quantification of Folate Receptor Isoforms and Potential Isoform-based Diagnosis in Breast Cancer. (2015). *Sci Rep.* 5:16733.

Reviewers' comments:

Reviewer #1 (Remarks to the Author):

The authors have made significant efforts to improve their manuscript. The concerns raised in the previous submission were carefully addressed, language has been improved and figure quality is dramatically better.

Overall this is a large amount of work, combining discovery proteomics and targeted validations. While none of these markers are going to enter the clinic any time soon, this work presents the important first steps towards translational implementation. Similar papers have been published in Nature Communications.

I recommend publication of this manuscript.

Reviewer #2 (Remarks to the Author):

The manuscript is much improved with changes in Title and rewriting. However, Manuscript presentation would improve with the following modifications.

- 1) IRB approval for the 29 OSCC samples from the University of Oulu, Finland, assuming the authors have obtained, needs to be stated.
- 2) A better IHC figure of PGK1 in figure 4A would make the data acceptable to the readers
- 3) Sentence in lines 308-311 of the results section and lines 541-545 of the discussion are vague. Abundance in line 309 could be taken as higher expression instead of level of expression the sentence is trying to convey. It is better to make it clear that lower expression of CSTB and NDRG1 are associated with local recurrence, development of second primary tumor and poorer disease free survival (figure 5 and table 2). However, higher expression of PGK1 and ITGAV are associated with local recurrence, development of second primary tumor and poorer disease free survival. Similar statements could be made in the discussion for lines 541-545.

Reviewer #3 (Remarks to the Author):

Reply to revision:

I do not think that the the points raised in the previous round of review have been satisfactorily addressed. Critical issues still remain as exemplified below.

General problems with the manuscript are as follows:

1. Vague statements are made throughout the manuscript, with in many cases borderline or missing statistical support. It is unclear if the authors themselves believe/suggest that the markers that are presented have true clinical value or not. This was pointed out by myself and reviewer 2 in the initial review, and it has not been corrected. In order to exemplify this I summarized the findings in relation to CSTB here below (point I-V):

I. Discovery using MS:

CSTB was found lower in ITF than in inner tumor (Figure 3a).

Finding based on t-test without correction for multiple testing, i.e. not statistically significant.

Association with clinicopathological data suggested connection between CSTB and treatment,

unclear in what way (Table 1).

II. Evaluation using IHC:

CSTB was evaluated in 114 cases using IHC (Table S19, panel2) in ITF and inner tumor. Differential staining scores were calculated and reported in Table S20. This analysis indicate the following differential staining of CSTB: ITF>Inner in 49 cases, ITF=Inner in 29 cases, ITF<Inner in 36 cases, i.e. there are more cases where CSTB is higher in ITF than in inner tumors, not confirming what was suggested in the discovery phase.

Still, the authors conclude that "Increased CSTB and NDRG1 expression was identified in the inner tumor, according to the discovery results" (line274).

In addition, from what I could understand, no statistical test was used for the interpretation of IHC scorings.

Still, multivariate analysis indicated a link between cases with CSTB in ITF<Inner and local relapse.

III. Evaluation using SRM in saliva:

Cohort containing 40 patient samples (14 NO, 26 N+)

Protein level analysis indicate lower CSTB level in saliva from N+ patients.

Peptide level analysis of CSTB indicate no difference between NO and N+ for 2 peptides and lower signal in N+ patients for one peptide (the shortest peptide, Figure 6b), i.e. the majority of peptides indicate no difference in CSTB between NO and N+. Still, the SRM method is reporting lower CSTB protein level in saliva from N+ patients (Figure 6a) which is difficult to explain/understand.

The statistical test chi-square did not support a connection between CSTB and lymph node metastasis (line 360-367).

IV. Machine learning for prediction of lymph node metastasis:

Based on machine learning the authors propose predictive power of peptides, the top one being the single CSTB peptide that differed between NO and N+ patients.

V. In summary, it is still not clear if CSTB is expressed higher in inner tumor, or if there is a connection to lymph node metastasis.

2. The findings reported have questionable clinical value for several reasons. Due to non-transparent reporting of findings and unclear support of statements as discussed above, the generalizability of these findings in clinical samples is difficult to assess. From a practical point of view it is difficult to envision how the proposed markers should be used in the clinic. Are the authors suggesting that SRM should be used on saliva samples to look for missing peptides in order to predict lymph node metastasis?

3. From a methods point of view, none of the used analytical procedures are novel. The analytical depth of the analysis is far from comprehensive, most current high-quality clinical proteomics studies based on tissue samples quantify at least 6000 proteins across all samples. Here, the overlap analysis was based on a few hundred proteins. Even for micro-dissected samples these numbers are low.

4. In general it is very difficult to follow the reasoning of the study. The reason is that too much text mass is spend on describing details when not needed (e.g. when reporting findings of questional value from clinicopathological data or when describing machine learning method details that are better fitted in the methods section). Conversely, when a more detailed description is warranted it is lacking (e.g. more transparent and clear description of the statistical and clinical value of specific markers).

For all of the above mentioned reasons I do not find the manuscript suitable for publication in Nature Communications.

Reviewers' comments:

Reviewer #1 (Remarks to the Author):

The authors have made significant efforts to improve their manuscript. The concerns raised in the previous submission were carefully addressed, language has been improved and figure quality is dramatically better.

Overall this is a large amount of work, combining discovery proteomics and targeted validations. While none of these markers are going to enter the clinic any time soon, this work presents the important first steps towards translational implementation. Similar papers have been published in Nature Communications.

I recommend publication of this manuscript.

Response: We appreciated the comments.

Reviewer #2 (Remarks to the Author):

The manuscript is much improved with changes in Title and rewriting. However, Manuscript presentation would improve with the following modifications.

1) IRB approval for the 29 OSCC samples from the University of Oulu, Finland, assuming the authors have obtained, needs to be stated.

Response: This information was included in the material and methods: The use of OTSCC samples was approved by National Supervisory Authority for Welfare and Health (Valvira, Dnro 7449/06.01.03.01/2013).

2) A better IHC figure of PGK1 in figure 4A would make the data acceptable to the readers

Response: A better figure was included in the revised version of the manuscript.

3) Sentence in lines 308-311 of the results section and lines 541-545 of the discussion are vague. Abundance in line 309 could be taken as higher expression instead of level of expression the sentence is trying to convey. It is better to make it clear that lower expression of CSTB and NDRG1 are associated with local recurrence, development of second primary tumor and poorer disease free survival (figure 5 and table 2). However, higher expression of PGK1 and ITGAV are associated with local recurrence, development of second primary tumor and poorer disease free survival. Similar statements could be made in the discussion for lines 541-545.

Response: Thank you for the suggestions to improve these descriptions, all of them were included both in results and discussion in the revised version of the manuscript.

Reviewer #3 (Remarks to the Author):

Reply to revision:

I do not think that the the points raised in the previous round of review have been satisfactorily addressed. Critical issues still remain as exemplified below.

General problems with the manuscript are as follows:

1. Vague statements are made throughout the manuscript, with in many cases borderline or missing statistical support. It is unclear if the authors themselves believe/suggest that the markers that are presented have true clinical value or not. This was pointed out by myself and reviewer 2 in the initial review, and it has not been corrected.

Response: In summary, i) we rewrote the statements that are possibly considered as vague in the revised manuscript; ii) We included information of the association of CSTB and treatment; iii) we gave further details of the analysis of IHC results to avoid incorrect interpretation; iv) we improved the abstract to restate the phases of the study and the main findings; v) we improved the conclusion of the manuscript to highlight the main findings of the study and stated the clinical value of this study, including limitations regarding clinical implementation (lines 490-501). We gave details of these issues point-by-point in the specific issues.

Although our study is currently the largest investigation of OSCC candidate markers using tissue in the discovery phase and two parallel verification phase with immunohistochemistry and SRM-MS in saliva, we need to emphasize the requirement of a validation step in a larger cohort of patients with longitudinally collected samples, in order to consider the application in the clinical practice.

Regarding the clinical value, with no doubt, the reviewer's suggestion is coherent, but at this stage of the study, it is not possible to suggest or affirm the end point of monitoring CSTB protein in oral cancer patients from a clinical point of view. Therefore, the clinical value of this study is:

For the discovery phase, considering the histopathological-based prognostic value of the invasive tumor front (ITF) region, we used histopathology-guided laser microdissection and discovery proteomics of FFPE tissues to isolate six unique regions of neoplastic islands and tumor stroma. Potential candidates with prognostic values were indicated. These candidates were evaluated in verification phases, which indicated:

- i) CSTB was shown to be expressed at low expression levels in the ITF of neoplastic islands, and by multivariate analysis was considered as an independent marker for local recurrence in patients with OSCC. The result shown here is significant, but it requires replication and validation. We foresee the validation of CSTB as a marker of local recurrence in a larger cohort because it can be used in clinical practice to plan the treatment modality and treatment monitoring.*
- ii) The same for SRM-MS of saliva, which indicates the combination of specific peptides from LTA4H, COL6A1 and CSTB distinguishes N+ and N0 patients. It is important to clarify the difference between the identification of this signature as a potential prognostic signature and creating a predictive prognostic model with this signature. We may create in the future a model using this signature and apply it to other samples and to any other group of samples. The performance of this predictive model is simulated in this study by repeated cross-validation. So far, this is the best performance found in the search of identifying potential signatures that can distinguish oral cancer patients regarding lymph node metastasis development (N+) or not (N0).*

With that in mind, we stated that the results are robust and confident for the verification phase to make a bridge with the next step of prognostic biomarkers validation for oral cancer. This path of biomarker discovery is in great agreement with literature (Whiteaker et al. 2011; Rifai, Gillette, and Carr 2006).

Regarding the mentioned comments of reviewer 2 in the initial review, in the last revision he/she stated "The manuscript is much improved with changes in Title and rewriting" and suggested minor points to improve manuscript presentation, we, therefore, concluded that our explanations met his/her expectations.

In order to exemplify this I summarized the findings in relation to CSTB here below (point I-V):

I. Discovery using MS:

CSTB was found lower in ITF than in inner tumor (Figure 3a).

Finding based on t-test without correction for multiple testing, i.e. not statistically significant.

Response: *This question was commented by the reviewer 1 and 3 in the first round of revision and we have explained our reasons in the last revision. The reviewer 1 was satisfied with our explanation. We repeated our previous explanations and gave additional reasons to convince the reviewer 3.*

1. We applied Benjamini-Hochberg and Permutation-based FDR using the Perseus software for multiple testing corrections, which resulted in no proteins with significant difference between the compared groups (front and inner tumor). Also, we tested the multiple hypothesis corrections using Storey's test (Storey, Taylor, and Siegmund 2004). The *q*-values were calculated considering a maximum FDR of 5%, which also resulted in no significant proteins from the neoplastic islands and 13 significant proteins from the stroma dataset, as listed in the previous Response Letter. Considering that this phase of analysis was not the endpoint of this study, the fact we did not adjust the *p*-value is not a limitation of this study. In fact, this was confirmed by the findings of the verification phase using two parallel approaches, evaluating tissue and biofluid, which were able to confirm the targets identified in the discovery phase. Conversely, the test we did with multiple test correction using Storey's method removed proteins that presented an association to the prognosis of oral cancer patients by immunohistochemistry and SRM-MS. Therefore, we would have lost this valuable information if we had filtered out those proteins from discovery phase.

2. Besides, correction for multiple testing is not considered a *sine qua non* rule in proteomics data analysis and has its limitations:

- i) Recent study of Pascovici et al. (2016) lists some key factors that make multiple testing corrections less effective in proteomics, and in summary suggested that "multiple testing corrections method may also fail to detect any true positives even when many exist" (false negatives) and "it should be employed as a useful tool but not be regarded as a required rubber stamp". Therefore, in any field the data have to be treated individually respecting rules to have high-confidence results, without prejudicing what the biology is telling.
- ii) Burt et al. (2017) shows that early phase clinical studies are typically underpowered. The authors mention: "In addition, an asymmetry between the two types of statistical errors means that 'false-negatives' are more likely to occur than 'false-positives,' are cumulative, and exit the development process upon discovery, preventing the verification of their 'falseness' in higher-powered follow-up studies. The resulting 'false-negatives' mean loss and delay of effective treatments to patients and could be worth billions of dollars in untreated morbidity and mortality, and loss of commercial benefits to treatment developers. The simulations shown in this study provide information about the magnitude and correlated of the 'false-negatives' to support informed developmental decisions, and suggest that higher-powered early phase studies are worth the investment. The findings require replication, validation using a spectrum of therapeutic areas and developmental scenarios, and debate by the relevant treatment development stakeholders".
- iii) Additionally, there are several examples of discovery proteomics data that show the multiple correction test was also not applied, as illustrated by the recent work from Kim et al. (2016), in which target proteins were selected from a previous work of the group involving discovery proteomics of extracapsular prostate cancer samples, with no multiple test correction of *p*-value (Kim et al. 2012), and likely, a recent manuscript of Mathias Mann group (Marakalala et al. 2016) that also used FFPE and laser microdissection tissues of patients.

3. In addition, considering the samples from neoplastic islands, due to the high number of proteins with $p\text{-value} > 0.05$, the FDR method, if applied, will tend to eliminate all the variables in order to reduce the chances of identifying false positives. On the other hand, this does not guarantee that we are maintaining all the possibilities of false negatives; in other words, among the identified proteins, even though there is a considerable probability of false positives, there is also a probability of false negatives that we need to consider. By eliminating all the proteins, we increase false negative rate and might lose important ones. This is an issue that has been previously reported in the literature and reinforces the fact that it is problematic to adjust the p -value in all cases (<http://www.biostathandbook.com/multiplecomparisons.html>).

4. For stroma, we could have performed FDR correction, as we tested in the first revision and we could have selected $FDR < 5\%$, 10% or 20% , depending on the amount of risk we wanted to take in having false positives and eliminating false negatives.

5. If we had applied the p -value correction (Storey method, $p < 0.05$) as we tested in previous version, we would not have any differential proteins from neoplastic islands, and only 13 differential proteins from stroma, from which only integrin alpha-V (ITGAV) and fascin (FSCN1) remained in the list. On the other hand, the other proteins from this list of 13 proteins did not show clinical correlations, and therefore, there is a chance that these proteins are also false positives. In contrast, some proteins that were excluded by not reaching significance after p -value correction, had a clinical correlation, and could be false negatives. In summary, the goal of multiple comparisons corrections is to reduce the number of false positives, but an unfortunate product of correcting for multiple comparisons is that you may also lose a number of true positives. The smaller the sample size you are working with, the higher is the number of false negatives produced by the selection by p -value correction.

6. Therefore, all these reasons indicate that the use of Student's t -test without correction for multiple testing does not decrease the quality or the meaning fullness of the discovery phase.

Association with clinicopathological data suggested connection between CSTB and treatment, unclear in what way (Table 1).

Response:

i) In the first review, the reviewer 3 asked us to include the information about treatment and second primary tumor: **“Table S1 holds the clinicopathological data for the discovery cohort, but it does not contain information about treatment or second primary tumor. Still, Table 1 indicate that there is a statistically significant association between e.g. treatment and CSTB or PGK1 and second primary tumor.”**

Our response was **“As requested, the missing information was added to Supplementary Table 1 and 17”** and we included in the footnote from Table 1 the three categories of treatment: (1) surgery, (2) surgery and radiotherapy, (3) combination of surgery, radiation and chemotherapy.

ii) Additional explanation about the correlation of CSTB expression with the treatment (Table 1) was included in the current revised manuscript (lines 440-444). These results indicated that the patients that received the treatments combining surgery, radiotherapy and chemotherapy correlated with lower

expression of CSTB in the ITF. This therapeutic modality is prescribed to patients with advanced/aggressive tumors. Accordingly, we further demonstrated in IHC analysis that the low expression of CSTB in the ITF was also associated with local recurrence, which results from aggressive or advanced tumors.

iii) To give further details of the results, we included the box plots of all the proteins described in the Table 1 in the Supplementary Figure 6.

II. Evaluation using IHC:

CSTB was evaluated in 114 cases using IHC (Table S19, panel2) in ITF and inner tumor. Differential staining scores were calculated and reported in Table S20. This analysis indicate the following differential staining of CSTB: ITF>Inner in 49 cases, ITF=Inner in 29 cases, ITF<Inner in 36 cases, i.e. there are more cases where CSTB is higher in ITF than in inner tumors, not confirming what was suggested in the discovery phase.

Still, the authors conclude that “Increased CSTB and NDRG1 expression was identified in the inner tumor, according to the discovery results” (line274). In addition, from what I could understand, no statistical test was used for the interpretation of IHC scorings.

Response: The reviewer is correct that in Table S20 we indicated ITF>Inner in 49 cases, ITF=Inner in 29 cases and ITF<Inner in 36 cases. However, the reviewer may have confused the two forms of data presentation: protein expression (absolute levels) versus categorical classification (frequency) of levels of protein expression.

i) First, we performed the analysis of protein expression based on visual inspection of three blind and independent pathologists (kappa=0.706), in a multi-centric analysis, and the slides were scored according to the method described in Table S18.

Intensity	%	COMB
0 – negative	0: 0%	Intensity + % = 0 to 6
1 – weak	1: 1 to 25%	
2 – moderate	2: 26 to 50%	
3 – strong	3: > 50%	

ii) We generated the Table S19, which has two kinds of data: protein expression (combined score from 0 to 6, Table S18) and the frequency of occurrence (patients).

		Invasive Tumor Fronte (ITF)								Inner Tumor							
		Score (Intensity + % of immunostaining)								Score (Intensity + % of immunostaining)							
		0	1	2	3	4	5	6	n (total)	0	1	2	3	4	5	6	n (total)
Region	Gene name																
Neoplastic Island	CSTB	3	-	77	17	6	10	1	114	44	2	16	11	22	13	6	114
	LTA4H	36	-	6	10	25	24	16	117	63	-	8	20	13	8	5	117
	NDRG1	26	-	8	10	27	32	15	118	20	-	-	9	26	28	35	118
	PGK1	31	-	23	13	29	17	5	118	43	-	15	20	24	13	3	118
Tumor Stroma	COL6A1	1	-	-	1	8	17	66	93	7	-	-	4	13	23	46	93
	ITGAV	54	-	3	-	7	4	12	80	62	-	-	4	4	3	7	80
	MB	80	-	3	-	2	1	-	86	81	-	1	2	2	-	-	86

iii) We show here the bar plots below to further illustrate the data from the two proteins questioned by the reviewer:

The results indicate that the total frequency for CSTB is 114 and for NDRG1 is 118.

CSTB: we demonstrated that in low intensity score, most of the patients showed CSTB with low intensity in the ITF and in the high intensity score, most of patients have higher CSTB expression in the inner tumor.

NDRG1: we demonstrated that in lower intensity score, most of the patients showed NDRG1 with low intensity in the ITF and in the high intensity score, most of the patients have higher NDRG1 expression in the inner tumor.

Therefore, based on these analyses guided by the Table S19, we concluded that IHC confirmed the results of discovery phase for CSTB and NDRG1, as we previously indicated in the Figure 4 and stated in lines 248-251 of the main text.

iv) The visualization of these data in Figure 4 of the manuscript was also included in the first version of the manuscript to help with the interpretation of this result.

Therefore, we decided to use a combinative semiquantitative scoring method based on valued points from 0 to 6 (intensity+% of immunostaining) to increase sensitivity, maximizing the detection and repeatability of 788 slides that were analyzed. The statistical analysis with this extended tissue score is not applicable at this phase (Sundquist et al. 2017; Bello et al. 2011; Wang et al. 2017). A smaller number of score categories could reduce the sensitivity of the scoring method and a large number would lead to less repeatability (Fedchenko and Reifenrath 2014; Shackelford et al. 2002).

v) In the next step, the ordinal scale was combined in a total score allowing us to perform the association with clinical parameters using cross-tabulation and chi-square test, Kaplan-Meier and Cox proportional hazard model. We evaluated the association of tissue staining with clinicopathological parameters. To be able to perform this analysis, we combined the tissue scoring (0 to 6) to generate the clinical scoring (0, 1, 2), as described in Table S18 and below.

Expression difference	Score
ITF – Inner tumor = -6 to 6	
0 = 0	0 = equal expression
-6 to -1 = 1	1 = lower expression in invasive tumor front
1 to 6 = 2	2 = higher expression in invasive tumor front

Regarding CSTB results, Table S20 indicated ITF>Inner in 49 cases, ITF=Inner in 29 cases, ITF<Inner in 36 cases. It is important to emphasize that the frequency of patients in each score does not invalidate the results of expression, because this is only the number of patients and NOT how high or low is the intensity for both ITF and inner tumor.

With respect, to avoid misinterpretation, we illustrated by bar plots the results of CSTB in Table S21 to demonstrate which tissue score and its respective frequency are present in each clinical score (0=equal, 1-lowITF, 2-high ITF). We observed that the 49 patients included as ITF>Inner (High ITF) are mostly composed of patients where the tissue has the tissue staining scored as 2 in the ITF and 0 in the Inner.

Besides, the ITF<Inner (Low ITF) includes mostly the low-end score of ITF and high-end score of inner tumor (Please note that the y-axis scales are different among the plots). For the equal expression group, the patients are distributed within the low and high-end scores. To improve the understanding, we replaced in the Supplementary Tables 21 and 22, the symbols “Low ITF” or “High ITF” with the symbols suggested by the reviewer 3 “ITF<Inner, ITF>Inner and ITF=inner” to avoid misunderstandings.

Therefore, this analysis describes the frequency of patients in each category (low ITF, equal and high ITF), but not how much high or how much low is the variation in the staining intensity, which was previously reported in the tissue scale (0 to 6). Therefore, at this point when data are categorized into low ITF, equal and high ITF (clinical score 0, 1, 2), it is not appropriate to compare with the discovery phase.

Still, multivariate analysis indicated a link between cases with CSTB in ITF<Inner and local relapse.

Response: This is one of the most significant results of the manuscript that indicated that CSTB at low expression levels in the ITF of neoplastic islands is an independent marker for local recurrence. It means that the 36 patients categorized with low ITF (ITF<Inner) developed local relapse in higher frequency than the 49 cases classified as high ITF (ITF>Inner) by Cox proportional hazard model multivariate analysis.

III. Evaluation using SRM in saliva:

Cohort containing 40 patient samples (14 N0, 26 N+)

Protein level analysis indicate lower CSTB level in saliva from N+ patients.

Peptide level analysis of CSTB indicate no difference between N0 and N+ for 2 peptides and lower signal in N+ patients for one peptide (the shortest peptide, Figure 6b), i.e. the majority of peptides indicate no difference in CSTB between N0 and N+.

Response:

i) In the revision process, we included more patients for SRM-MS analysis and in total it is 40 patient samples. The results showed at protein level lower CSTB abundance in N+ patients compared with N0 patients (adj. p-value<0.05).

ii) The reviewer 1 suggested evaluating the combination of peptides to improve the performance and, for that, we evaluated also the peptides individually, similar to Kim et al. (2016). In the results section and Figure 6, we indicated that, among the CSTB peptides, only HDELTYF was statistically significant at adj. p-value<0.05. Regarding the mention of “shortest peptide” referring to HDELTYF compared to the other two peptides, which did not reach the statistical significance at adj. p-value<0.05, we would like to detail below the selection of proteotypic peptides for CSTB:

- The selection of CSTB peptides for SRM-MS analysis was based on SRMAtlas. To contextualize the selection of the three proteotypic peptides for CSTB, we observed that five proteotypic peptides were previously described in SRMAtlas. Two of them, SQVVAGTNYFIK and VHVGDEDFVHLR, were selected because they were also identified in our DDA data (discovery phase in tissue analysis). Among the other three peptides of the SRMAtlas, one has a cysteine and two methionine residues (MMCGAPSATQPATAETQHIADQVR), which are prone to be carbamidomethylated and oxidized, respectively, and it was not selected. We opted for selection the HDELTYF instead of VFQSLPHENKPLTLSNYQTNK, because according to SRMAtlas HDELTYF was monitored in a different number of instruments and there is a potential missed cleavage in the longer peptide, which although lysine is followed by a proline, we preferred to avoid this potential missed cleavage (Rodriguez et al. 2008).

Protein	External Links	Pre AA	Sequence	Fol AA	Adj SS	Source	Q1_mz	Q1_chg	Q3_mz	Q3_chg	Ion	Rank	RI	SSRT	RT_Cat	N_map	QTOF	QTOF_CE	QTrap5500	QQQ	IonTrap	QQQ	QTRAP	
P04080		X	K	VHVGDEDFVHLR	V	1.99	QQQ	474.91	3	237.14	1	b2	1	3735	27.3	27.9	1							
P04080		X	K	VHVGDEDFVHLR	V	1.99	QQQ	474.91	3	425.26	1	y3	2	1990	27.3	27.9	1							
P04080		X	K	VHVGDEDFVHLR	V	1.99	QQQ	474.91	3	336.20	1	b3	3	1325	27.3	27.9	1							
P04080		X	K	VHVGDEDFVHLR	V	1.99	QQQ	474.91	3	593.80	2	y10	4	1102	27.3	27.9	1							
P04080		X	K	SQVVAGTNYFIK	V	1.68	QQQ	663.86	2	216.10	1	b2	1	8551	27.7	31.1	1							
P04080		X	K	SQVVAGTNYFIK	V	1.68	QQQ	663.86	2	315.17	1	b3	2	2780	27.7	31.1	1							
P04080		X	K	SQVVAGTNYFIK	V	1.68	QQQ	663.86	2	913.48	1	y8	3	996	27.7	31.1	1							
P04080		X	K	SQVVAGTNYFIK	V	1.68	QQQ	663.86	2	842.44	1	y7	4	813	27.7	31.1	1							
P04080		X	R	VFQSLPHENKPLTLSNYQTNK	A	1.82	QTOF	615.32	4	218.4472	3	y5	1	2576	32.8	28.4	1							
P04080		X	R	VFQSLPHENKPLTLSNYQTNK	A	1.82	QTOF	615.32	4	738.0506	3	y19	2	1306	32.8	28.4	1							
P04080		X	R	VFQSLPHENKPLTLSNYQTNK	A	1.82	QTOF	615.32	4	787.0734	3	y20	3	1161	32.8	28.4	1							
P04080		X	R	VFQSLPHENKPLTLSNYQTNK	A	1.82	QTOF	615.32	4	247.1447	1	b2	4	812	32.8	28.4	1							
P04080		X	-	MM[C160]GAPSATQPATAETQHIADQVR	S	0.80	QQQ	857.73	3	1011.01	2	y19	1	15889	26.4	29.9	1							
P04080		X	-	MM[C160]GAPSATQPATAETQHIADQVR	S	0.80	QQQ	857.73	3	419.73	2	y7	2	2968	26.4	29.9	1							
P04080		X	-	MM[C160]GAPSATQPATAETQHIADQVR	S	0.80	QQQ	857.73	3	674.34	3	y19	3	2673	26.4	29.9	1							
P04080		X	-	MM[C160]GAPSATQPATAETQHIADQVR	S	0.80	QQQ	857.73	3	768.89	2	y14	4	2588	26.4	29.9	1							
P04080		X	K	HDELTYF	-	0.63	QTOF	462.71	2	253.0937	1	b2	1	3433	25.4	30.9	1							
P04080		X	K	HDELTYF	-	0.63	QTOF	462.71	2	382.1363	1	b3	2	1894	25.4	30.9	1							
P04080		X	K	HDELTYF	-	0.63	QTOF	462.71	2	495.2203	1	b4	3	957	25.4	30.9	1							
P04080		X	K	HDELTYF	-	0.63	QTOF	462.71	2	596.2680	1	b5	4	759	25.4	30.9	1							

- *In addition, the peptide HDELTYF meets other criteria for peptide selection as we described in main text and in literature (Lange et al. 2008; Gallien, Duriez, and Domon 2011): peptide length (5-22 residues) (Deutsch, Lam, and Aebersold 2008; Krokhin 2006); moderate hydrophobicity, peptide uniqueness to CSTB.*
- iii) *Besides that, we used heavy-isotope-labeled peptide standards and the analysis of this peptide met all the quality criteria by using dotp, rdotp and other important features (co-elution of light and heavy counterpart peptide, the proximity to the predicted retention time, and the reproducibility among peptides of the same proteins and replicates) to help us to define the correct peaks. Therefore, there is no doubt that this peptide was correctly assigned.*
- iv) *In addition, the ability of the peptides to have different behavior even from the same proteins lead us to perform all the analysis at peptide level suggested by the reviewer 1, demonstrating that we can change the performance of specific analyte if you tested it in at diverse level and a recent manuscript has shown similar strategies (Kim et al. 2016). Surprisingly, this is exactly what happened in this study, where this peptide of CSTB, combined with two other peptides, one from LTA4H and the other from COL6A1, were able to differentiate the two groups of patients, N0 and N+.*
- v) *One additional comment, in discovery proteomics we usually evaluate peptides and discuss at protein level and in targeted proteomics, we have been doing similarly, but more recently the importance to interpret at peptide level has been emerged and this is what we performed according to the suggestion of reviewer 1. In fact, the combination of LTA4H, COL6A1, and CSTB peptides in saliva outperformed the protein signature in the ability to distinguish between N0 and N+ patients.*
- vi) *Results of machine learning methods reinforce the top predictive power of this peptide compared to the other two peptides:*
 - a- *CSTB peptide alone (Pep12) presented high AUC level compared to other single peptides.*
 - b- *On the other hand, CSTB peptide (Pep12) contributed with Pep8 and Pep9 to form S2, which together with S1 and S3 are the top prognostic signatures shown in this study;*
 - c- *In this analysis, we have not evaluated the predictive power of the other two CSTB peptides (Pep13 and Pep14) since only peptides that passed through the filtering stage were evaluated by cross-validation;*
 - d- *To answer if Pep13 and Pep14 have the same predictive performance than Pep12, we now evaluated their AUC and it was of 0.376 and 0.264 for Pep14 and Pep13, very low in comparison with Pep12.*
 - e- *We have also tested combining the peptides (Pep12, Pep13, Pep14) and it resulted in 0.675 of AUC, still very low in comparison with the other signatures.*
 - f- *This pattern of lowering the AUC when combining all peptides from the same protein was found in many other signatures: some peptide signatures have higher AUC when combining only one peptide from different proteins, as happens in the highest AUCs found (S1, S2, and S3 (Figure 7, Table 3).*

Still, the SRM method is reporting lower CSTB protein level in saliva from N+ patients (Figure 6a) which is difficult to explain/understand. The statistical test chi-square did not support a connection between CSTB and lymph node metastasis (line 360-367).

Response:

- i) Regarding this comment: “SRM method is reporting lower CSTB protein level in saliva from N+ patients (Figure 6a) which is difficult to explain/understand”, we need to contextualize that to the relationship between the discovery and verification phases: as we demonstrated in the first figure of the manuscript, the verification phases were performed in parallel, and therefore, we verified the candidate proteins using both tissue by IHC and saliva by SRM-MS. IHC was chosen as one of the verification phases because of its broad applications in the routine practice of tissue histopathology and SRM-MS of saliva was chosen because saliva represents a promising non-invasive tool to be evaluated in clinical practice. With that in mind, our experimental design for the verification phase using saliva is somewhat independent of IHC results. In addition, in the SRM-MS analysis, we chose to divide patients into two groups, without and with lymph node-compromised, because it is the most important prognostic factor in oral cancer.
- ii) Besides, the CSTB levels have a strong association with relapse in IHC (cross-tabulation and chi-square test, Kaplan-Meier and Cox proportional hazard model), but it did not show association with lymph node metastasis in IHC analysis (Figure 5, Tables S22 and 23). This in fact corroborated with SRM-MS and clinicopathological analysis (cross-tabulation and chi-square test), which also did not show any association between CSTB levels and lymph node metastasis (Table S30). Therefore, the fact that CSTB is in low abundance in saliva and it does not have association with lymph node metastasis is in line with previous findings.
- iii) However, as commented in a previous question, the ability of the peptides to have different behavior even from the same proteins lead us to perform all the analysis at peptide level as suggested by the reviewer 1. One peptide of CSTB, combined with two other peptides of LTA4H and COL6A1 were able to differentiate the groups of patients, N0 and N+. With that, it is very interesting, that in fact CSTB also has a link with lymph-node metastasis, but at peptide level.
- iv) We are not sure the reviewer is thinking in the direction that CSTB is an independent recurrence marker and that the groups in saliva samples should represent this clinical data. Anyway, to avoid any expectation in the manuscript regarding defining patient group according to local recurrence for the SRM-MS analysis, we included the phrase in the revised manuscript: “Future studies could evaluate the CSTB, PGK1, LTA4H levels in saliva according to recurrence and/or second primary tumor, which were relevant

clinical data associated with their expression in the tissues (Table 1, Fig. 5) and are also clinical challenges for therapeutic management in OSCC.”

IV. Machine learning for prediction of lymph node metastasis:

Based on machine learning the authors propose predictive power of peptides, the top one being the single CSTB peptide that differed between N0 and N+ patients.

Response: The reviewer is correct; CSTB peptide is shown as the top predictive power peptide (Figure 7c), according to the estimation made by the cross-validation if only single peptides' AUCs are compared. In this case, CSTB peptide has an AUC of 73.5%. The goal of this analysis was to identify a combination/signature of peptides that could best predict samples as N0 and N+. In that case, we show in Figure 7d-e that S2 composed of LTA4H, COL6A1, and CSTB's peptides has the best AUC (82.8%). Since the estimated AUC of signature S2 is higher than all signatures formed by a single and combined peptides and proteins, we concluded that CSTB peptide discriminates better the classes N0 and N+ when combined with LTA4H and COL6A1's peptides. This is an interesting result and suggests the importance of working with peptides when looking for a prognostic signature.

We would like to revise machine learning steps below to avoid partial interpretations of results step-by-step:

- i. Pep12 (CSTB) has a higher AUC when combined with Pep8 (LTA4H) and Pep9 (COL6A1) (Figure 7c). We chose the best signature based on ROC AUC estimated by repeated cross-validation. AUC was used as a criterion because we were working with unbalanced classes.*
- ii. In the next step, we simulated a case in which the classes are balanced, and the number of samples increased, by adding synthetic samples, to evaluate the behavior of the signature ranks. S2 was still selected as the best predictor. We estimated that by building a classifier considering all the training samples (32), oversampling the training set using SMOTE to balance the classes, we could correctly assign 77.20% of the future samples, 76.90% of N0 samples and 77.80% of N+ samples, with ROC AUC of 85.90%. The estimation was performed by a repeated stratified 10-fold cross-validation, resulting in 1,000 models fitted with balanced training subsets and tested using only original samples (non-synthetic).*
- iii. We have shown, in addition to the other results and by machine learning methods, that the combination of the specific peptides of LTA4H, CSTB, and COL6A1, can be used to distinguish patients as N0 and N+. A larger training set would be necessary, as well as more validation steps.*

Regarding CSTB peptide:

- i. CSTB peptide alone (Pep12) presented higher ROC AUC than other single peptides.*
- ii. On the other hand, CSTB peptide (Pep12) contributed with Pep8 and Pep9 to form the signature S2, which together with S1 and S3 are the top prognostic signatures shown in this study.*

- iii. *In this analysis, we have not evaluated the predictive power of the other two CSTB peptides (Pep13 and Pep14) since only peptides that passed through the filtering stage were evaluated by cross-validation.*
- iv. *To answer if Pep13 and Pep14 have the same predictive performance than Pep12, we now evaluated their ROC AUC, and it was of 0.376 and 0.264 for Pep14 and Pep13, respectively, extremely low in comparison with Pep12.*
- v. *We have also tested the combination of peptides (Pep12, Pep13, Pep14) and it resulted in 0.675 of AUC, still very low in comparison with the other signatures.*
- vi. *This pattern of lowering the AUC when combining all peptides from the same protein was found in many other signatures: some peptide signatures have higher AUC when combining only one peptide from different proteins, as happens in the highest AUCs found (S1, S2, and S3 (Figure 7, Table 3)).*

V. In summary, it is still not clear if CSTB is expressed higher in inner tumor, or if there is a connection to lymph node metastasis.

Response: We think that the reviewer summarized here the same questions that were raised above, which lead us to give similar answer:

We need to contextualize that the relationship between the discovery and verification phase:

- i) *The discovery phase was designed considering the prognostic value of the invasive tumor front (ITF) region of neoplastic islands and tumor stroma. Robust potential candidates with prognostic values were indicated. These candidates were evaluated in the verification phases.*
- ii) *The verification phase was performed in parallel, and therefore, we verified the candidate proteins using both tissue by IHC and saliva by SRM-MS. IHC was chosen as one of the verification phases because of its broad applications in the routine practice of tissue histopathology and SRM-MS of saliva was chosen because saliva represents a promising non-invasive tool to be evaluated in clinical practice. With that in mind, our experimental design for the verification phase using saliva is independent of IHC results. In addition, the SRM-MS analysis, we chose to divide patients into two groups, without and with lymph node-compromised, because it is the most important factor for prognosis in oral cancer (Larsen et al. 2009; Ho et al. 2017), and its presence is highly associated with a worse prognosis (Scully and Bagan 2009).*
- iii) *Besides, the CSTB levels have a strong association with relapse in IHC (cross-tabulation and chi-square test, Kaplan-Meier and Cox proportional hazard model), but it did not show association with lymph node metastasis in IHC analysis (Figure 5, Tables S22 and 23). This in fact corroborated with SRM-MS and clinicopathological analysis (cross-tabulation and chi-square test), which also did not show any association between CSTB levels and lymph node metastasis (Table S30). Therefore, the fact that CSTB is in low abundance in saliva and it does not have association with lymph node metastasis is in line with previous findings.*

- iv) *However, as commented in the previous question, the ability of the peptides to have different behavior even from the same proteins lead us to perform all the analysis at peptide level as suggested by the reviewer 1. One peptide of CSTB, combined with two other peptides of LTA4H and COL6A1 were able to differentiate the groups of patients, N0 and N+. With that, it is very interesting, that in fact CSTB also has a link with lymph-node metastasis, but at peptide level.*
- v) *We are not sure the reviewer is thinking in the direction that CSTB is an independent recurrence marker and that the groups in saliva samples should represent these clinical data. Anyway, to avoid any expectation in the manuscript regarding defining patient group according to local recurrence for the SRM analysis, we included the phrase in the revised manuscript: “Future studies could evaluate the CSTB, PGK1, LTA4H levels in saliva according to recurrence and/or second primary tumor, which were relevant clinical data associated with their expression in the tissues (Table 1, Fig. 5) and are also clinical challenges for therapeutic management in OSCC.”*

2. The findings reported have questionable clinical value for several reasons. Due to non-transparent reporting of findings and unclear support of statements as discussed above, the generalizability of these findings in clinical samples is difficult to assess. From a practical point of view it is difficult to envision how the proposed markers should be used in the clinic. Are the authors suggesting that SRM should be used on saliva samples to look for missing peptides in order to predict lymph node metastasis?

Response:

2.1. *We stated the workflow of this study to guide the reviewer to the relevant findings:*

- i) *Main objective of the study was to indicate prognostic markers for oral cancer;*
- ii) *We used two phases: discovery and verification/target phases;*
- iii) *Discovery phase: to indicate candidate proteins differentially expressed between invasive tumor front and inner tumor of neoplastic island and stroma tumor, considering the prognostic value of the invasive tumor front (ITF);*
- iv) *IHC was chosen as one of the verification phases because of its broad applications in routine clinical practice, and in parallel, SRM-MS of saliva was also performed, since saliva represents a promising non-invasive tool to be evaluated in clinical practice. For that, we analyzed clinically stratified saliva samples according to the most important prognostic factor for oral cancer associated with poor prognosis, which is the lymph node metastasis.*
- v) *Our findings underscored:*
 - *IHC reveals CSTB is an independent marker for local recurrence in patients with OSCC;*
 - *SRM-MS of saliva indicates the combination of specific peptides from LTA4H, COL6A1 and CSTB to distinguish N+ and N0 patients.*

2.2 *Before clinical implementation, extensive longitudinal study with large-sized independent patient cohorts is still necessary. Several examples in the literature show the importance of further steps of validation before claiming that the markers have clinical use (Surinova et al. 2015; Kim et al. 2016; Whiteaker et al. 2011; Rifai, Gillette,*

and Carr 2006) and therefore, we were very careful with the conclusion about clinical applications.

2.3. The reviewer issue **“envision how the proposed markers should be used in the clinic”** With no doubt, the reviewer’s suggestion is coherent, but at this stage of the study, it is not possible to suggest or affirm the end point of monitoring CSTB protein in oral cancer patients from a clinical point of view. CSTB was shown to be expressed at low expression levels in the ITF of neoplastic islands, and by multivariate analysis was considered as an independent marker for local recurrence in patients with OSCC. Again, before clinical implementation, extensive longitudinal study with large-sized independent patient cohorts is still necessary. The result shown here is significant, but it requires replication and validation. We firmly expect the validation of CSTB as a marker of local recurrence in a larger cohort because it has potential to be used in clinical practice to plan the treatment modality and treatment monitoring.

The same for SRM-MS of saliva, which indicates that the combination of specific peptides from LTA4H, COL6A1, and CSTB can predict patients as being N+ and N0. It is important to clarify the difference between the identification of this signature as a potential prognostic signature and the construction of a predictive prognostic model with this signature and apply it in a clinic. A model could be created in a future if we have a larger training set. The performance of a future predictive model was estimated in this study by repeated cross-validation. Still, it requires posterior analysis, due to variations that may occur on the training set distribution. We simulated the sample size increase and tested the prediction power of all possible signatures again. The simulation was performed using an oversampling technique named SMOTE, that created additional samples synthetically, in a way that classes were balanced. The repeated cross-validation was used in this step too, applying SMOTE on each training subset. Each of the 1,000 models was tested with different test subsets formed by only original samples (non-synthetic). Despite all the intentional variation on training subsets caused by the repeated cross-validation and by adding synthetic data, S2 still was selected as the best signature for prediction.

It is important to highlight that the targeted phase of immunohistochemistry and SRM-MS in independent cohorts verified prognostic signature markers that may have applications in the routine clinical practice. The combination of tissue histopathology and very promising non-invasive biofluid saliva may drive prognostic decisions that can contribute to precise treatment protocols and reduction of tumor local relapse or lymph node metastasis.

With that in mind, we stated that the results are robust and confident for the verification phase to make a bridge with the next step of prognostic biomarkers validation for oral cancer. This path of biomarker discovery is in agreement with the literature (Whiteaker et al. 2011; Rifai, Gillette, and Carr 2006).

2.4. Regarding this question: **“Are the authors suggesting that SRM should be used on saliva samples to look for missing peptides in order to predict lymph node metastasis”**. It is important to make it clear to the reviewer our main accomplishments regarding the machine learning results and what we can state so far:

- i. We have shown, by machine learning methods, that the combination of the specific peptides of LTA4H+CSTB+COL6A1 (S2) is capable of distinguishing samples from the two classes N0 and N+ better than other possible signatures according to the

prediction scores estimated by the repeated cross-validation applied on the training set. To recall, the data used in this study is not missing peptides, but we worked here with peptide abundance (measured by the ratio light to heavy and ratio N0 to N+ of 40 patient saliva samples in a total of 239 MS run)).

- ii. Considering all the experiments and results involving the proteins and peptides of this signature, and the estimated prediction power of a classifier built considering the selected signature against the others, we believe that S2 can be used to build a classifier to predict the development of lymph node metastasis (N0 or N+) if a larger set of samples is considered to train the final model. The repeated cross-validation simulates the variation of samples that may occur when increasing the future training set. Also, we simulated the sample size increase and tested the prediction power of all possible signatures again. The simulation was performed using an oversampling technique named SMOTE, that created additional samples synthetically, in a way that classes were balanced. The repeated cross-validation was used in this step too, applying SMOTE on each training subset. Each of the 1,000 models was tested with different test subsets formed by only original samples (nonsynthetic). Despite all the intentional variation on training subsets caused by the repeated cross-validation and by adding synthetic data, S2 still was selected as the best signature.*
- iii. After numerous tests, performed by repeated cross-validation and class balancing, with the available samples in the current study, we obtained 85.9% AUC, with 77.2% of accuracy, 76.9% of sensitivity and 77.8% of specificity for S2 (LTA4H+CSTB+COL6A1).*
- iv. We have found the signature with the highest prediction scores. Each possible combination of features (signature) was tested by creating thousands of models using different sets of samples as a training set and tested by the remaining samples in each iteration of a repeated stratified 10-fold cross-validation. That is, we have estimated, by using different samples as the training set to fit classification models, that the use of signature S2 results in 82.8% of ROC AUC (AUROC). To test our AUROC estimation, we have built a final classification model using all the 32 samples from the training set and the signature S2. We tested this model using the 8 samples from the independent test set, which were not used to find the best signature. The result is 86.7% of AUC, which is in between the AUROC standard deviation estimated by cross-validation. Finally, we tested if balancing the classes by the imputation of new synthetic samples, estimated by SMOTE technique, it would increase the AUC of a predictor model. Applying the same repeated cross-validation method, we estimated that applying SMOTE on each of the training subsets, balancing the classes, increases the prediction power from 82.8% to 85.9% of AUROC.*
- v. Therefore, we have found a peptide prognostic signature with estimated good prediction performance (AUROC > 80%) on the task of classifying new samples as N+ and N0. We reinforce that still extensive longitudinal study with large-sized patient cohorts is still necessary to further be able to build a classifier to predict the development of lymph node metastasis (N0 and N+) using the signature proposed here, and new tests are required before clinical implementation.*

2.3. To further state and make clearer and more objective our findings, we improved the conclusion of the revised manuscript: “In summary, the discovery phase enabled us to spatially map the proteome of neoplastic islands and their surrounding stroma of

oral squamous cell carcinoma, identifying potential proteins with prognostic value. The targeted phases of immunohistochemistry and SRM-MS in independent cohorts verified prognostic signature markers that may have applications in routine clinical practice of tissue histopathology and in very promising non-invasive biofluid saliva, driving prognostic decisions that may lead to precise treatment protocols and reduction of tumor local relapse or lymph node metastasis. Extensive longitudinal study with large-sized independent patient cohorts is still necessary before clinical implementation. Here we indicate a robust prognostic signature with CSTB, at low protein expression levels in the invasive tumor front, as an independent marker for local recurrence. Also, we report a signature formed by LTA4H, COL6A1, and CSTB specific peptides, which has the best prediction performance among all possible combinations of peptides and proteins tested. We believe this signature can be used to build a predictor to distinguish patients with and without lymph node metastasis if large-sized independent patient samples are considered to train the model”.

3. From a methods point of view, none of the used analytical procedures are novel. The analytical depth of the analysis is far from comprehensive, most current high-quality clinical proteomics studies based on tissue samples quantify at least 6000 proteins across all samples. Here, the overlap analysis was based on a few hundred proteins. Even for micro-dissected samples these numbers are low.

Response: Considering the comments regarding the number of identified proteins, we revisited some recent studies and it is not very easy to find the complete information in the manuscripts that combined FFPE laser microdissection + similar area + similar type of tissues, considering the isolated island and stroma + detailed data processing that was used in this study to be able to have a real comparison.

Although the number of identified proteins does not always reflect the biological value of the proteins, we selected a recent important study of Mathias Mann group (Marakalala et al. 2016), using laser microdissection of FFPE tissues, showed a total of 4,406 proteins that were identified across five proteomes and an average of ~95% protein identifications were shared between at least two proteomes. In addition, they showed a hierarchical clustering analysis of 2,529 LFQ-protein intensities that were quantified in at least two of the five granuloma proteomes. However, unfortunately the area of laser microdissection was not clearly informed. Although we cannot compare with our study directly due to several reasons of different tissues (stroma and island vs total area), different isolated areas, in-gel digestion vs in-solution digestion, different amount of samples injected, different data processing in terms of filtering of valid values, we can point out that we provided in this study comparable results. We isolated an average microdissected tissue area of 0.1 mm², and for small neoplastic islands, 1 mm² for large neoplastic islands, and 1 mm² for stroma. Notably, this analysis was performed without peptide fractionation step.

The reviewer 1 translated this observation “While this is not completely novel, the overall package and rigor is definitely novel and a number of potentially interesting observations were made” in a more positive way. We believe that proteomics field instead of creating subjective thresholds needs to urgently contribute to robust candidate markers for future clinical implementation.

*This study shows that we were able to identify a panel of **novel** prognostic signature candidates going from a defined question in the discovery phase to verification in targeted phases. The findings of this study are original in the sense that this is the first study that CSTB is revealed as an independent marker for local*

recurrence in patients with OSCC and, the combination of LTA4H, COL6A1, and CSTB specific peptides in saliva is the most relevant prognostic signature to distinguish N+ and NO patients. Altogether, these results will be of interest to the field of clinical proteomics and head and neck cancer.

4. In general it is very difficult to follow the reasoning of the study. The reason is that too much text mass is spend on describing details when not needed (e.g. when reporting findings of questional value from clinicopathological data or when describing machine learning method details that are better fitted in the methods section). Conversely, when a more detailed description is warranted it is lacking (e.g. more transparent and clear description of the statistical and clinical value of specific markers).

Response:

4.1- Regarding the comment “too much text”, we can explain that the second version of the manuscript increased considerably due to additional experiments that the Reviewer 1 suggested. The Reviewer 2 also suggested that we included a better description regarding the 7 prognostic markers, leading the addition of 2 full paragraphs in the discussion. Among the changes we performed after the first review, we included one session of protein candidate prioritization, machine learning analysis and discussion of the seven candidate proteins. With that, the manuscript text increased about 21%. In the current review, we performed further revision and changed the machine learning methods to the supplementary material and also decreased the text regarding the description of clinical data, once they are already reported in the tables, figures and respective legends.

4.2- We restated below the workflow of the study to guide the reviewer to understand the meaning and clinical value of the study:

- i) Main objective of the study was to indicate prognostic markers for oral cancer;*
- ii) We used two phases: discovery and verification/target phases;*
- iii) Discovery phase: to indicate candidate proteins based on differential expression between invasive tumor front and inner tumor of neoplastic island and stroma, considering the prognostic value of the invasive tumor front (ITF);*
- iv) IHC was chosen as one of the verification phases because of its broad applications in routine clinical practice, and in parallel, SRM-MS of saliva was also selected, since saliva represents a promising non-invasive tool to be evaluated in clinical practice. For that, we analyzed clinically stratified saliva samples according to the most important prognostic factor for oral cancer associated with poor prognosis.*
 - IHC reveals CSTB as an independent marker for local recurrence in patients with OSCC;*
 - SRM-MS of saliva indicates the combination of peptides of LTA4H, COL6A1, and CSTB in saliva can distinguish N+ and NO patients. The peptide signature shown in this study can be used to construct a classifier to predict the development of lymph node metastasis (NO or N+) in a new set of patients;*
- v) We strongly expect the validation of these promising candidate markers in a larger cohort, because both CSTB level in tissue and LTA4H, COL6A1 and CSTB specific peptide levels in saliva can be used in clinical practice*

to assist planning the treatment modality/treatment monitoring and the reduction of tumor recurrence or lymph node metastasis.

vi) Before clinical implementation, extensive longitudinal study with large-sized independent patient cohorts is still necessary.

4.3- We included the detailed analysis of the data presented in Table 1. We also described above the step-by-step performed in this study to guide the interpretation of IHC.

4.4- It is not clear to the authors, what the reviewer means in terms of transparency, because we made available all the discovery proteomics data in ProteomeXchange Consortium (<http://proteomecentral.proteomexchange.org>) via the PRIDE partner repository and the SRM data are in Panorama with password created for the reviewers. In addition, after the first review, we have in total 39 supplementary tables and 14 supplementary figures.

We hope that we have addressed the concerns of Reviewer 3 in order to consider our study for publication in Nature Communications.

Sincerely,

Adriana Franco Paes Leme
Laboratório Nacional de Biociências, LNBio, CNPEM
Campinas, SP, Brazil

References

- Bello, Ibrahim O., Marilena Vered, Dan Dayan, Alex Dobriyan, Ran Yahalom, Kalle Alanen, Pentti Nieminen, Saara Kantola, Esa Läärä, and Tuula Salo. 2011. "Cancer-Associated Fibroblasts, a Parameter of the Tumor Microenvironment, Overcomes Carcinoma-Associated Parameters in the Prognosis of Patients with Mobile Tongue Cancer." *Oral Oncology* 47 (1): 33–38. doi:10.1016/j.oraloncology.2010.10.013.
- Burt, T., K. S. Button, H. H.Z. Thom, R. J. Noveck, and M. R. Munafò. 2017. "The Burden of the 'False-Negatives' in Clinical Development: Analyses of Current and Alternative Scenarios and Corrective Measures." *Clinical and Translational Science* 10 (6): 470–79. doi:10.1111/cts.12478.
- Deutsch, Eric W., Henry Lam, and Ruedi Aebersold. 2008. "PeptideAtlas: A Resource for Target Selection for Emerging Targeted Proteomics Workflows." *EMBO Reports*. doi:10.1038/embor.2008.56.
- Fedchenko, Nickolay, and Janin Reifenrath. 2014. "Different Approaches for Interpretation and Reporting of Immunohistochemistry Analysis Results in the Bone Tissue - a Review." *Diagnostic Pathology*. doi:10.1186/s13000-014-0221-9.
- Gallien, Sebastien, Elodie Duriez, and Bruno Domon. 2011. "Selected Reaction Monitoring Applied to Proteomics." *Journal of Mass Spectrometry*. doi:10.1002/jms.1895.
- Ho, Allen S., Sungjin Kim, Mourad Tighiouart, Cynthia Gudino, Alain Mita, Kevin S. Scher, Anna Laury, et al. 2017. "Metastatic Lymph Node Burden and Survival in Oral Cavity Cancer." *Journal of Clinical Oncology* 35 (31): 3601–9. doi:10.1200/JCO.2016.71.1176.
- Kim, Yune, Vladimir Ignatchenko, Cindy Q. Yao, Irina Kalatskaya, Julius O. Nyalwidhe, Raymond S. Lance, Anthony O. Gramolini, et al. 2012. "Identification of Differentially Expressed Proteins in Direct Expressed Prostatic Secretions of Men with Organ-Confined Versus Extracapsular Prostate Cancer." *Molecular & Cellular Proteomics* 11 (12): 1870–84. doi:10.1074/mcp.M112.017889.
- Kim, Yune, Jouhyun Jeon, Salvador Mejia, Cindy Q. Yao, Vladimir Ignatchenko, Julius O. Nyalwidhe, Anthony O. Gramolini, et al. 2016. "Targeted Proteomics Identifies Liquid-Biopsy Signatures for Extracapsular Prostate Cancer." *Nature Communications* 7. doi:10.1038/ncomms11906.
- Krokhin, Oleg V. 2006. "Sequence-Specific Retention Calculator. Algorithm for Peptide Retention Prediction in Ion-Pair RP-HPLC: Application to 300- and 100-?? Pore Size C18 Sorbents." *Analytical Chemistry* 78 (22): 7785–95. doi:10.1021/ac060777w.
- Lange, Vinzenz, Paola Picotti, Bruno Domon, and Ruedi Aebersold. 2008. "Selected Reaction Monitoring for Quantitative Proteomics: A Tutorial." *Molecular Systems Biology* 4 (1): 222. doi:10.1038/msb.2008.61.
- Larsen, S R, J Johansen, J A Sørensen, and A Krogdahl. 2009. "The Prognostic Significance of Histological Features in Oral Squamous Cell Carcinoma." *J Oral Pathol Med* 38 (December 2004): 657–62. doi:10.1111/j.1600-0714.2009.00797.x.
- Marakalala, Mohlopheni J, Ravikiran M Raju, Kirti Sharma, Yanjia J Zhang, Eliseo A Eugenin, Brendan Prideaux, Isaac B Daudelin, et al. 2016. "Inflammatory Signaling in Human Tuberculosis Granulomas Is Spatially Organized." *Nature*

- Medicine* 22 (5): 531–38. doi:10.1038/nm.4073.
- Pascovici, Dana, David C.L. Handler, Jemma X. Wu, and Paul A. Haynes. 2016. “Multiple Testing Corrections in Quantitative Proteomics: A Useful but Blunt Tool.” *Proteomics* 16 (18): 2448–53. doi:10.1002/pmic.201600044.
- Rifai, Nader, Michael A Gillette, and Steven A Carr. 2006. “Protein Biomarker Discovery and Validation: The Long and Uncertain Path to Clinical Utility.” *Nature Biotechnology* 24 (8): 971–83. doi:10.1038/nbt1235.
- Rodriguez, Jesse, Nitin Gupta, Richard D. Smith, and Pavel A. Pevzner. 2008. “Does Trypsin Cut before Proline?” *Journal of Proteome Research* 7 (1): 300–305. doi:10.1021/pr0705035.
- Scully, Crispian, and Jose Bagan. 2009. “Oral Squamous Cell Carcinoma Overview.” *Oral Oncology* 45 (4–5): 301–8. doi:10.1016/j.oraloncology.2009.01.004.
- Shackelford, Cynthia, Gerald Long, Jeffrey Wolf, Carlin Okerberg, and Ronald Herbert. 2002. “Qualitative and Quantitative Analysis of Nonneoplastic Lesions in Toxicology Studies.” *Toxicologic Pathology* 30 (1): 93–96. doi:10.1080/01926230252824761.
- Storey, John D., Jonathan E. Taylor, and David Siegmund. 2004. “Strong Control, Conservative Point Estimation and Simultaneous Conservative Consistency of False Discovery Rates: A Unified Approach.” *Journal of the Royal Statistical Society. Series B: Statistical Methodology* 66 (1): 187–205. doi:10.1111/j.1467-9868.2004.00439.x.
- Sundquist, Elias, Joonas H Kauppila, Johanna Veijola, Rayan Mroueh, Petri Lehenkari, Saara Laitinen, Juha Risteli, et al. 2017. “Tenascin-C and Fibronectin Expression Divide Early Stage Tongue Cancer into Low- and High-Risk Groups.” *British Journal of Cancer* 116 (5): 640–48. doi:10.1038/bjc.2016.455.
- Surinova, Silvia, Meena Choi, Sha Tao, Peter J Schüffler, Ching-Yun Chang, Timothy Clough, Kamil Vysloužil, et al. 2015. “Prediction of Colorectal Cancer Diagnosis Based on Circulating Plasma Proteins.” *EMBO Molecular Medicine* 7 (9): 1166–78. doi:10.15252/emmm.201404873.
- Wang, Qing, Ming Zhang, Tyler Tomita, Joshua T. Vogelstein, Shibin Zhou, Nickolas Papadopoulos, Kenneth W. Kinzler, and Bert Vogelstein. 2017. “Selected Reaction Monitoring Approach for Validating Peptide Biomarkers.” *Proceedings of the National Academy of Sciences* 114 (51): 13519–24. doi:10.1073/pnas.1712731114.
- Whiteaker, Jeffrey R., Chenwei Lin, Jacob Kennedy, Liming Hou, Mary Trute, Izabela Sokal, Ping Yan, et al. 2011. “A Targeted Proteomics-Based Pipeline for Verification of Biomarkers in Plasma.” *Nature Biotechnology* 29 (7): 625–34. doi:10.1038/nbt.1900.

REVIEWERS' COMMENTS:

Reviewer #1 (Remarks to the Author):

**Manuscript has been further improved.
All my concerns have been addressed.**